# Boosting Ray Search Procedure of Hard-label Attacks with Transfer-based Priors

**Chen Ma** [1,2]    **Xinjie Xu** [1]    **Shuyu Cheng** [3]    **Qi Xuan** [1,2]

[1] Institute of Cyberspace Security, Zhejiang University of Technology, Hangzhou 310023, China
[2] Binjiang Institute of Artificial Intelligence, ZJUT, Hangzhou 310056, China
[3] JQ Investments, Shanghai 200122, China
{machen,xuanqi}@zjut.edu.cn, xxj1018@foxmail.com, csy530216@126.com

## Abstract

One of the most practical and challenging types of black-box adversarial attacks is the hard-label attack, where only the top-1 predicted label is available. One effective approach is to search for the optimal ray direction from the benign image that minimizes the $\ell_p$ norm distance to the adversarial region. The unique advantage of this approach is that it transforms the hard-label attack into a continuous optimization problem. The objective function value is the ray's radius, which can be obtained via binary search at a high query cost. Existing methods use a "sign trick" in gradient estimation to reduce the number of queries. In this paper, we theoretically analyze the quality of this gradient estimation and propose a novel prior-guided approach to improve ray search efficiency both theoretically and empirically. Specifically, we utilize the transfer-based priors from surrogate models, and our gradient estimators appropriately integrate them by approximating the projection of the true gradient onto the subspace spanned by these priors and random directions, in a query-efficient manner. We theoretically derive the expected cosine similarities between the obtained gradient estimators and the true gradient, and demonstrate the improvement achieved by incorporating priors. Extensive experiments on the ImageNet and CIFAR-10 datasets show that our approach significantly outperforms 11 state-of-the-art methods in terms of query efficiency.

## 1 Introduction

Adversarial attacks represent a major security threat to deep neural networks (DNNs), where subtle, imperceptible perturbations are crafted to cause misclassifications. To assess DNN robustness and uncover vulnerabilities, the research community has developed various adversarial attack strategies. As a result, adversarial attacks and defenses have become a focal point in AI security research.

Based on the available information about the target model, adversarial attacks can be broadly classified into white-box and black-box attacks. White-box attacks, such as those proposed by Madry et al. (2018); Moosavi-Dezfooli et al. (2016), rely on the target model's gradients with respect to the input, making them less practical in real-world applications. Black-box attacks, by contrast, are often more feasible, as they do not require knowledge of model parameters or gradients. As a subset of black-box attacks, transfer-based attacks generate adversarial examples using white-box models with the aim of generalizing to other models. While transfer-based attacks do not involve querying the target model, their success rate is inconsistent. Alternatively, query-based black-box attacks iteratively interact with the target model to achieve higher success rates. These attacks can be categorized into two subtypes: score-based and decision-based (also known as hard-label) attacks. Score-based attacks (Ma et al., 2021a) utilize the model's output logits to guide the attack, whereas hard-label attacks rely solely on top-1 predicted labels, making them particularly practical when only label information is accessible. In this work, we focus on the problem of reducing query complexity in hard-label attacks.

The difficulty of hard-label attacks is that the labels can only be flipped near the classification decision boundary, and thus the objective function is discontinuous. As a result, the attack requires solving a high-dimensional combinatorial optimization problem, which is challenging. Common approaches (Chen et al., 2020; Brendel et al., 2018) start with a sample containing large adversarial perturbations

and iteratively reduce the distortion by moving along the decision boundary towards a benign image. However, these methods lack convergence guarantees. To reformulate the problem as a continuous optimization task, ray-search methods have been introduced. Typical approaches such as OPT (Cheng et al., 2019), Sign-OPT (Cheng et al., 2020), and RayS (Chen & Gu, 2020) aim to minimize an objective function $g(\theta)$, which is defined as the shortest $\ell_p$ norm distance along the ray direction $\theta$ from the benign image to the adversarial region. This function value can be evaluated using a binary search method. Leveraging the smooth and continuous nature of decision boundaries, $g(\theta)$ is locally continuous, making it amenable to zeroth-order (ZO) optimization with a gradient estimator. OPT employs a random gradient-free (RGF) estimator, but it incurs high query cost due to the binary search in finite differences. Sign-OPT reduces the query complexity by using the sign of the directional derivative in gradient estimation, but it significantly sacrifices gradient accuracy.

To solve this problem and improve query efficiency, we employ the same objective function $g(\theta)$ and propose incorporating the transfer-based priors into gradient estimation. An ideal prior is the gradient of $g(\theta)$ from a surrogate model, but it cannot be easily obtained since $g(\theta)$ is non-differentiable due to the binary search. Instead, we propose a surrogate loss, whose gradient is proportional to that of $g(\theta)$, to obtain the prior. Once the transfer-based priors are obtained, we must design better gradient estimators that effectively integrate these priors. This is particularly challenging under the hard-label restriction, as accurately determining the value of $g(\theta)$ is costly. As a result, previous prior-guided methods for score-based attacks such as PRGF (Cheng et al., 2021; Dong et al., 2022) are not suitable in this context. Thus, we need to explore how to improve the gradient estimator with additional priors while minimizing queries. To achieve this, we propose two algorithms: Prior-Sign-OPT and Prior-OPT. They estimate the gradient in a query-efficient manner by approximating the projection of the true gradient onto a subspace spanned by priors and randomly sampled vectors. We provide a thorough theoretical analysis to validate their effectiveness and offer theoretical comparisons between Sign-OPT and our approach. In particular, Prior-OPT achieves a better approximation of the subspace projection with only slightly more queries, and can adaptively adjust the weight of each prior based on its quality, striking a balance between gradient accuracy and query efficiency. While several methods (Brunner et al., 2019; Shi et al., 2023) attempt to combine transfer- and decision-based attacks, they lack theoretical guarantees and often perform poorly. Crucially, in the hard-label setting, these approaches fail to effectively address the challenge of appropriately weighing the prior when it deviates significantly from the true gradient. Our approach resolves this issue and naturally scales to priors from multiple surrogate models, demonstrating further improvement in attack performance.

To summarize, our main contributions are as follows.

1. **Novelty in hard-label attacks.** We address the problem of introducing the transfer-based priors into hard-label attacks by employing the subspace projection approximation, which significantly improves the accuracy of gradient estimation with slightly more queries. Our approach not only strikes a balance between gradient estimation and query efficiency, but also elegantly integrates priors from multiple surrogate models to further improve performance.

2. **Novelty in theoretical analysis.** We analyze the quality of our gradient estimators and that of the orthogonal variant of Sign-OPT, enabling theoretical comparisons. To our knowledge, this is the first work to derive the expected cosine similarities between estimators of the Sign-OPT family and the true gradient, theoretically guaranteeing performance improvement.

3. **Extensive experiments.** Extensive experiments conducted on the ImageNet and CIFAR-10 datasets show that our approach outperforms 11 state-of-the-art methods significantly.

## 2  RELATED WORK

Hard-label attacks can be categorized into boundary-search and ray-search approaches.

The boundary-search approaches start from a large perturbation or an image of the target class and then reduce distortions by iteratively moving along the decision boundary towards the original image. Boundary Attack (BA) (Brendel et al., 2018) is an early representative method, and its query efficiency is relatively low. HopSkipJumpAttack (HSJA) (Chen et al., 2020) estimates the gradient at the decision boundary to update the sample and then finds the next boundary point by moving it towards the benign image. Tangent Attack (TA) and Generalized Tangent Attack (G-TA) (Ma et al., 2021b) find an optimal tangent point on a virtual hemisphere or semi-ellipsoid to efficiently generate

the adversarial example. CGBA (Reza et al., 2023) conducts a boundary search along a semicircular path on a restricted 2D plane to find the boundary point. To avoid gradient estimation, SurFree (Maho et al., 2021) and Triangle Attack (Wang et al., 2022) find the adversarial example in a DCT subspace to improve query efficiency. Evolutionary (Dong et al., 2019) adopts the (1+1)-CMA-ES, a simple yet effective variant of Covariance Matrix Adaptation Evolution Strategy, to efficiently generate adversarial examples. Adaptive History-driven Attack (AHA) (Li et al., 2021) gathers data of previous queries as the prior for current sampling, which improves the random walk optimization.

The ray-search approaches aim to find an optimal direction $\theta$ that reaches the nearest adversarial region. As mentioned in Section 1, it is challenging to address both the high query complexity issue of OPT and the low estimation accuracy issue of Sign-OPT. RayS (Chen & Gu, 2020) avoids gradient estimation and employs a hierarchical search step to efficiently find the optimal direction. However, RayS only supports untargeted $\ell_\infty$-norm attacks. Since the query efficiency of previous ray-search approaches has not surpassed that of boundary-search methods, they have attracted less research interest and remain insufficiently studied. We note that the mechanisms of OPT and Sign-OPT remain poorly understood, and their inefficiency stems from the limited precision in gradient estimation.

Several methods attempt to combine transfer- and decision-based attacks, but the critical issue, namely how to weigh the prior when it deviates significantly from the true gradient, has not been well addressed. For example, Biased Boundary Attack (BBA) (Brunner et al., 2019), Customized Iteration and Sampling Attack (CISA) (Shi et al., 2023) and Small-Query Black-Box Attack (SQBA) (Park et al., 2024) set the prior's coefficient empirically rather than through theoretical analysis. In contrast, our approach dynamically calculates optimal coefficients, improving gradient estimation accuracy.

# 3 THE PROPOSED APPROACH

## 3.1 THE GOAL OF HARD-LABEL ATTACKS

Given a $k$-class classifier $f : \mathbb{R}^d \to \mathbb{R}^k$ and a benign image $\mathbf{x} \in [0, 1]^d$ which is correctly classified by $f$, the adversary aims to find an adversarial example $\mathbf{x}_{\text{adv}}$ with the minimum perturbation such that $f(\mathbf{x}_{\text{adv}})$ outputs an incorrect prediction. Formally, we formulate the attack goal as:

$$\min_{\mathbf{x}_{\text{adv}}} d(\mathbf{x}_{\text{adv}}, \mathbf{x}) \quad \text{s.t.} \quad \Phi(\mathbf{x}_{\text{adv}}) = 1, \tag{1}$$

where $d(\mathbf{x}_{\text{adv}}, \mathbf{x}) := \|\mathbf{x}_{\text{adv}} - \mathbf{x}\|_p$ is the $\ell_p$ norm distortion, and $\Phi(\cdot)$ is a success indicator function:

$$\Phi(\mathbf{x}_{\text{adv}}) := \begin{cases} 1 & \text{if } \hat{y} = y_{\text{adv}} \text{ in the targeted attack,} \\ & \text{or } \hat{y} \neq y \text{ in the untargeted attack,} \\ 0 & \text{otherwise,} \end{cases} \tag{2}$$

where $\hat{y} = \arg\max_{i \in \{1, \dots, k\}} f(\mathbf{x}_{\text{adv}})_i$ is the top-1 predicted label of $f$, $y \in \mathbb{R}$ is the true label of $\mathbf{x}$, and $y_{\text{adv}} \in \mathbb{R}$ is a target class label. In this study, we follow Cheng et al. (2019; 2020) to reformulate the problem (1) as the problem of finding the ray direction of the shortest distance from $\mathbf{x}$ to the adversarial region:

$$\min_{\theta \in \mathbb{R}^d \setminus \{\mathbf{0}\}} g(\theta) \quad \text{where} \quad g(\theta) := \inf \left\{ \lambda : \lambda > 0, \Phi\left(\mathbf{x} + \lambda \frac{\theta}{\|\theta\|}\right) = 1 \right\}. \tag{3}$$

Note that $g(\theta) = +\infty$ when the set is empty, since $\inf \emptyset = +\infty$ by convention. Finally, the adversarial example is $\mathbf{x}^* = \mathbf{x} + g(\theta^*) \frac{\theta^*}{\|\theta^*\|}$, and $\theta^*$ is the optimal solution of problem (3).

## 3.2 THE OPTIMIZATION OF SEARCHING RAY DIRECTIONS

The previous works (Cheng et al., 2019; 2020) attempt to optimize the problem (3) by using ZO methods. However, the restriction of hard-label access results in a high query cost of the gradient estimation, because obtaining a single value of $g(\theta)$ requires performing a binary search with multiple queries, and the gradient estimation with finite difference requires multiple computations of $g(\theta)$. Sign-OPT (Cheng et al., 2020) replaces the finite-difference term $g(\theta + \sigma\mathbf{u}) - g(\theta)$ with $\text{sign}(g(\theta + \sigma\mathbf{u}) - g(\theta))$, which improves query efficiency by only using a single query (Eq. (8)). However, it significantly reduces the accuracy of the gradient estimation. We propose to incorporate transfer-based priors to enhance accuracy without significantly increasing query complexity, thus achieving an optimal balance between query complexity and estimation accuracy.

The first challenge is how to obtain a transfer-based prior $\nabla g_{\hat{f}}(\theta)$ from a surrogate model $\hat{f}$, where $g_{\hat{f}}(\theta)$ represents the shortest distance along the direction $\theta$ to the adversarial region of $\hat{f}$. This is challenging because $g_{\hat{f}}(\theta)$ is typically evaluated using binary search, making it non-differentiable. To address this, for any non-zero vector $\theta_0 \in \mathbb{R}^d$ such that $g_{\hat{f}}(\theta_0) < +\infty$, we define a surrogate function $h(\theta, \lambda)$ such that $\nabla g_{\hat{f}}(\theta_0) = c \cdot \nabla_\theta h(\theta_0, \lambda_0)$, where $\lambda_0 = g_{\hat{f}}(\theta_0)$ is treated as a constant during differentiation. Here, $\lambda$ is a scalar, and $c$ is a non-zero constant. The surrogate function $h(\theta, \lambda)$ is defined as the negative Carlini & Wagner (C&W) loss function of $\hat{f}$:

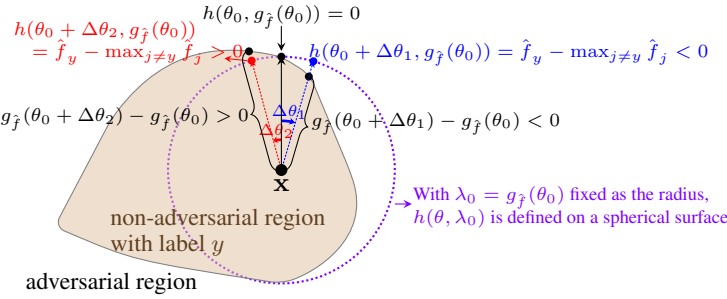

Figure 1: Geometrical explanation of $\nabla g_{\hat{f}}(\theta_0) \propto \nabla_\theta h(\theta_0, \lambda_0)$ by taking an untargeted attack as an example. When $g_{\hat{f}}(\theta)$ reduces/increases with a small $\Delta\theta$, $h(\theta, \lambda)$ changes at a similar rate. The formal proof is in Appendix A.

$$h(\theta, \lambda) := \begin{cases} \hat{f}_y - \max_{j \neq y} \hat{f}_j, & \text{if untargeted attack,} \\ \max_{j \neq \hat{y}_{\text{adv}}} \hat{f}_j - \hat{f}_{\hat{y}_{\text{adv}}}, & \text{if targeted attack,} \end{cases} \tag{4}$$

where $\hat{f}_i := \hat{f}\left(\mathbf{x} + \lambda \cdot \frac{\theta}{\|\theta\|}\right)_i$ is an abbreviation for the $i$-th element of the output of $\hat{f}$, and $\mathbf{x}$ is the original image. Any non-zero scalar can be used as $\lambda$, but the specific value $\lambda_0 = g_{\hat{f}}(\theta_0)$ yields a gradient proportional to $\nabla g_{\hat{f}}(\theta_0)$. The value $\lambda_0$ is obtained through binary search, where $h(\theta_0, \lambda_0)$ represents the negative C&W loss at the decision boundary of the surrogate model $\hat{f}$. The geometric explanation and formal proof of $\nabla g_{\hat{f}}(\theta_0) = c \cdot \nabla_\theta h(\theta_0, \lambda_0)$ are presented in Fig. 1 and Appendix A.

In targeted attacks, determining an appropriate $\lambda_0$ value becomes a challenging task. This is because the spatial distribution of classification regions, along with the shape and extent of the decision boundaries, varies across different models. Although we can locate the region corresponding to the predefined target class $y_{\text{adv}}$ along the $\theta$ direction in the target model $f$, the same direction may not lead to the region of the class $y_{\text{adv}}$ in a surrogate model $\hat{f}$. Therefore, we must set a new target class $\hat{y}_{\text{adv}}$ before determining $\lambda_0$ and computing Eq. (4). See Appendix B for detailed steps.

Given $s$ non-zero vectors $\mathbf{k}_1, \ldots, \mathbf{k}_s$ representing transfer-based priors from $s$ surrogate models and $q - s$ randomly sampled vectors $\mathbf{r}_i \sim \mathcal{N}(\mathbf{0}, \mathbf{I})$ for $i = 1, \ldots, q - s$, our objective is to estimate a gradient $\mathbf{v}^* \approx \nabla g(\theta)$ as accurately as possible using these $q$ vectors. In *the score-based attack setting*, there is a subspace projection estimator theory (Meier et al., 2019; Cheng et al., 2021) that can solve this problem. Based on this theory, the optimal estimated gradient $\mathbf{v}^*$ that maximizes its similarity with the true gradient is given by Proposition 3.1 in the score-based setting[1].

**Proposition 3.1.** *(Optimality of the subspace projection estimator) Let $\mathbf{k}_1, \ldots, \mathbf{k}_s$ and $\mathbf{r}_1, \ldots, \mathbf{r}_{q-s}$ be defined above; let $S := \text{span}\{\mathbf{k}_1, \ldots, \mathbf{k}_s, \mathbf{r}_1, \ldots, \mathbf{r}_{q-s}\}$ denote the subspace spanned by these vectors. Then the optimal $\mathbf{v}^*$ in $S$ that maximizes $\overline{\nabla g(\theta)}^\top \mathbf{v}$ subject to $\|\mathbf{v}\| = 1$ is the $\ell_2$-normalized projection of $\nabla g(\theta)$ onto $S$, denoted as $\mathbf{v}^* := \overline{\nabla g(\theta)_S}$.*

According to Proposition 3.1, finding the optimal approximate gradient is equivalent to finding a projection of the true gradient onto a low-dimensional subspace $S$ spanned by all available vectors. The projection of a vector onto a subspace $S$ can be calculated by summing its projections onto the orthonormal basis of $S$. To achieve this, we construct an orthonormal basis of $S$ via Gram-Schmidt orthonormalization, which transforms $\mathbf{k}_1, \ldots, \mathbf{k}_s, \mathbf{r}_1, \ldots, \mathbf{r}_{q-s}$ into an orthonormal basis $\mathbf{p}_1, \ldots, \mathbf{p}_s, \mathbf{u}_1, \ldots, \mathbf{u}_{q-s}$. Note that $\mathbf{p}_1, \ldots, \mathbf{p}_s$ correspond to $\mathbf{k}_1, \ldots, \mathbf{k}_s$, and $\mathbf{u}_1, \ldots, \mathbf{u}_{q-s}$ correspond to $\mathbf{r}_1, \ldots, \mathbf{r}_{q-s}$. Then, we can compute the projection of $\nabla g(\theta)$ onto $S$ by Eq. (5):

$$\mathbf{v}^* = \sum_{i=1}^{s} \nabla g(\theta)^\top \mathbf{p}_i \cdot \mathbf{p}_i + \sum_{i=1}^{q-s} \nabla g(\theta)^\top \mathbf{u}_i \cdot \mathbf{u}_i. \tag{5}$$

---

[1]Throughout this paper, for any vector $\mathbf{x}$, we denote its $\ell_2$-normalized version by $\overline{\mathbf{x}}$, where $\overline{\mathbf{x}} := \mathbf{x}/\|\mathbf{x}\|$.

Given the queried function values, $\nabla g(\theta)^\top \mathbf{u}$ for the unit $\ell_2$-norm vector $\mathbf{u}$ can be approximated by the finite-difference method, without requiring backpropagation:

$$\nabla g(\theta)^\top \mathbf{u} \approx \frac{g(\theta + \sigma \mathbf{u}) - g(\theta)}{\sigma}, \tag{6}$$

where $\sigma$ is a small positive number. By plugging Eq. (6) into Eq. (5), we can easily calculate $\mathbf{v}^*$ in the score-based setting as $\mathbf{v}^* = \sum_{i=1}^{s} \frac{g(\theta + \sigma \mathbf{p}_i) - g(\theta)}{\sigma} \cdot \mathbf{p}_i + \sum_{i=1}^{q-s} \frac{g(\theta + \sigma \mathbf{u}_i) - g(\theta)}{\sigma} \cdot \mathbf{u}_i$. However, in *the hard-label setting*, the finite-difference requires a large number of queries due to the binary search of $g(\cdot)$. We propose two algorithms to reduce query cost by computing the approximate projection, i.e., Prior-Sign-OPT and Prior-OPT. With $s$ priors, Prior-Sign-OPT uses Eq. (7) to improve performance:

$$\mathbf{v}^* = \sum_{i=1}^{s} \mathrm{sign}(g(\theta + \sigma \mathbf{p}_i) - g(\theta)) \cdot \mathbf{p}_i + \sum_{i=1}^{q-s} \mathrm{sign}(g(\theta + \sigma \mathbf{u}_i) - g(\theta)) \cdot \mathbf{u}_i. \tag{7}$$

Eq. (7) is similar to the formula of Sign-OPT, benefiting from using only a single query to calculate the sign of the directional derivative (Cheng et al., 2020):

$$\mathrm{sign}(g(\theta + \sigma \mathbf{u}_i) - g(\theta)) = \begin{cases} +1, & f\left(\mathbf{x} + g(\theta)\frac{\theta + \sigma \mathbf{u}_i}{\|\theta + \sigma \mathbf{u}_i\|}\right) = y, \\ -1, & \text{otherwise.} \end{cases} \tag{8}$$

The accuracy of the estimated gradient is crucial in optimization. A natural way to assess accuracy is via the following metrics: $\mathbb{E}[\gamma]$ and $\mathbb{E}[\gamma^2]$, where $\gamma$ is the cosine similarity between the estimated and true gradients. We propose a novel approach to compute $\mathbb{E}[\gamma]$ and $\mathbb{E}[\gamma^2]$ for Sign-OPT, Prior-Sign-OPT, and Prior-OPT. Our baseline extends Sign-OPT (Cheng et al., 2020) by employing orthogonal random vectors, while retaining the original name to maintain consistency within the method family.

**Theorem 3.2.** *For the Sign-OPT estimator approximated by Eq.* (6) *(defined as Eq. (44)), we let* $\gamma := \overline{\mathbf{v}}^\top \nabla g(\theta)$ *be its cosine similarity to the true gradient, where* $\overline{\mathbf{v}} := \frac{\mathbf{v}}{\|\mathbf{v}\|}$, *then*

$$\mathbb{E}[\gamma] = \sqrt{q}\frac{\Gamma(\frac{d}{2})}{\Gamma(\frac{d+1}{2})\sqrt{\pi}}, \tag{9}$$

$$\mathbb{E}[\gamma^2] = \frac{1}{d}\left(\frac{2}{\pi}(q-1) + 1\right). \tag{10}$$

The proof of Theorem 3.2 is included in Appendix C.1. For Prior-Sign-OPT, we have Theorem 3.3.

**Theorem 3.3.** *For the Prior-Sign-OPT estimator approximated by Eq.* (6) *(defined as Eq. (82)), we let* $\gamma := \overline{\mathbf{v}^*}^\top \overline{\nabla g(\theta)}$ *be its cosine similarity to the true gradient, where* $\overline{\mathbf{v}^*} := \frac{\mathbf{v}^*}{\|\mathbf{v}^*\|}$, *then*

$$\mathbb{E}[\gamma] = \frac{1}{\sqrt{q}}\left[\sum_{i=1}^{s}|\alpha_i| + (q-s)\sqrt{1 - \sum_{i=1}^{s}\alpha_i^2} \cdot \frac{\Gamma(\frac{d-s}{2})}{\Gamma(\frac{d-s+1}{2})\sqrt{\pi}}\right], \tag{11}$$

$$\mathbb{E}[\gamma^2] = \frac{1}{q}\left[\left(\sum_{i=1}^{s}|\alpha_i|\right)^2 + \frac{q-s}{d-s}\left(\frac{2}{\pi}(q-s-1) + 1\right)\left(1 - \sum_{i=1}^{s}\alpha_i^2\right)\right.$$
$$\left. + 2\left(\sum_{i=1}^{s}|\alpha_i|\right)(q-s)\sqrt{1 - \sum_{i=1}^{s}\alpha_i^2} \cdot \frac{\Gamma(\frac{d-s}{2})}{\Gamma(\frac{d-s+1}{2})\sqrt{\pi}}\right], \tag{12}$$

*where* $\alpha_i := \mathbf{p}_i^\top \overline{\nabla g(\theta)}$ *is the cosine similarity between the $i$-th prior and the true gradient.*

The proof of Theorem 3.3 is presented in Appendix C.2. Now we can compare $\mathbb{E}[\gamma]$ of Sign-OPT (Eq. (9)) and Prior-Sign-OPT (Eq. (11)). In Sign-OPT, applying Jensen's inequality yields the bound $\mathbb{E}[\gamma] \leq \sqrt{\mathbb{E}[\gamma^2]} = \sqrt{(2(q-1) + \pi)/(\pi d)}$. When $q \ll d$, $\mathbb{E}[\gamma]$ becomes very small, resulting in poor performance. In contrast, Prior-Sign-OPT with a single prior can improve performance. For instance, when attacking an image of size $32 \times 32 \times 3$, and using parameters $q = 200$ and $s = 1$, if $0.01422 \leq |\alpha_1| \leq 0.611$, $\mathbb{E}[\gamma]$ of Prior-Sign-OPT surpasses that of Sign-OPT. However, Prior-Sign-OPT may underperform Sign-OPT in certain cases, such as when $|\alpha_1| \geq 0.612$ in the example above, because it applies sign-based multipliers to both random vectors and priors. Intuitively, random

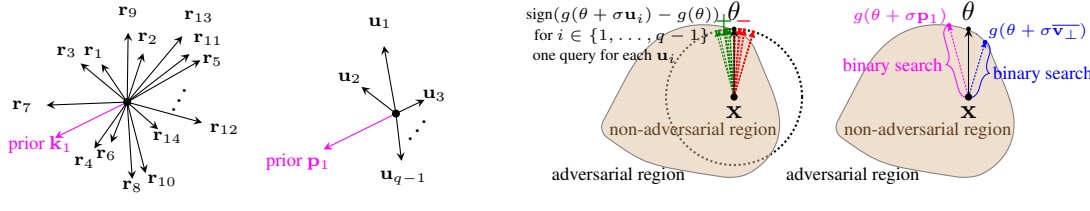

(a) Get prior $\mathbf{k}_1$ and sample $\mathbf{r}_i$.  (b) Orthonormal basis.  (c) Estimate a sign-based $\mathbf{v}_\perp$.  (d) Estimate $\mathbf{v}^*$ with $\mathbf{v}_\perp$ and $\mathbf{p}_1$.

Figure 2: Simplified two-dimensional illustration of the gradient estimation of Prior-OPT with a single transfer-based prior. In Fig. 2a, we first sample random vectors $\mathbf{r}_i \sim \mathcal{N}(\mathbf{0}, \mathbf{I})$ for $i = 1, \ldots, q-1$ and obtain a transfer-based prior $\mathbf{k}_1$ using $\nabla_\theta h(\theta, \lambda)$, where $h(\theta, \lambda)$ is defined in Eq. (4). Then, as shown in Fig. 2b, we perform Gram-Schmidt orthonormalization on these vectors to obtain an orthonormal basis $\mathbf{p}_1, \mathbf{u}_1, \ldots, \mathbf{u}_{q-1}$, where $\mathbf{p}_1 = \mathbf{k}_1$. Next, we estimate $\mathbf{v}_\perp$ based on Eq. (14) with $\mathbf{u}_1, \ldots, \mathbf{u}_{q-1}$ (Fig. 2c), where each $\text{sign}(g(\theta + \sigma\mathbf{u}_i) - g(\theta))$ requires only a single query (Eq. (8)). Finally, as shown in Fig. 2d, we estimate a gradient $\mathbf{v}^*$ based on Eq. (13) with $\mathbf{p}_1$ and $\mathbf{v}_\perp$, where the values of $g(\theta + \sigma\mathbf{p}_1)$ and $g(\theta + \sigma\overline{\mathbf{v}_\perp})$ are obtained via the binary search with multiple queries.

vectors $\mathbf{u}_1, \ldots, \mathbf{u}_{q-s}$ have relatively consistent cosine similarities with the true gradient as they are identically distributed, allowing for efficient sign-based estimation. In contrast, the cosine similarities between the priors $\mathbf{p}_1, \ldots, \mathbf{p}_s$ and the true gradient differ, requiring more precise estimation. To address this, we propose Prior-OPT that treats priors and random vectors differently. Fig. 2 illustrates the process of gradient estimation in Prior-OPT, which is based on the following formula:

$$\mathbf{v}^* = \sum_{i=1}^{s} \frac{g(\theta + \sigma\mathbf{p}_i) - g(\theta)}{\sigma} \cdot \mathbf{p}_i + \frac{g(\theta + \sigma\overline{\mathbf{v}_\perp}) - g(\theta)}{\sigma} \cdot \overline{\mathbf{v}_\perp}, \tag{13}$$

where $\overline{\mathbf{v}_\perp}$ is the $\ell_2$ normalization of $\mathbf{v}_\perp$, and $\mathbf{v}_\perp$ is obtained by:

$$\mathbf{v}_\perp := \sum_{i=1}^{q-s} \text{sign}(g(\theta + \sigma\mathbf{u}_i) - g(\theta)) \cdot \mathbf{u}_i. \tag{14}$$

Since random vectors $\mathbf{u}_1, \ldots, \mathbf{u}_{q-s}$ poorly align with $\nabla g(\theta)$, we aggregate them into a single vector $\mathbf{v}_\perp$ using a less accurate estimator in Eq. (14), which is orthogonal to all priors. Compared with Eq. (7), Eq. (13) provides a more accurate projection approximation. We now present Theorem 3.4.

**Theorem 3.4.** *For the Prior-OPT estimator approximated by Eq.* (6) *(defined as Eq.* (114)*), we let* $\gamma := \overline{\mathbf{v}^*}^\top \overline{\nabla g(\theta)}$ *be its cosine similarity to the true gradient, where* $\overline{\mathbf{v}^*} := \frac{\mathbf{v}^*}{\|\mathbf{v}^*\|}$, *then*

$$\mathbb{E}[\gamma] \geq \sqrt{\sum_{i=1}^{s} \alpha_i^2 + \frac{(q-s)(1 - \sum_{i=1}^{s} \alpha_i^2)}{\pi} \left( \frac{\Gamma(\frac{d-s}{2})}{\Gamma(\frac{d-s+1}{2})} \right)^2}, \tag{15}$$

$$\mathbb{E}[\gamma] \leq \sqrt{\sum_{i=1}^{s} \alpha_i^2 + \frac{1}{d-s} \left( \frac{2}{\pi}(q-s-1) + 1 \right) \left( 1 - \sum_{i=1}^{s} \alpha_i^2 \right)}, \tag{16}$$

$$\mathbb{E}[\gamma^2] = \sum_{i=1}^{s} \alpha_i^2 + \frac{1}{d-s} \left( \frac{2}{\pi}(q-s-1) + 1 \right) \left( 1 - \sum_{i=1}^{s} \alpha_i^2 \right), \tag{17}$$

*where* $\alpha_i := \mathbf{p}_i^\top \overline{\nabla g(\theta)}$ *is the cosine similarity between the $i$-th prior and the true gradient.*

The proof of Theorem 3.4 is included in Appendix C.3. $\mathbb{E}[\gamma^2]$ is the second-order moment, which reflects the magnitude of $\gamma$ in a statistical sense. Under certain assumptions, $\mathbb{E}[\gamma^2]$ directly affects the convergence rate of optimization algorithms (Cheng et al., 2021). Intuitively, a larger $\gamma$ indicates more accurate gradient estimation, leading to faster optimization and improved query efficiency. Next, we compare $\mathbb{E}[\gamma^2]$ for Prior-OPT (Eq. (17)) and Sign-OPT (Eq. (10)). Under the reasonable assumption that $q \ll d$, Prior-OPT outperforms Sign-OPT if $\sum_{i=1}^{s} \alpha_i^2 > \frac{2s}{\pi d}$ (where $\frac{2s}{\pi d}$ is an approximate value), which is easily satisfied for large $d$. See Appendix D for details.

Algorithm 1 summarizes our attack procedure. The initialization of $\theta_0$ has two options in *untargeted attacks*: (1) $\theta_0^{\text{RND}}$: we select the best direction with the smallest distortion from 100 random directions

---

**Algorithm 1** Prior-Sign-OPT and Prior-OPT attack

---

**Input:** benign image $\mathbf{x}$, objective function $g(\cdot)$, attack success indicator $\Phi(\cdot)$ defined in Eq. (2), iteration $T$, method $m \in \{$ `Prior-OPT, Prior-Sign-OPT` $\}$, the initialization strategy of untargeted attacks `init` $\in \{\theta_0^{\text{PGD}}, \theta_0^{\text{RND}}\}$, the maximum gradient norm $\mathbf{g}_{\max}$, attack norm $p \in \{2, \infty\}$, surrogate models $\mathbb{S} = \{\hat{f}_1, \ldots, \hat{f}_s\}$.
**Output:** adversarial example $\mathbf{x}^*$ that satisfies $\Phi(\mathbf{x}^*) = 1$.
$\tilde{\mathbf{x}}_0 \leftarrow \text{PGD}(\mathbf{x}, \hat{f}_1)$ if `init` $= \theta_0^{\text{PGD}}$, otherwise a random $\tilde{\mathbf{x}}_0$ that satisfies $\Phi(\tilde{\mathbf{x}}_0) = 1$ is selected for the $\theta_0^{\text{RND}}$ strategy; ▷ the targeted attack selects an image from the target class as $\tilde{\mathbf{x}}_0$.
$\theta_0 \leftarrow \frac{\tilde{\mathbf{x}}_0 - \mathbf{x}}{\|\tilde{\mathbf{x}}_0 - \mathbf{x}\|_2}, \quad d_0 \leftarrow \|\tilde{\mathbf{x}}_0 - \mathbf{x}\|_p$;
**for** $t$ **in** $1, \ldots, T$ **do**
    **for** $\hat{f}_i$ **in** $\mathbb{S}$ **do**
        $\lambda_{t-1} \leftarrow \text{BinarySearch}(\mathbf{x}, \theta_{t-1}, \hat{f}_i, \Phi)$;
        $\mathbf{k}_i \leftarrow \nabla_\theta h(\theta_{t-1}, \lambda_{t-1})$ on $\hat{f}_i$ with $\lambda_{t-1}$ treated as a constant in differentiation; ▷ obtain $s$ transfer-based priors.
    **end for**
    $\mathbf{r}_i \sim \mathcal{N}(\mathbf{0}, \mathbf{I})$ for $i = 1, \ldots, q - s$;
    $\mathbf{p}_1, \ldots, \mathbf{p}_s, \mathbf{u}_1, \ldots, \mathbf{u}_{q-s} \leftarrow$ Gram-Schmidt orthonormalization($\{\mathbf{k}_1, \ldots, \mathbf{k}_s, \mathbf{r}_1, \ldots, \mathbf{r}_{q-s}\}$);
    Estimate a gradient $\mathbf{v}^*$ using Eq. (7) if $m = $ `Prior-Sign-OPT`, otherwise using Eq. (13);
    $\mathbf{v}^* \leftarrow \text{ClipGradNorm}(\mathbf{v}^*, \mathbf{g}_{\max})$;
    $\eta^* \leftarrow \text{LineSearch}(\mathbf{x}, \mathbf{v}^*, \Phi, d_{t-1}, \theta_{t-1})$; ▷ search step size.
    $\theta_t \leftarrow \theta_{t-1} - \eta^* \mathbf{v}^*, \quad \theta_t \leftarrow \frac{\theta_t}{\|\theta_t\|_2}$;
    $d_t \leftarrow \|g(\theta_t) \cdot \theta_t\|_p$;
**end for**
**return** $\mathbf{x}^* \leftarrow \mathbf{x} + g(\theta_T)\frac{\theta_T}{\|\theta_T\|_2}$;

---

as $\theta_0$; (2) $\theta_0^{\text{PGD}}$: we apply PGD (Madry et al., 2018) to attack a surrogate model $\hat{f}_1$ to initialize $\theta_0$, which uses the transfer-based attack as initialization. In *targeted attacks*, we initialize $\theta_0$ with an image $\tilde{\mathbf{x}}_0$ selected from the target class in the training set. In each iteration, the algorithm first calculates the gradient of Eq. (4) on each surrogate model $\hat{f}_i$ in $\mathbb{S}$ to obtain the priors $\mathbf{k}_1, \ldots, \mathbf{k}_s$. Then, we combine these priors and the randomly sampled vectors $\mathbf{r}_1, \ldots, \mathbf{r}_{q-s}$ into a list $\mathbb{L}$, where the priors are positioned ahead of the random vectors. After performing Gram-Schmidt orthonormalization on $\mathbb{L}$, the orthonormal vectors $\mathbf{p}_1, \ldots, \mathbf{p}_s, \mathbf{u}_1, \ldots, \mathbf{u}_{q-s}$ are obtained, representing an orthonormal basis of the subspace. With these orthonormal vectors, we estimate the gradient $\mathbf{v}^*$ using Eq. (7) for Prior-Sign-OPT or Eq. (13) for Prior-OPT, respectively. Then, we employ the gradient clipping technique to address the large-norm gradient problem caused by finite difference. Finally, we use line search to find the optimal step size $\eta^*$ and perform a gradient descent step to minimize $g(\theta)$.

## 4 EXPERIMENTS

### 4.1 EXPERIMENTAL SETTING

**Datasets.** All experiments are conducted on two datasets, i.e., CIFAR-10 (Krizhevsky & Hinton, 2009) and ImageNet (Deng et al., 2009). The image sizes are $32 \times 32 \times 3$ for CIFAR-10, and either $299 \times 299 \times 3$ or $224 \times 224 \times 3$ for ImageNet. We randomly select 1,000 images from the validation sets for experiments. In targeted attacks, for the same target class, we use the same image $\tilde{\mathbf{x}}_0$ as the initialization for all methods. We set the target label as $y_{adv} = (y + 1) \mod C$, where $y$ is the true label and $C$ is the number of classes. Results of the CIFAR-10 dataset are presented in Appendix G.5.

**Method Setting.** The hyperparameter settings of all methods are listed in Appendix F. In the experiments, surrogate models are denoted as subscripts in the method names. For instance, Prior-OPT$_{\text{ResNet50\&ConViT}}$ means using ResNet-50 and ConViT as the surrogate models for Prior-OPT, and Prior-OPT$_{\theta_0^{\text{PGD}} + \text{ResNet50}}$ applies PGD attack on the surrogate model ResNet-50 to initialize $\theta_0$.

**Compared Methods.** To provide a comprehensive comparison, we select 11 state-of-the-art hard-label attacks, including Sign-OPT, SVM-OPT (Cheng et al., 2020), HSJA (Chen et al., 2020),

Triangle Attack (Wang et al., 2022), TA, G-TA (Ma et al., 2021b), SurFree (Maho et al., 2021), GeoDA (Rahmati et al., 2020), Evolutionary (Dong et al., 2019), BBA (Brunner et al., 2019), and SQBA (Park et al., 2024). SQBA, Triangle Attack, GeoDA, and our $\theta_0^{\text{PGD}}$ initialization strategy (e.g., Prior-OPT$_{\theta_0^{\text{PGD}} + \text{ResNet50}}$) only support untargeted attacks. Both BBA and SQBA use a single surrogate model, denoted as a subscript in the method name (e.g., SQBA$_{\text{ResNet50}}$).

**Target Models and Surrogate Models.** In the ImageNet dataset, we select 8 neural network architectures as the target models, including Convolutional Neural Networks (CNNs) and Vision Transformers (ViTs). The selected target models are Inception-v3 (Szegedy et al., 2016), Inception-v4 (Szegedy et al., 2017), ResNet-101 (He et al., 2016), SENet-154 (Hu et al., 2018), ResNeXt-101 $(64 \times 4d)$ (Xie et al., 2017), Vision Transformer (ViT) (Dosovitskiy et al., 2021), Swin Transformer (Liu et al., 2021), and Global Context Vision Transformer (GC ViT) (Hatamizadeh et al., 2023). The Inception-v3 and Inception-v4 require a resolution of $299 \times 299$ for the input images, and we select Inception-ResNet-v2 (IncResV2) and Xception as the surrogate models. ResNet-50 (He et al., 2016) and ConViT (D'Ascoli et al., 2021) are selected as the surrogate models for the remaining target models. In the attacks on defense models, we use the adversarially trained (AT) surrogate models (e.g., AT(ResNet110)), which are marked as subscripts in the method names, such as Prior-OPT$_{\text{AT(ResNet110)}}$.

**Evaluation Metric.** All methods are evaluated using the mean distortion over 1,000 images as $\frac{1}{|\mathbf{X}|} \sum_{\mathbf{x} \in \mathbf{X}} (\|\mathbf{x}_{\text{adv}} - \mathbf{x}\|_p)$ under different query budgets, where $\mathbf{X}$ is the test set and $p \in \{2, \infty\}$ is the attack norm. We also report the attack success rate (ASR) under the specific query budget, which is defined as the percentage of samples with distortions below a threshold $\epsilon$. In $\ell_2$-norm attacks, we set the threshold $\epsilon = \sqrt{0.001 \times d}$ on the ImageNet dataset, where $d$ is the image dimension. Following Li et al. (2021), we calculate the area under the curve (AUC) of $\ell_2$ distortions versus queries.

## 4.2 COMPARISON WITH STATE-OF-THE-ART ATTACKS

Table 1: Mean $\ell_2$ distortions of different query budgets on the ImageNet dataset.

| Target Model | Method | Untargeted Attack | | | | | Targeted Attack | | | | | | |
|---|---|---|---|---|---|---|---|---|---|---|---|---|---|
| | | @1K | @2K | @5K | @8K | @10K | @1K | @2K | @5K | @8K | @10K | @15K | @20K |
| Inception-v4 | HSJA (Chen et al., 2020) | 75.392 | 44.530 | 20.567 | 14.194 | 11.645 | 95.876 | 79.001 | 52.176 | 39.190 | 32.951 | 24.546 | 19.522 |
| | TA (Ma et al., 2021b) | 67.496 | 42.233 | 20.352 | 14.175 | 11.694 | 78.883 | 61.990 | 40.669 | 31.506 | 27.111 | 21.079 | 17.319 |
| | G-TA (Ma et al., 2021b) | 67.842 | 41.946 | 19.962 | 13.865 | 11.448 | 79.297 | 62.291 | 40.529 | 30.941 | 26.427 | 20.268 | 16.569 |
| | Sign-OPT (Cheng et al., 2020) | 86.716 | 48.233 | 18.258 | 11.067 | 8.786 | 80.366 | 65.200 | 42.866 | 32.104 | 27.526 | 20.394 | 16.281 |
| | SVM-OPT (Cheng et al., 2020) | 89.863 | 47.914 | 18.297 | 11.091 | 8.839 | 79.807 | 65.590 | 43.426 | 33.090 | 28.797 | 22.354 | 18.795 |
| | GeoDA (Rahmati et al., 2020) | 29.157 | 20.119 | 12.487 | 11.010 | 9.688 | | | | | | | |
| | Evolutionary (Dong et al., 2019) | 61.966 | 42.665 | 20.815 | 13.382 | 10.839 | 81.761 | 65.060 | 43.021 | 32.120 | 27.385 | 19.942 | 15.610 |
| | SurFree (Maho et al., 2021) | 51.685 | 38.482 | 22.845 | 16.374 | 13.818 | 84.925 | 74.887 | 55.991 | 44.475 | 39.004 | 29.354 | 23.153 |
| | Triangle Attack (Wang et al., 2022) | 27.217 | 25.853 | 23.743 | 22.581 | 22.132 | - | - | - | - | - | - | - |
| | SQBA$_{\text{IncResV2}}$ (Park et al., 2024) | 26.134 | 19.035 | 11.189 | 8.432 | 7.417 | - | - | - | - | - | - | - |
| | SQBA$_{\text{Xception}}$ (Park et al., 2024) | 23.672 | 17.424 | 10.502 | 8.036 | 7.115 | - | - | - | - | - | - | - |
| | BBA$_{\text{IncResV2}}$ (Brunner et al., 2019) | 38.782 | 28.437 | 18.757 | 15.474 | 14.191 | 66.746 | 56.283 | 41.324 | 34.066 | 30.942 | 25.757 | 22.630 |
| | BBA$_{\text{Xception}}$ (Brunner et al., 2019) | 43.317 | 31.519 | 20.504 | 16.712 | 15.282 | 63.069 | 53.363 | 39.740 | 33.166 | 30.221 | 25.438 | 22.561 |
| | Prior-Sign-OPT$_{\text{IncResV2}}$ | 81.991 | 42.403 | 12.835 | 7.365 | 5.842 | 74.597 | 55.421 | 31.856 | 22.958 | 19.513 | 14.361 | 11.665 |
| | Prior-Sign-OPT$_{\text{IncResV2\&Xception}}$ | 77.683 | 37.099 | 9.058 | 5.195 | 4.199 | 69.526 | 49.368 | **26.882** | **19.324** | **16.697** | **12.821** | **10.769** |
| | Prior-Sign-OPT$_{\theta_0^{\text{PGD}} + \text{IncResV2}}$ | 23.596 | 15.347 | 8.074 | 5.729 | 4.863 | - | - | - | - | - | - | - |
| | Prior-OPT$_{\text{IncResV2}}$ | 49.279 | 18.135 | 5.718 | 4.451 | 4.027 | 67.300 | 49.842 | 33.477 | 27.602 | 25.281 | 21.837 | 19.800 |
| | Prior-OPT$_{\text{IncResV2\&Xception}}$ | 42.541 | 13.418 | **3.919** | **3.321** | **3.119** | 60.211 | 42.631 | 27.547 | 23.011 | 21.441 | 19.193 | 17.983 |
| | Prior-OPT$_{\theta_0^{\text{PGD}} + \text{IncResV2}}$ | **22.852** | **12.194** | 6.568 | 5.114 | 4.548 | - | - | - | - | - | - | - |
| ViT | HSJA (Chen et al., 2020) | 37.813 | 19.386 | 9.031 | 6.604 | 5.637 | 61.491 | 44.853 | 23.947 | 16.926 | 14.152 | 10.791 | 8.922 |
| | TA (Ma et al., 2021b) | 37.923 | 19.867 | 9.078 | 6.636 | 5.674 | 52.110 | 36.455 | **20.536** | 15.145 | 12.885 | 10.158 | 8.609 |
| | G-TA (Ma et al., 2021b) | 37.425 | 19.347 | 8.948 | 6.496 | 5.643 | 52.550 | 36.720 | 20.857 | 15.436 | 13.255 | 10.490 | 8.933 |
| | Sign-OPT (Cheng et al., 2020) | 51.120 | 25.290 | 8.559 | 5.482 | 4.572 | 55.941 | 41.867 | 23.784 | 16.541 | 13.873 | 10.129 | 8.267 |
| | SVM-OPT (Cheng et al., 2020) | 55.802 | 26.580 | 9.242 | 5.938 | 5.070 | 56.002 | 41.899 | 23.909 | 17.273 | 14.848 | 11.739 | 10.320 |
| | GeoDA (Rahmati et al., 2020) | 18.880 | 12.904 | 8.039 | 7.153 | 6.313 | | | | | | | |
| | Evolutionary (Dong et al., 2019) | 40.382 | 25.709 | 11.925 | 7.974 | 6.719 | 57.141 | 40.187 | 21.782 | 15.191 | 12.795 | 9.677 | 8.311 |
| | SurFree (Maho et al., 2021) | 28.228 | 19.016 | 10.194 | 7.321 | 6.303 | 70.337 | 53.129 | 30.054 | 20.595 | 16.908 | 11.794 | 9.204 |
| | Triangle Attack (Wang et al., 2022) | **12.789** | 12.144 | 11.064 | 10.411 | 10.097 | - | - | - | - | - | - | - |
| | SQBA$_{\text{ResNet50}}$ (Park et al., 2024) | 21.741 | 14.004 | 7.738 | 5.861 | 5.201 | - | - | - | - | - | - | - |
| | SQBA$_{\text{ConViT}}$ (Park et al., 2024) | 12.886 | 9.762 | 6.240 | 4.947 | 4.452 | - | - | - | - | - | - | - |
| | BBA$_{\text{ResNet50}}$ (Brunner et al., 2019) | 29.755 | 20.053 | 12.580 | 10.375 | 9.567 | **43.231** | **33.365** | 21.889 | 17.635 | 16.046 | 13.726 | 12.463 |
| | BBA$_{\text{ConViT}}$ (Brunner et al., 2019) | 22.716 | 16.153 | 10.893 | 9.193 | 8.595 | 45.588 | 35.227 | 22.865 | 18.325 | 16.614 | 14.028 | 12.623 |
| | Prior-Sign-OPT$_{\text{ResNet50}}$ | 50.161 | 27.953 | 9.474 | 5.872 | 4.850 | 55.095 | 40.480 | 22.354 | 15.626 | 13.201 | 9.789 | 8.048 |
| | Prior-Sign-OPT$_{\text{ResNet50\&ConViT}}$ | 46.196 | 23.869 | 7.327 | 4.694 | 3.967 | 53.925 | 38.418 | 20.673 | **14.422** | **12.153** | **9.090** | **7.544** |
| | Prior-Sign-OPT$_{\theta_0^{\text{PGD}} + \text{ResNet50}}$ | 29.912 | 18.425 | 7.848 | 5.175 | 4.331 | - | - | - | - | - | - | - |
| | Prior-OPT$_{\text{ResNet50}}$ | 42.838 | 22.704 | 8.848 | 6.024 | 5.195 | 54.348 | 40.930 | 24.408 | 18.117 | 15.803 | 12.638 | 11.070 |
| | Prior-OPT$_{\text{ResNet50\&ConViT}}$ | 26.495 | **11.287** | **4.929** | **3.937** | **3.609** | 53.369 | 40.002 | 24.706 | 19.148 | 17.116 | 14.114 | 12.650 |
| | Prior-OPT$_{\theta_0^{\text{PGD}} + \text{ResNet50}}$ | 29.099 | 17.754 | 8.208 | 5.782 | 5.009 | - | - | - | - | - | - | - |

Table 2: Mean $\ell_2$ distortions of the different numbers of priors on the ImageNet dataset.

| Method | Priors | Target Model: ResNet-101[1] | | | | | Target Model: Swin Transformer[2] | | | | | Target Model: GC ViT[2] | | | | |
|---|---|---|---|---|---|---|---|---|---|---|---|---|---|---|---|---|
| | | @1K | @2K | @5K | @8K | @10K | @1K | @2K | @5K | @8K | @10K | @1K | @2K | @5K | @8K | @10K |
| Sign-OPT | no prior | 37.248 | 21.235 | 8.982 | 5.811 | 4.754 | 86.373 | 53.399 | 20.686 | 12.406 | 9.899 | 57.903 | 35.762 | 14.763 | 9.047 | 7.185 |
| Prior-Sign-OPT | 1 prior | 34.150 | 18.733 | 6.111 | 3.718 | 3.019 | 84.124 | 52.882 | 20.344 | 11.880 | 9.254 | 57.171 | 36.949 | 14.963 | 8.931 | 6.899 |
| | 2 priors | 32.848 | 17.548 | 5.121 | 3.136 | 2.593 | 77.459 | 43.062 | 13.614 | 7.903 | 6.331 | 54.896 | 32.418 | 11.012 | 6.651 | 5.342 |
| | 3 priors | 31.156 | 15.455 | 4.074 | 2.527 | 2.122 | 73.110 | 37.852 | 10.264 | 5.939 | 4.778 | 52.744 | 28.939 | 8.707 | 5.245 | 4.215 |
| | 4 priors | 29.984 | 14.707 | 3.698 | 2.333 | 1.989 | 70.246 | 34.470 | 8.526 | 5.066 | 4.169 | 50.256 | 26.027 | 6.435 | 3.804 | 3.212 |
| | 5 priors | **29.601** | **14.195** | **3.573** | **2.275** | **1.951** | **67.616** | **32.225** | **7.321** | **4.219** | **3.467** | **48.935** | **24.821** | **6.123** | **3.601** | **2.893** |
| Prior-OPT | 1 prior | 18.355 | 7.100 | 2.840 | 2.324 | 2.158 | 69.432 | 39.447 | 16.536 | 11.241 | 9.625 | 50.467 | 29.091 | 11.537 | 7.311 | 5.948 |
| | 2 priors | 17.373 | 6.465 | 2.454 | 2.096 | 1.979 | 41.152 | 17.977 | 7.289 | 5.453 | 4.896 | 36.055 | 16.176 | 6.094 | 4.413 | 3.747 |
| | 3 priors | 15.373 | 5.350 | 1.919 | 1.714 | 1.653 | **36.636** | 13.877 | 5.166 | 4.008 | 3.687 | **33.181** | 13.005 | 4.702 | 3.644 | 3.264 |
| | 4 priors | **15.422** | **5.220** | **1.849** | **1.654** | **1.596** | 38.343 | 12.650 | 3.784 | 3.027 | 2.850 | 34.396 | 10.994 | 3.047 | 2.356 | 2.171 |
| | 5 priors | 15.556 | 5.395 | 1.881 | 1.672 | 1.605 | 37.712 | **12.070** | **3.488** | **2.747** | **2.577** | 33.351 | **10.369** | **2.921** | **2.329** | **2.159** |

[1] Five surrogate models: ResNet-50, SENet-154, ResNeXt-101 (64 × 4d), VGG-13, SqueezeNet v1.1
[2] Five surrogate models: ResNet-50, ConViT, CrossViT, MaxViT, ViT

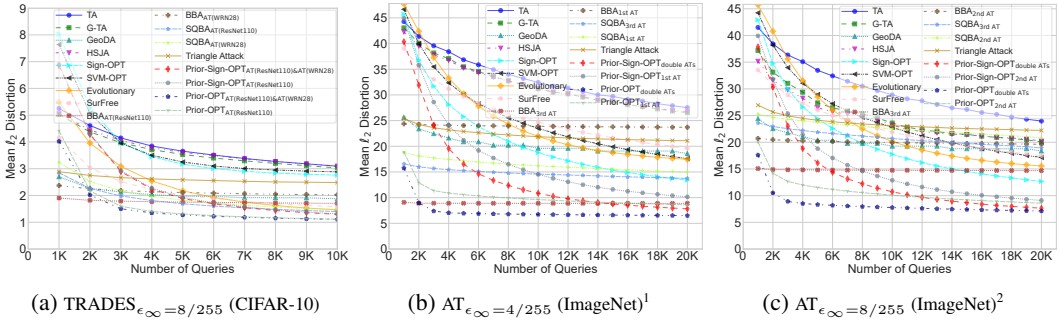

(a) TRADES$_{\epsilon_\infty = 8/255}$ (CIFAR-10)  (b) AT$_{\epsilon_\infty = 4/255}$ (ImageNet)[1]  (c) AT$_{\epsilon_\infty = 8/255}$ (ImageNet)[2]

Figure 3: Mean distortions of untargeted attacks on the defense models equipped with the ResNet-50.

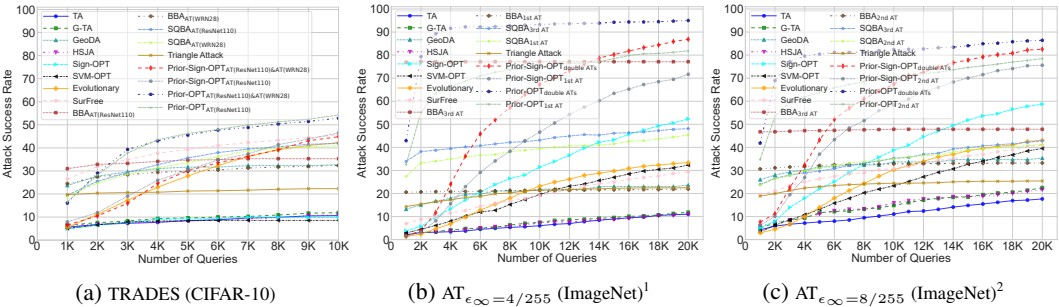

(a) TRADES (CIFAR-10)  (b) AT$_{\epsilon_\infty = 4/255}$ (ImageNet)[1]  (c) AT$_{\epsilon_\infty = 8/255}$ (ImageNet)[2]

Figure 4: Attack success rates of untargeted attacks with $\ell_2$-norm constraint against defense models.

**Results of Attacks against Undefended Models.** Table 1 shows the results of attacks against undefended models on the ImageNet dataset. Additional results are in Appendix G.5. In summary:

(1) In untargeted attacks (Table 1 and Fig. 7), the performance of Prior-OPT significantly surpasses that of all methods, and using multiple surrogate models yields better performance than using a single surrogate model. In addition, the PGD initialization ($\theta_0^{\text{PGD}}$) proves effective in early iterations, because it establishes a high-quality initial attack direction $\theta_0$ through transfer-based attacks.

(2) In targeted attacks, Prior-OPT outperforms Prior-Sign-OPT when the query budget is below 5,000, while Prior-Sign-OPT performs better in later iterations with more queries.

(3) Table 2 and Fig. 5c demonstrate that using more surrogate models (priors) can boost performance.

---

[1] 1st AT: AT(ResNet-50, $\epsilon_{\ell_\infty} = 8/255$), 3rd AT: AT(ResNet-50, $\epsilon_{\ell_2} = 3$), double ATs: combination of both
[2] 2nd AT: AT(ResNet-50, $\epsilon_{\ell_\infty} = 4/255$), 3rd AT: AT(ResNet-50, $\epsilon_{\ell_2} = 3$), double ATs: combination of both

**Results of Attacks against Defense Models.** We conduct untargeted attack experiments against two types of defense models, i.e., adversarial training (AT) (Madry et al., 2018) and TRADES (Zhang et al., 2019). Figs. 3 and 4 show that Prior-OPT with two surrogate models (Prior-OPT$_{\text{double ATs}}$) achieves the best performance on the ImageNet dataset and the CIFAR-10 dataset.

## 4.3 COMPREHENSIVE UNDERSTANDING OF PRIOR-OPT

In the ablation studies, we conduct control experiments based on theoretical analysis results and attacks on real images (Fig. 5). In Figs. 5a, 5b, and 5c, we set the image dimension to $d = 3,072$ and use $\mathbb{E}[\gamma]$ (Eq. (9) for Sign-OPT, Eq. (11) for Prior-Sign-OPT, Eq. (15) and Eq. (16) for the lower and upper bound of Prior-OPT) as the metric for gradient estimation accuracy, where $\gamma = \overline{\mathbf{v}^*}^\top \overline{\nabla g(\theta)}$. Figs. 5a and 5c are based on $q = 50$. Fig. 5a uses one prior and shows that Prior-OPT and Prior-Sign-OPT outperform Sign-OPT with different values of $\alpha$. Fig. 5a also shows that Prior-Sign-OPT performs well when $\alpha$ is small and $\mathbb{E}[\gamma]$ decreases when $\alpha$ is close to 1. This is because when we set $\alpha = 1$, $\mathbb{E}[\gamma] = 1/\sqrt{q}$ in Eq. (11). Fig. 5b shows that $\mathbb{E}[\gamma]$ monotonically increases with $q$ for each method, and Prior-Sign-OPT performs worse than Sign-OPT when $q > 500$. Fig. 5c validates that the performance can be improved when more priors are available, and prioritizing surrogate models with larger $\alpha$ values outperforms random selection. Fig. 5c is consistent with the conclusion of the experimental results in Table 2. Fig. 5d shows the untargeted attack results of Prior-OPT against Swin Transformer with varying $q$ on ImageNet. A smaller $q$ achieves better performance in the early iterations, but becomes less effective in the late stage of iterations with a higher number of queries.

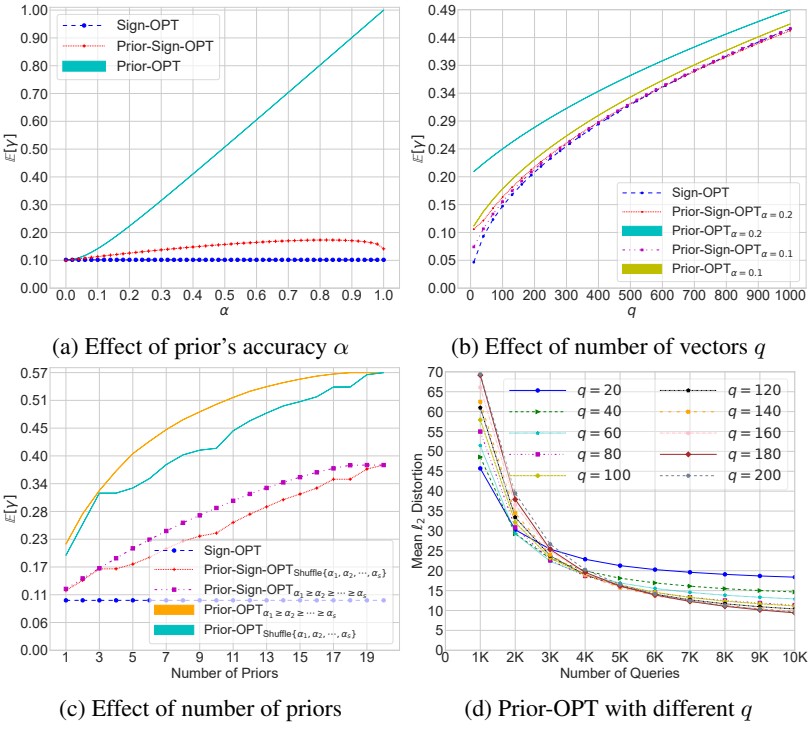

(a) Effect of prior's accuracy $\alpha$

(b) Effect of number of vectors $q$

(c) Effect of number of priors

(d) Prior-OPT with different $q$

Figure 5: Results of ablation studies. Figs. 5a, 5b, and 5c are based on Eqs. (9), (11), (15), and (16).

## 5 CONCLUSION

In this paper, we propose novel hard-label attacks (i.e., Prior-OPT and Prior-Sign-OPT) that incorporate transfer-based priors into the gradient estimation of the ray direction and significantly boost the attack performance. Through theoretical analysis, we prove the effectiveness of our approach: we derive expressions for the expected cosine similarities between the estimated and true gradients, enabling theoretical comparison against the baseline. Therefore, our analysis offers a comprehensive understanding of Prior-OPT and Prior-Sign-OPT. Lastly, we evaluate our approach through extensive experiments, demonstrating superior performance compared to state-of-the-art methods.

## ACKNOWLEDGMENTS

This work was supported by Key R&D Program of Zhejiang under Grant No. 2024C01164, by Zhejiang Provincial Natural Science Foundation of China under Grant No. LMS25F020005, and by the National Natural Science Foundation of China under Grant No. U21B2001.

## ETHICS STATEMENT

We affirm our commitment to the ICLR Code of Ethics to ensure that our research on adversarial examples and AI security adheres to the highest ethical standards. Our work aims to identify potential vulnerabilities in AI systems with the intention of enhancing their security and resilience. Specifically, our approach can be integrated into the evaluation process for a model's robustness, enabling the study and implementation of targeted defense strategies and providing effective support to strengthen the security specifications of artificial intelligence models. We acknowledge the dual-use nature of this research. We have taken steps to responsibly share our findings, encouraging their use in developing robust defense mechanisms. We remain open to discussions about any ethical concerns that may arise and are dedicated to contributing positively to the field of AI security.

## REPRODUCIBILITY STATEMENT

We have taken several steps to ensure the reproducibility of our research. The appendix provides comprehensive details on the algorithm settings, computational resources, and theoretical proofs. To further support reproducibility, we provide the complete attack code for our approach and all baseline methods at `https://github.com/machanic/hard_label_attacks`. These resources are intended to enable others to replicate our experiments.

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

APPENDIX

## A  PROOF FOR SURROGATE GRADIENT COMPUTATION

For a surrogate model $\hat{f}$, we define $f$ as the negative C&W loss function as follows.

$$f(\mathbf{x}') := \begin{cases} \hat{f}(\mathbf{x}')_y - \max_{j \neq y} \hat{f}(\mathbf{x}')_j, & \text{if untargeted attack,} \\ \max_{j \neq \hat{y}_{\text{adv}}} \hat{f}(\mathbf{x}')_j - \hat{f}(\mathbf{x}')_{\hat{y}_{\text{adv}}}, & \text{if targeted attack,} \end{cases} \tag{18}$$

where $\hat{f}(\mathbf{x}')_i$ denotes the $i$-th element of the output of $\hat{f}(\mathbf{x}')$. Note that $h(\theta, \lambda)$ defined in Eq. (4) of the main text equals $f\left(\mathbf{x} + \lambda \cdot \frac{\theta}{\|\theta\|}\right)$. Now we consider $g(\theta)$, the distance from the benign image $\mathbf{x}$ to the adversarial region along the ray direction $\theta$, as defined in Eq. (3). For any $\mathbf{x}'$, $\Phi(\mathbf{x}') = 1 \Leftrightarrow f(\mathbf{x}') \leq 0$, so we have

$$g(\theta) = \inf\left\{\lambda : \lambda > 0, f\left(\mathbf{x} + \lambda \frac{\theta}{\|\theta\|}\right) \leq 0\right\}, \tag{19}$$

where $\inf(\cdot)$ denotes the infimum of a subset of $\mathbb{R}$. We define $g(\theta) = +\infty$ when no valid $\lambda$ exists, since $\inf \emptyset = +\infty$ by convention. Now we can prove the following proposition.

**Proposition A.1.** *If $f$ is continuous, $f(\mathbf{x}) > 0$, then given any $\theta \neq \mathbf{0}$ s.t. $g(\theta) < +\infty$, we have $g(\theta) > 0$ and $f\left(\mathbf{x} + g(\theta)\frac{\theta}{\|\theta\|}\right) = 0$.*

*Proof.* We first prove that $g(\theta) > 0$. We begin by defining the function $f_\theta(\lambda) := f\left(\mathbf{x} + \lambda \cdot \frac{\theta}{\|\theta\|}\right)$. Note that $f_\theta$ is continuous (w.r.t. $\lambda$) because $f$ is continuous. Since $f_\theta(0) = f(\mathbf{x}) > 0$, there exists $\delta > 0$ such that $\forall 0 < \lambda < \delta$, $f_\theta(\lambda) > 0$. Since $g(\theta) < +\infty$, by the definition we have $g(\theta) = \inf(\{\lambda : \lambda > 0, f_\theta(\lambda) \leq 0\}) \geq \delta > 0$.

Next, we want to prove $f\left(\mathbf{x} + g(\theta)\frac{\theta}{\|\theta\|}\right) = 0$, i.e., $f_\theta(g(\theta)) = 0$. To simplify notation in the proof, let us denote $A_\theta := \{\lambda : \lambda > 0, f_\theta(\lambda) \leq 0\}$.

If $f_\theta(g(\theta)) < 0$, then since $f_\theta$ is continuous and $g(\theta) > 0$, there exists $\epsilon > 0$ such that $f_\theta(g(\theta) - \epsilon) < 0$ and $g(\theta) - \epsilon > 0$. Therefore, we have $g(\theta) - \epsilon \in A_\theta$, which implies that $g(\theta) > \inf(A_\theta)$. This contradicts the definition $g(\theta) = \inf(A_\theta)$.

If $f_\theta(g(\theta)) > 0$, then there exists $\epsilon > 0$ such that $f_\theta(\lambda) > 0$ holds for all $g(\theta) \leq \lambda \leq g(\theta) + \epsilon$. This means that $[g(\theta), g(\theta) + \epsilon] \cap A_\theta = \emptyset$. Noting that $g(\theta)$ is a lower bound of $A_\theta$, this implies that $g(\theta) + \epsilon$ is also a lower bound of $A_\theta$, which contradicts the definition $g(\theta) = \inf(A_\theta)$.

Therefore $f_\theta(g(\theta)) = 0$. $\qquad\square$

Next, we show how to calculate $\nabla g(\theta)$ based on some weak assumptions.

**Theorem A.2.** *Suppose $f$ is continuously differentiable[2] and $f(\mathbf{x}) > 0$. Let $h(\theta, \lambda) = f\left(\mathbf{x} + \lambda \frac{\theta}{\|\theta\|}\right)$. For any $\theta_0 \neq \mathbf{0}$ s.t. $g(\theta_0) < +\infty$, let $\lambda_0 = g(\theta_0)$, and assume that $\frac{\partial h}{\partial \lambda}(\theta_0, \lambda_0) \neq 0$, then we conclude that $g$ is differentiable at $\theta_0$, and*

$$\nabla g(\theta_0) = -\frac{1}{\frac{\partial h}{\partial \lambda}(\theta_0, \lambda_0)} \nabla_\theta h(\theta_0, \lambda_0). \tag{20}$$

*Remark* A.3. The assumptions in the theorem are rather weak. $f(\mathbf{x}) > 0$ (the unperturbed sample can be successfully classified) is a standard assumption; $g(\theta_0) < +\infty$ is a common assumption, necessary for ray search procedure to work; $f$ is continuously differentiable almost everywhere under common network architectures. The only special condition required here is that $\frac{\partial h}{\partial \lambda}(\theta_0, \lambda_0) \neq 0$, which is generally satisfied unless a specific function $f$ is explicitly constructed to violate it. Intuitively, as $\lambda$ increases, the function value decreases from a positive value to a non-positive value, and the derivative w.r.t. $\lambda$ is typically non-zero when the function value crosses zero.

---

[2]A function $f$ is said to be continuously differentiable if all partial derivatives of $f$ exist and are continuous.

*Proof.* Since $(\theta, \lambda) \mapsto \mathbf{x} + \lambda \frac{\theta}{\|\theta\|}$ is continuously differentiable at $\{(\theta, \lambda) : \theta \in \mathbb{R}^d, \lambda \in \mathbb{R}, \theta \neq \mathbf{0}\}$ and $f$ is continuously differentiable everywhere, $h$ is continuously differentiable when $\theta \neq \mathbf{0}$ by the chain rule. By Proposition A.1, $h(\theta_0, \lambda_0) = 0$. Since $\frac{\partial h}{\partial \lambda}(\theta_0, \lambda_0) \neq 0$, by the Implicit Function Theorem (see Theorem 1 in Rio Branco de Oliveira (2012)), there exists a neighborhood $\Theta \subseteq \mathbb{R}^d$ of $\theta_0$ and an open interval $\Lambda := (\lambda_0 - \eta, \lambda_0 + \eta)$ such that for each $\theta \in \Theta$, there is a unique $\lambda \in \Lambda$ satisfying $h(\theta, \lambda) = 0$. Since $\theta$ uniquely determines $\lambda$, we define $\tilde{g} : \Theta \to \Lambda$ satisfying $h(\theta, \tilde{g}(\theta)) = 0$ for all $\theta \in \Theta$. Moreover, the Implicit Function Theorem tells us that $\tilde{g}$ is continuously differentiable, and

$$\nabla \tilde{g}(\theta_0) = -\frac{1}{\frac{\partial h}{\partial \lambda}(\theta_0, \lambda_0)} \nabla_\theta h(\theta_0, \lambda_0). \tag{21}$$

Now it suffices to prove $g$ is differentiable at $\theta_0$ and $\nabla g(\theta_0) = \nabla \tilde{g}(\theta_0)$. We shall prove that there exists a neighborhood of $\theta_0$ in which $g$ and $\tilde{g}$ are equal. Since $h(\theta, \tilde{g}(\theta)) = 0$, from the definition of $g$, we have $g(\theta) \leq \tilde{g}(\theta) < +\infty$ for all $\theta \in \Theta$. By Proposition A.1, $h(\theta, g(\theta)) = 0$, so the uniqueness in Implicit Function Theorem tells us that $\forall \theta \in \Theta$, if $\lambda_0 - \eta < g(\theta) < \lambda_0 + \eta$, then $g(\theta) = \tilde{g}(\theta)$. Since $g(\theta) \leq \tilde{g}(\theta) < \lambda_0 + \eta$, it suffices to prove that $g(\theta) > \lambda_0 - \eta$.

Now we prove that there exists a neighborhood $\Theta'$ of $\theta_0$ such that $\forall \theta \in \Theta'$, $\forall \lambda \in [0, \lambda_0 - \eta]$, $h(\theta, \lambda) > 0$ (this would imply that $\forall \theta \in \Theta'$, $g(\theta) > \lambda_0 - \eta$, since $h(\theta, g(\theta)) = 0$ by Proposition A.1). To prove that, we first note that $\forall \lambda \in [0, \lambda_0 - \eta]$, $h(\theta_0, \lambda) > 0$ since $g(\theta_0) = \lambda_0 > \lambda_0 - \eta$. Since $h(\theta_0, \lambda)$ is continuous w.r.t. $\lambda$, by the Extreme Value Theorem, $h(\theta_0, \lambda)$ on $\lambda \in [0, \lambda_0 - \eta]$ could attain the minimum $h(\theta_0, \lambda^*)$ which is positive, so there exists $\epsilon > 0$ such that $\forall \lambda \in [0, \lambda_0 - \eta]$, $h(\theta_0, \lambda) \geq \epsilon$. We pick a bounded closed neighborhood of $\theta_0$, denoted by $\Theta''$ such that $\mathbf{0} \notin \Theta''$. $h$ is continuous on the compact set $\{(\theta, \lambda) : \theta \in \Theta'', \lambda \in [0, \lambda_0 - \eta]\}$, so by Heine-Cantor Theorem, $h$ is uniformly continuous on the same set. This implies that there exists $\delta > 0$ such that for all $\theta \in \Theta''$ satisfying $\|\theta - \theta_0\| < \delta$, we have $|h(\theta, \lambda) - h(\theta_0, \lambda)| < \epsilon$ and hence $h(\theta, \lambda) > 0$ for all $\lambda \in [0, \lambda_0 - \eta]$. Setting $\Theta' = \Theta'' \cap \{\theta : \|\theta - \theta_0\| < \delta\}$, we have $\forall \theta \in \Theta'$, $\forall \lambda \in [0, \lambda_0 - \eta]$, $h(\theta, \lambda) > 0$, and thus the proposition at the beginning of this paragraph is proven, i.e., $\forall \theta \in \Theta'$, $g(\theta) > \lambda_0 - \eta$.

Therefore, we have proven that there exists a neighborhood of $\theta_0$, $\Theta \cap \Theta'$, in which $g$ and $\tilde{g}$ are equal. Since the differentiability at $\theta_0$ and the gradient only rely on the function value in a neighborhood of $\theta_0$, $g$ is differentiable at $\theta_0$ and $\nabla g(\theta_0) = \nabla \tilde{g}(\theta_0)$. By Eq. (21), the proof is completed. □

# B  ACQUISITION OF TRANSFER-BASED PRIORS IN TARGETED ATTACKS

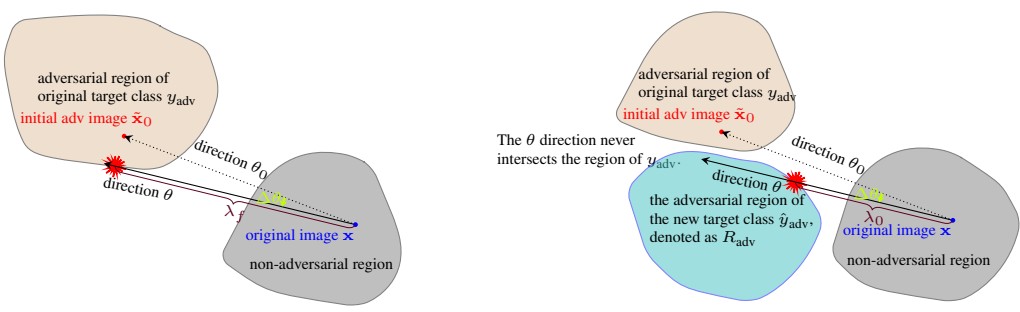

(a) The $\theta$ direction in the target model $f$.  (b) The $\theta$ direction in the surrogate model $\hat{f}$.

Figure 6: Illustration of setting the new target class $\hat{y}_{\text{adv}}$ and $\lambda_0$ before obtaining priors.

In targeted attacks, obtaining transfer-based priors is more challenging. Fig. 6 illustrates that $\theta_0$ is initialized as the direction from the original image $\mathbf{x}$ to an initial adversarial image $\tilde{\mathbf{x}}_0$, which is selected from the target class $y_{\text{adv}}$. The $\theta_0$ direction is used as the initial direction in both the surrogate model and the target model. If both models classify $\tilde{\mathbf{x}}_0$ correctly, the $\theta_0$ direction leads to the region of the target class $y_{\text{adv}}$ in both models. During the optimization process, a small perturbation $\Delta\theta$ is added to $\theta_0$, resulting in a new direction $\theta$. Although the $\theta$ direction may successfully guide the attack towards the adversarial region of the target class $y_{\text{adv}}$ in the target model $f$, it may not lead to the same region in the surrogate model $\hat{f}$. This is a result of the varying decision boundaries between the two models, which differ in both shape and extent. Therefore, $g_{\hat{f}}(\theta)$ becomes infinitely large in

this case, as shown in Fig. 6b. To address this issue, we set a new target class $\hat{y}_{\text{adv}}$ and $\lambda_0$ before computing the transfer-based priors. The procedure is as follows:

(1) Starting at the original image $\mathbf{x}$, we move along the $\theta$ direction in the surrogate model $\hat{f}$. The region located at a distance of $\lambda_f + 1$ along $\theta$ is defined as the new adversarial region $R_{\text{adv}}$ with label $\hat{y}_{\text{adv}}$, where $\lambda_f$ represents the shortest distance from $\mathbf{x}$ along $\theta$ to the adversarial region of the original target class $y_{\text{adv}}$ in the target model $f$. The shortest distance from $\mathbf{x}$ to $R_{\text{adv}}$ along the $\theta$ direction is denoted as $\lambda_0$, as shown in Fig. 6b.

(2) If no adversarial region with the label $\hat{y}_{\text{adv}}$ different from the true label $y$ is found in the previous steps, we then search for the first adversarial region $R_{\text{adv}}$ along the $\theta$ direction in the surrogate model $\hat{f}$, within a distance range of 0 to 200. The label of $R_{\text{adv}}$ is denoted as $\hat{y}_{\text{adv}}$, and the shortest distance from $\mathbf{x}$ to $R_{\text{adv}}$ along the $\theta$ direction is denoted as $\lambda_0$.

## C    THEORETICAL ANALYSIS OF SIGN-OPT, PRIOR-SIGN-OPT, AND PRIOR-OPT

**Lemma C.1.** *Suppose* $\mathbf{u} \sim \mathcal{U}(\mathbb{S}_{d-1})$ *where* $\mathcal{U}(\mathbb{S}_{d-1})$ *denotes the uniform distribution on the unit hypersphere in* $\mathbb{R}^d$. *Suppose* $\mathbf{g}$ *is a fixed vector in* $\mathbb{R}^d$ *with* $\|\mathbf{g}\| = 1$. *Let* $\beta := \mathbf{u}^\top \mathbf{g}$. *Then*

$$\mathbb{E}[|\beta|] = \frac{\Gamma(\frac{d}{2})}{\Gamma(\frac{d+1}{2})\sqrt{\pi}}, \tag{22}$$

$$\mathbb{E}[\beta^2] = \frac{1}{d}, \tag{23}$$

*where* $\Gamma$ *is the gamma function.*

*Proof.* Let $\mathbf{a} \sim \mathcal{N}(\mathbf{0}, \mathbf{I}) \in \mathbb{R}^d$, then we let $\mathbf{u} = \frac{\mathbf{a}}{\|\mathbf{a}\|}$. Hence $\beta = \frac{\mathbf{a}^\top \mathbf{g}}{\|\mathbf{a}\|}$. Note that $\frac{\mathbf{a}}{\|\mathbf{a}\|}$ and $\|\mathbf{a}\|$ are independent because the distribution of $\frac{\mathbf{a}}{\|\mathbf{a}\|}$ is always the uniform distribution on the unit hypersphere given any restriction to the value of $\|\mathbf{a}\|$. Therefore, $\beta = \frac{\mathbf{a}^\top \mathbf{g}}{\|\mathbf{a}\|}$ and $\|\mathbf{a}\|$ are also independent, so $|\beta|$ and $\|\mathbf{a}\|$ are independent. Noting that $|\beta|\|\mathbf{a}\| = |\mathbf{a}^\top \mathbf{g}|$, we have

$$\mathbb{E}[|\mathbf{a}^\top \mathbf{g}|] = \mathbb{E}[|\beta|]\mathbb{E}[\|\mathbf{a}\|]. \tag{24}$$

Since $\mathbf{a}^\top \mathbf{g}$ is a affine transformation of the multivariate Gaussian variable $\mathbf{a}$, $\mathbf{a}^\top \mathbf{g}$ also has a Gaussian distribution with the mean 0 and the variance $\mathbf{g}^\top \mathbf{I} \mathbf{g} = 1$, so $\mathbf{a}^\top \mathbf{g} \sim \mathcal{N}(0, 1)$. Therefore, $|\mathbf{a}^\top \mathbf{g}|$ follows the folded normal distribution (actually its special case: the half-normal distribution), and by the formula in Tsagris et al. (2014),

$$\mathbb{E}[|\mathbf{a}^\top \mathbf{g}|] = \frac{\sqrt{2}}{\sqrt{\pi}}. \tag{25}$$

$\|\mathbf{a}\|$ follows the chi distribution with $d$ degrees of freedom, so by the formula of its mean

$$\mathbb{E}[\|\mathbf{a}\|] = \sqrt{2}\frac{\Gamma(\frac{d+1}{2})}{\Gamma(\frac{d}{2})}. \tag{26}$$

Substituting Eq. (25) and Eq. (26) into Eq. (24), we have proved Eq. (22).

Since $\beta$ and $\|\mathbf{a}\|$ are independent, similarly to Eq. (24), we have

$$\mathbb{E}[(\mathbf{a}^\top \mathbf{g})^2] = \mathbb{E}[\beta^2]\mathbb{E}[\|\mathbf{a}\|^2]. \tag{27}$$

Since $\mathbf{a} \sim \mathcal{N}(\mathbf{0}, \mathbf{I})$, $\mathbb{E}[\mathbf{a}\mathbf{a}^\top] = \mathbf{I}$. Hence $\mathbb{E}[(\mathbf{a}^\top \mathbf{g})^2] = \mathbb{E}[\mathbf{g}^\top \mathbf{a} \cdot \mathbf{a}^\top \mathbf{g}] = \mathbf{g}^\top \mathbb{E}[\mathbf{a}\mathbf{a}^\top]\mathbf{g} = \|\mathbf{g}\|^2 = 1$, and $\mathbb{E}[\|\mathbf{a}\|^2] = \mathbb{E}[\text{Tr}(\mathbf{a}\mathbf{a}^\top)] = \text{Tr}(\mathbb{E}[\mathbf{a}\mathbf{a}^\top]) = d$. By Eq. (27), Eq. (23) has been proved.

$\square$

**Lemma C.2.** *Suppose* $\mathbf{g}$ *is a fixed vector in* $\mathbb{R}^d$ *with* $\|\mathbf{g}\| = 1$. *Suppose* $\mathbf{p}$ *is another fixed vector in* $\mathbb{R}^d$ *with* $\|\mathbf{p}\| = 1$, *and let* $\beta_p := \mathbf{g}^\top \mathbf{p}$. *Let* $\mathbf{u}$ *be a random vector uniformly sampled from the unit*

*hypersphere in the $(d-1)$-dimensional subspace orthogonal to $\mathbf{p}$. Specifically, $\mathbf{u}$ can be constructed as $\mathbf{u} = \overline{\boldsymbol{\xi} - \boldsymbol{\xi}^\top \mathbf{p} \cdot \mathbf{p}}$ where $\boldsymbol{\xi} \sim \mathcal{U}(\mathbb{S}_{d-1})$. Let $\beta_\perp := \mathbf{g}^\top \mathbf{u}$, then*

$$\mathbb{E}[|\beta_\perp|] = \frac{\Gamma(\frac{d-1}{2})}{\Gamma(\frac{d}{2})\sqrt{\pi}}\sqrt{1-\beta_p^2}. \tag{28}$$

*Proof.* To observe the property of $\beta_\perp$, since $\mathbf{u}$ is orthogonal to $\mathbf{p}$, we do the following decomposition for $\mathbf{g}$:

$$\mathbf{g} = \mathbf{g}^\top \mathbf{p} \cdot \mathbf{p} + \mathbf{g}_\perp = \beta_p \mathbf{p} + \mathbf{g}_\perp, \tag{29}$$

where $\mathbf{g}_\perp := \mathbf{g} - \mathbf{g}^\top \mathbf{p} \cdot \mathbf{p}$ denotes the projection of $\mathbf{g}$ to the $(d-1)$-dimensional subspace orthogonal to $\mathbf{p}$. By expanding the inner product, we have

$$\|\mathbf{g}_\perp\|^2 = 1 - 2\beta_p^2 + \beta_p^2 = 1 - \beta_p^2, \tag{30}$$

so $\|\mathbf{g}_\perp\| = \sqrt{1-\beta_p^2}$. Meanwhile,

$$\beta_\perp = \mathbf{g}^\top \mathbf{u} = (\mathbf{g}^\top \mathbf{p} \cdot \mathbf{p} + \mathbf{g}_\perp)^\top \mathbf{u} = \mathbf{g}_\perp^\top \mathbf{u}. \tag{31}$$

Therefore, $\beta_\perp$ is essentially the inner product between a random vector uniformly sampled from the unit hypersphere in a $(d-1)$-dimensional subspace and a fixed vector with norm $\sqrt{1-\beta_p^2}$ in this subspace. Taking the expectation of the absolute values on both sides of Eq. (31), we have

$$\mathbb{E}[|\beta_\perp|] = \mathbb{E}[|\mathbf{g}_\perp^\top \mathbf{u}|] = \|\mathbf{g}_\perp\|\mathbb{E}[|\overline{\mathbf{g}_\perp}^\top \mathbf{u}|]. \tag{32}$$

Since both $\overline{\mathbf{g}_\perp}$ and $\mathbf{u}$ reside in the $(d-1)$-dimensional subspace orthogonal to $\mathbf{p}$, it follows that $\mathbb{E}[|\overline{\mathbf{g}_\perp}^\top \mathbf{u}|]$ corresponds to $\mathbb{E}[|\beta|]$ in Eq. (22), with $d$ replaced by $d-1$. Therefore,

$$\mathbb{E}[|\beta_\perp|] = \|\mathbf{g}_\perp\|\mathbb{E}[|\overline{\mathbf{g}_\perp}^\top \mathbf{u}|] = \frac{\Gamma(\frac{d-1}{2})}{\Gamma(\frac{d}{2})\sqrt{\pi}}\sqrt{1-\beta_p^2}, \tag{33}$$

and the proof is completed. $\square$

**Lemma C.3.** *Let $\beta$ be as defined in Lemma C.1, then the probability density function of $\beta$ is (note that $-1 \leq \beta \leq 1$)*

$$p(\beta) = \frac{\left(\sqrt{1-\beta^2}\right)^{d-3}}{\mathrm{B}\left(\frac{d-1}{2}, \frac{1}{2}\right)}, \tag{34}$$

*where $\mathrm{B}(\cdot, \cdot)$ is the beta function.*

*Proof.* We note that when $-1 \leq x \leq 0$, $P(\beta \leq x)$ is equal to the ratio of the surface area of the hyperspherical cap of a hypersphere in $\mathbb{R}^d$ to the surface area of the hypersphere. For a hyperspherical cap with height $h$ on a unit hypersphere, its surface area is $\frac{1}{2}A_d I_{2h-h^2}\left(\frac{d-1}{2}, \frac{1}{2}\right)$, where $A_d$ is the surface area of the unit hypersphere in $\mathbb{R}^d$ and $I(\cdot, \cdot)$ is the regularized incomplete beta function. To compute $P(\beta \leq x)$ for $-1 \leq x \leq 0$, substituting $h = x + 1$ and dividing the area by $A_d$, we have

$$P(\beta \leq x) = \frac{1}{2}I_{1-x^2}\left(\frac{d-1}{2}, \frac{1}{2}\right), \tag{35}$$

where $I$ is the regularized incomplete beta function, defined as

$$I_x(a, b) = \frac{\int_0^x t^{a-1}(1-t)^{b-1}\mathrm{d}t}{\mathrm{B}(a, b)}. \tag{36}$$

Hence the probability density function is

$$p_\beta(x) = \frac{\partial}{\partial x}P(\beta \leq x) \tag{37}$$

$$= \frac{1}{2} \cdot \frac{-2x}{\mathrm{B}\left(\frac{d-1}{2}, \frac{1}{2}\right)}\left(1-x^2\right)^{\frac{d-1}{2}-1}\left(1-(1-x^2)\right)^{\frac{1}{2}-1} \tag{38}$$

$$= \frac{-x}{|x|} \cdot \frac{\left(\sqrt{1-x^2}\right)^{d-3}}{\mathrm{B}\left(\frac{d-1}{2}, \frac{1}{2}\right)} \tag{39}$$

$$= \frac{\left(\sqrt{1-x^2}\right)^{d-3}}{\mathrm{B}\left(\frac{d-1}{2}, \frac{1}{2}\right)}. \tag{40}$$

Note that the last equality holds because $x \leq 0$.

Therefore, we have proven Eq. (34) for $\beta \leq 0$. When $\beta > 0$, the formula is the same due to the symmetry. The proof is completed. □

## C.1 ANALYSIS FOR SIGN-OPT

We can compute the projection of $\nabla g(\theta)$ onto $S$ with $s = 0$ by summing over all its projection onto the orthonormal basis:

$$\mathbf{v} := \sum_{i=1}^{q} \frac{g(\theta + \sigma \mathbf{u}_i) - g(\theta)}{\sigma} \cdot \mathbf{u}_i, \tag{41}$$

where $\{\mathbf{u}_1, \ldots, \mathbf{u}_q\}$ is a uniformly random orthonormal set of $q$ vectors in $\mathbb{R}^d$, so $\mathbf{u}_i \sim \mathcal{U}(\mathbb{S}_{d-1})$ for any $i \leq q$. However, in hard-label attacks the coefficients above for each basis vector are costly to estimate. In the Sign-OPT estimator, each coefficient is replaced by its sign which is much easier to obtain using hard-label queries:

$$\mathbf{v} = \sum_{i=1}^{q} \text{sign}(g(\theta + \sigma \mathbf{u}_i) - g(\theta)) \cdot \mathbf{u}_i. \tag{42}$$

In the following analysis, we assume that $g$ is differentiable at $\theta$ so that we have $g(\theta + \sigma \mathbf{u}) - g(\theta) = \sigma \cdot \nabla g(\theta)^\top \mathbf{u} + o(\sigma)$ where $\lim_{\sigma \to 0} \frac{o(\sigma)}{\sigma} = 0$ for any unit vector $\mathbf{u}$. We further assume that $\sigma$ is sufficiently small so that we can omit $o(\sigma)$. In practice, if the target model is deterministic, picking a small $\sigma$ is feasible until the numerical error dominates. Therefore, in the following analysis we assume that

$$\text{sign}(g(\theta + \sigma \mathbf{u}) - g(\theta)) \approx \text{sign}(\nabla g(\theta)^\top \mathbf{u}), \tag{43}$$

where $\mathbf{u}$ is a unit vector in $\mathbb{R}^d$. Now we can write Sign-OPT in the following form:

$$\mathbf{v} = \sum_{i=1}^{q} \text{sign}(\nabla g(\theta)^\top \mathbf{u}_i) \cdot \mathbf{u}_i. \tag{44}$$

Now, we present the proof of Theorem 3.2 for the Sign-OPT estimator defined in Eq. (44).

*Proof.* Since $\{\mathbf{u}_i\}_{i=1}^{q}$ are orthonormal, we have $\|\mathbf{v}\| = \sqrt{\sum_{i=1}^{q} (\text{sign}(\nabla g(\theta)^\top \mathbf{u}_i))^2} = \sqrt{q}$. We note that $\text{sign}(\nabla g(\theta)^\top \mathbf{u}_i) = \text{sign}(\overline{\nabla g(\theta)}^\top \mathbf{u}_i)$, so

$$\mathbf{v} = \sum_{i=1}^{q} \text{sign}(\overline{\nabla g(\theta)}^\top \mathbf{u}_i) \cdot \mathbf{u}_i. \tag{45}$$

Hence

$$\gamma = \frac{\mathbf{v}^\top \overline{\nabla g(\theta)}}{\|\mathbf{v}\|} = \frac{1}{\sqrt{q}} \sum_{i=1}^{q} \text{sign}(\overline{\nabla g(\theta)}^\top \mathbf{u}_i) \cdot (\overline{\nabla g(\theta)}^\top \mathbf{u}_i) \tag{46}$$

$$= \frac{1}{\sqrt{q}} \sum_{i=1}^{q} \left| \overline{\nabla g(\theta)}^\top \mathbf{u}_i \right|. \tag{47}$$

Since $\nabla g(\theta)$ is a fixed vector w.r.t. the randomness of $\{\mathbf{u}_i\}_{i=1}^{q}$, and the marginal distribution of $\mathbf{u}_i$ is $\mathcal{U}(\mathbb{S}_{d-1})$ for any $i$, by Eq. (22) we have

$$\mathbb{E}[\gamma] = \frac{1}{\sqrt{q}} q \frac{\Gamma(\frac{d}{2})}{\Gamma(\frac{d+1}{2})\sqrt{\pi}} = \sqrt{q} \frac{\Gamma(\frac{d}{2})}{\Gamma(\frac{d+1}{2})\sqrt{\pi}}. \tag{48}$$

Computing $\mathbb{E}[\gamma^2]$ is more complicated. First we have

$$\gamma^2 = \frac{1}{q} \left( \sum_{i=1}^{q} \left| \overline{\nabla g(\theta)}^\top \mathbf{u}_i \right| \right)^2 \tag{49}$$

$$= \frac{1}{q} \sum_{i=1}^{q} \left( \overline{\nabla g(\theta)}^\top \mathbf{u}_i \right)^2 + \frac{1}{q} \sum_{i \neq j} \left| \overline{\nabla g(\theta)}^\top \mathbf{u}_i \right| \cdot \left| \overline{\nabla g(\theta)}^\top \mathbf{u}_j \right|. \tag{50}$$

For the first part, by Eq. (23) we have

$$\forall i, \mathbb{E}[(\overline{\nabla g(\theta)}^\top \mathbf{u}_i)^2] = \frac{1}{d}. \tag{51}$$

For the second part, let us denote $\beta_i := \overline{\nabla g(\theta)}^\top \mathbf{u}_i$ and $\beta_j := \overline{\nabla g(\theta)}^\top \mathbf{u}_j$. Then we need to compute for $i \neq j$:

$$\mathbb{E}[|\beta_i| \cdot |\beta_j|] = \mathbb{E}_{\mathbf{u}_i}[\mathbb{E}[(|\beta_i| \cdot |\beta_j|)|\mathbf{u}_i]] \tag{52}$$

$$= \mathbb{E}_{\mathbf{u}_i}[|\beta_i|\mathbb{E}[|\beta_j||\mathbf{u}_i]]. \tag{53}$$

Next, we aim to compute $\mathbb{E}[|\beta_j||\mathbf{u}_i]$. Since $\mathbf{u}_i$ and $\mathbf{u}_j$ are orthonormal, conditioned on $\mathbf{u}_i$, the vector $\mathbf{u}_j$ is uniformly distributed on the unit hypersphere in the $(d-1)$-dimensional subspace orthogonal to $\mathbf{u}_i$. When calculating the conditional expectation, we consider $\mathbf{u}_i$ to be fixed and use Lemma C.2. Specifically, in Lemma C.2 we let $\mathbf{g}$ be $\overline{\nabla g(\theta)}$ and let $\mathbf{p}$ be $\mathbf{u}_i$. Then we have

$$\mathbb{E}[|\beta_j||\mathbf{u}_i] = \frac{\Gamma(\frac{d-1}{2})}{\Gamma(\frac{d}{2})\sqrt{\pi}}\sqrt{1 - \beta_i^2}. \tag{54}$$

Substituting Eq. (54) into Eq. (53), we have

$$\mathbb{E}[|\beta_i| \cdot |\beta_j|] = \frac{\Gamma(\frac{d-1}{2})}{\Gamma(\frac{d}{2})\sqrt{\pi}}\mathbb{E}\left[|\beta_i|\sqrt{1 - \beta_i^2}\right]. \tag{55}$$

Here the distribution of $\beta_i$ is the same as that of $\beta$ in Lemma C.1, and we need to compute $\mathbb{E}[|\beta|\sqrt{1 - \beta^2}]$. By Eq. (22) and Eq. (34), we have

$$\frac{\Gamma(\frac{d}{2})}{\Gamma(\frac{d+1}{2})\sqrt{\pi}} = \mathbb{E}[|\beta|] = \int_{-1}^1 p(\beta)|\beta|\mathrm{d}\beta \tag{56}$$

$$= \int_{-1}^1 |\beta|\frac{\left(\sqrt{1 - \beta^2}\right)^{d-3}}{\mathrm{B}\left(\frac{d-1}{2}, \frac{1}{2}\right)}\mathrm{d}\beta, \tag{57}$$

so

$$\int_{-1}^1 |\beta|\left(\sqrt{1 - \beta^2}\right)^{d-3}\mathrm{d}\beta = \frac{\Gamma(\frac{d}{2})}{\Gamma(\frac{d+1}{2})\sqrt{\pi}}\mathrm{B}\left(\frac{d-1}{2}, \frac{1}{2}\right). \tag{58}$$

Hence

$$\mathbb{E}\left[|\beta|\sqrt{1 - \beta^2}\right] = \int_{-1}^1 |\beta|\sqrt{1 - \beta^2}p(\beta)\mathrm{d}\beta \tag{59}$$

$$= \frac{1}{\mathrm{B}\left(\frac{d-1}{2}, \frac{1}{2}\right)}\int_{-1}^1 |\beta|\left(\sqrt{1 - \beta^2}\right)^{d-2}\mathrm{d}\beta \tag{60}$$

$$= \frac{1}{\mathrm{B}\left(\frac{d-1}{2}, \frac{1}{2}\right)}\frac{\Gamma(\frac{d+1}{2})}{\Gamma(\frac{d+2}{2})\sqrt{\pi}}\mathrm{B}\left(\frac{d}{2}, \frac{1}{2}\right), \tag{61}$$

where the last equality is obtained by setting $d$ in Eq. (58) to $d + 1$. Therefore, by Eq. (55) we have

$$\mathbb{E}[|\beta_i| \cdot |\beta_j|] = \frac{1}{\pi}\frac{\mathrm{B}\left(\frac{d}{2}, \frac{1}{2}\right)}{\mathrm{B}\left(\frac{d-1}{2}, \frac{1}{2}\right)}\frac{\Gamma(\frac{d-1}{2})}{\Gamma(\frac{d}{2})}\frac{\Gamma(\frac{d+1}{2})}{\Gamma(\frac{d+2}{2})} \tag{62}$$

$$= \frac{1}{\pi}\frac{\Gamma(\frac{d}{2})\Gamma(\frac{d}{2})}{\Gamma(\frac{d+1}{2})\Gamma(\frac{d-1}{2})}\frac{\Gamma(\frac{d-1}{2})}{\Gamma(\frac{d}{2})}\frac{\Gamma(\frac{d+1}{2})}{\Gamma(\frac{d+2}{2})} \tag{63}$$

$$= \frac{1}{\pi}\frac{\Gamma(\frac{d}{2})}{\Gamma(\frac{d+2}{2})} \tag{64}$$

$$= \frac{2}{\pi d}. \tag{65}$$

Here, the second equality is due to the identity $\mathrm{B}(a, b) = \frac{\Gamma(a)\Gamma(b)}{\Gamma(a+b)}$, and the last equality is due to the identity $\Gamma(a + 1) = a\Gamma(a)$.

Taking the expectation on both sides of Eq. (50) and using Eq. (51) and Eq. (65), we have

$$\mathbb{E}[\gamma^2] = \frac{1}{q} \cdot q \cdot \frac{1}{d} + \frac{1}{q} \cdot q(q - 1) \cdot \frac{2}{\pi d} \tag{66}$$

$$= \frac{1}{d} + \frac{2(q - 1)}{\pi d} = \frac{1}{d}\left(\frac{2}{\pi}(q - 1) + 1\right). \tag{67}$$

The proof is completed. □

## C.2 ANALYSIS FOR PRIOR-SIGN-OPT

The Prior-Sign-OPT estimator is defined in Eq. (7). Note that there are $s$ priors $\{\mathbf{p}_1, \ldots, \mathbf{p}_s\}$ (we assume that they are normalized to have unit norm), and $\{\mathbf{u}_1, \mathbf{u}_2, \ldots, \mathbf{u}_{q-s}\}$ is a uniformly random orthonormal set of $q - s$ vectors in the $(d - s)$-dimensional subspace orthogonal to $\{\mathbf{p}_1, \ldots, \mathbf{p}_s\}$. For convenience we first consider the case of $s = 1$, and the analysis can easily be generalized to the case of $s > 1$.

### C.2.1 THE CASE OF $s = 1$

When $s = 1$, we rewrite the Prior-Sign-OPT estimator in the following form:

$$\mathbf{v}^* = \mathrm{sign}(\nabla g(\theta)^\top \mathbf{p}) \cdot \mathbf{p} + \sum_{i=1}^{q-1} \mathrm{sign}(\nabla g(\theta)^\top \mathbf{u}_i) \cdot \mathbf{u}_i, \tag{68}$$

where $\mathbf{p}$ is the prior vector with $\|\mathbf{p}\| = 1$, and $\{\mathbf{u}_i\}_{i=1}^{q-1}$ is the random orthonormal basis of the $(d - 1)$-dimensional subspace orthogonal to $\mathbf{p}$. Note that the directional derivative approximation is also employed, as in Eq. (43).

**Theorem C.4.** *For the Prior-Sign-OPT estimator defined in Eq. (68), we let $\gamma := \overline{\mathbf{v}^*}^\top \overline{\nabla g(\theta)}$ be its cosine similarity to the true gradient, where the notation $\overline{\mathbf{v}^*} := \frac{\mathbf{v}^*}{\|\mathbf{v}^*\|}$ is defined to be the $\ell_2$ normalization of the corresponding vector, then*

$$\mathbb{E}[\gamma] = \frac{1}{\sqrt{q}}\left[|\alpha| + (q - 1)\sqrt{1 - \alpha^2}\frac{\Gamma(\frac{d-1}{2})}{\Gamma(\frac{d}{2})\sqrt{\pi}}\right], \tag{69}$$

$$\mathbb{E}[\gamma^2] = \frac{1}{q}\left[\alpha^2 + \frac{q - 1}{d - 1}\left(\frac{2}{\pi}(q - 2) + 1\right)(1 - \alpha^2) + 2|\alpha|(q - 1)\sqrt{1 - \alpha^2}\frac{\Gamma(\frac{d-1}{2})}{\Gamma(\frac{d}{2})\sqrt{\pi}}\right], \tag{70}$$

*where $\alpha := \mathbf{p}^\top \overline{\nabla g(\theta)}$ is the cosine similarity between the prior and the true gradient.*

*Proof.* Note that the property of sign function (e.g., $\mathrm{sign}(\nabla g(\theta)^\top \mathbf{u}) = \mathrm{sign}(\overline{\nabla g(\theta)}^\top \mathbf{u})$), in Eq. (68), we denote

$$\mathbf{v}_\perp := \sum_{i=1}^{q-1} \mathrm{sign}(\nabla g(\theta)^\top \mathbf{u}_i) \cdot \mathbf{u}_i = \sum_{i=1}^{q-1} \mathrm{sign}(\overline{\nabla g(\theta)}^\top \mathbf{u}_i) \cdot \mathbf{u}_i, \tag{71}$$

then $\mathbf{v}^* = \mathrm{sign}(\alpha)\mathbf{p} + \mathbf{v}_\perp$. Now

$$\gamma = \frac{(\mathbf{v}^*)^\top \overline{\nabla g(\theta)}}{\|\mathbf{v}^*\|} \tag{72}$$

$$= \frac{1}{\sqrt{q}}(|\alpha| + \mathbf{v}_\perp^\top \overline{\nabla g(\theta)}). \tag{73}$$

The following argument is similar to that in the proof of Lemma C.2. Let $\mathbf{g} := \overline{\nabla g(\theta)}$, and let $\mathbf{g}_\perp := \mathbf{g} - \mathbf{g}^\top \mathbf{p} \cdot \mathbf{p}$ denote the projection of $\mathbf{g}$ to the $(d - 1)$-dimensional subspace orthogonal

to $\mathbf{p}$. It follows that $\mathbf{v}_\perp^\top \mathbf{g} = \mathbf{v}_\perp^\top \mathbf{g}_\perp$. Moreover, since $\{\mathbf{u}_i\}_{i=1}^{q-1}$ are orthogonal to $\mathbf{p}$, we have $\mathbf{v}_\perp = \sum_{i=1}^{q-1} \mathrm{sign}(\mathbf{g}_\perp^\top \mathbf{u}_i) \cdot \mathbf{u}_i$. Since $\{\mathbf{u}_i\}_{i=1}^{q-1}$ are uniformly distributed on the unit hypersphere in the $(d-1)$-dimensional subspace orthogonal to $\mathbf{p}$, and $\mathbf{g}_\perp$ also resides in this subspace, $\mathbf{v}_\perp$ can be considered as the Sign-OPT estimator for $\mathbf{g}_\perp$ as in Eq. (44), with $q$ replaced by $q-1$ and the effective dimension being $d-1$ rather than $d$. By Eq. (9) we have

$$\mathbb{E}[\overline{\mathbf{v}_\perp}^\top \overline{\mathbf{g}_\perp}] = \sqrt{q-1} \frac{\Gamma(\frac{d-1}{2})}{\Gamma(\frac{d}{2})\sqrt{\pi}}. \tag{74}$$

Noting that $\|\mathbf{g}_\perp\| = \sqrt{1-\alpha^2}$ by Eq. (30) and $\|\mathbf{v}_\perp\| = \sqrt{q-1}$, we have

$$\mathbb{E}[\mathbf{v}_\perp^\top \mathbf{g}] = \mathbb{E}[\mathbf{v}_\perp^\top \mathbf{g}_\perp] = \mathbb{E}[\overline{\mathbf{v}_\perp}^\top \overline{\mathbf{g}_\perp} \|\mathbf{v}_\perp\| \|\mathbf{g}_\perp\|] \tag{75}$$

$$= (q-1)\sqrt{1-\alpha^2} \frac{\Gamma(\frac{d-1}{2})}{\Gamma(\frac{d}{2})\sqrt{\pi}}. \tag{76}$$

Taking the expectation on both sides of Eq. (73) and substituting Eq. (76), Eq. (69) has been proved.

Next we derive $\mathbb{E}[\gamma^2]$. By Eq. (73) we have

$$\gamma^2 = \frac{1}{q}(|\alpha| + \mathbf{v}_\perp^\top \mathbf{g})^2 \tag{77}$$

$$= \frac{1}{q}(\alpha^2 + (\mathbf{v}_\perp^\top \mathbf{g})^2 + 2|\alpha|\mathbf{v}_\perp^\top \mathbf{g}). \tag{78}$$

As discussed in the paragraph preceding Eq. (74), $\mathbf{v}_\perp$ can be considered as the Sign-OPT estimator for $\mathbf{g}_\perp$ as in Eq. (44), with $q$ replaced by $q-1$ and the effective dimension being $d-1$ rather than $d$. By Eq. (10) we have

$$\mathbb{E}[(\overline{\mathbf{v}_\perp}^\top \overline{\mathbf{g}_\perp})^2] = \frac{1}{d-1}\left(\frac{2}{\pi}(q-2)+1\right). \tag{79}$$

Noting that $\|\mathbf{g}_\perp\| = \sqrt{1-\alpha^2}$ by Eq. (30) and $\|\mathbf{v}_\perp\| = \sqrt{q-1}$, we have

$$\mathbb{E}[(\mathbf{v}_\perp^\top \mathbf{g})^2] = \mathbb{E}[(\mathbf{v}_\perp^\top \mathbf{g}_\perp)^2] = \mathbb{E}[(\overline{\mathbf{v}_\perp}^\top \overline{\mathbf{g}_\perp})^2 \|\mathbf{v}_\perp\|^2 \|\mathbf{g}_\perp\|^2] \tag{80}$$

$$= \frac{q-1}{d-1}\left(\frac{2}{\pi}(q-2)+1\right)(1-\alpha^2). \tag{81}$$

Taking the expectation on both sides of Eq. (78) and substituting Eq. (76) and Eq. (81), Eq. (70) has been proved. $\qquad\square$

### C.2.2 THE CASE OF $s > 1$

In the case of $s > 1$, we rewrite the Prior-Sign-OPT estimator in the following form:

$$\mathbf{v}^* = \sum_{i=1}^{s} \mathrm{sign}(\nabla g(\theta)^\top \mathbf{p}_i) \cdot \mathbf{p}_i + \sum_{i=1}^{q-s} \mathrm{sign}(\nabla g(\theta)^\top \mathbf{u}_i) \cdot \mathbf{u}_i. \tag{82}$$

Now, we present the proof of Theorem 3.3 for the Prior-Sign-OPT estimator defined in Eq. (82).

*Proof.* The following argument is similar to that in the proof of Theorem C.4. Now we have

$$\mathbf{v}_\perp := \sum_{i=1}^{q-s} \mathrm{sign}(\nabla g(\theta)^\top \mathbf{u}_i) \cdot \mathbf{u}_i = \sum_{i=1}^{q-s} \mathrm{sign}(\overline{\nabla g(\theta)}^\top \mathbf{u}_i) \cdot \mathbf{u}_i, \tag{83}$$

then $\mathbf{v}^* = \sum_{i=1}^{s} \mathrm{sign}(\alpha_i)\mathbf{p}_i + \mathbf{v}_\perp$, and

$$\gamma = \frac{(\mathbf{v}^*)^\top \overline{\nabla g(\theta)}}{\|\mathbf{v}^*\|} \tag{84}$$

$$= \frac{1}{\sqrt{q}}\left(\sum_{i=1}^{s} |\alpha_i| + \mathbf{v}_\perp^\top \overline{\nabla g(\theta)}\right). \tag{85}$$

Let $\mathbf{g} := \overline{\nabla g(\theta)}$, and let $\mathbf{g}_\perp := \mathbf{g} - \sum_{i=1}^{s} \mathbf{g}^\top \mathbf{p}_i \cdot \mathbf{p}_i$ denote the projection of $\mathbf{g}$ to the $(d-s)$-dimensional subspace orthogonal to $\{\mathbf{p}_i\}_{i=1}^{s}$. It follows that $\mathbf{v}_\perp^\top \mathbf{g} = \mathbf{v}_\perp^\top \mathbf{g}_\perp$. Using the similar analysis to that in the case of $s = 1$, when $s > 1$ we have

$$\mathbb{E}[\overline{\mathbf{v}_\perp}^\top \overline{\mathbf{g}_\perp}] = \sqrt{q-s} \frac{\Gamma(\frac{d-s}{2})}{\Gamma(\frac{d-s+1}{2})\sqrt{\pi}}. \tag{86}$$

Similar to the derivation of Eq. (30), we can derive that $\|\mathbf{g}_\perp\| = \sqrt{1 - \sum_{i=1}^{s} \alpha_i^2}$. Since $\|\mathbf{v}_\perp\| = \sqrt{q-s}$, we have

$$\mathbb{E}[\mathbf{v}_\perp^\top \mathbf{g}] = \mathbb{E}[\mathbf{v}_\perp^\top \mathbf{g}_\perp] = \mathbb{E}[\overline{\mathbf{v}_\perp}^\top \overline{\mathbf{g}_\perp} \|\mathbf{v}_\perp\| \|\mathbf{g}_\perp\|] \tag{87}$$

$$= (q-s)\sqrt{1 - \sum_{i=1}^{s} \alpha_i^2} \frac{\Gamma(\frac{d-s}{2})}{\Gamma(\frac{d-s+1}{2})\sqrt{\pi}}. \tag{88}$$

Taking the expectation on both sides of Eq. (85) and substituting Eq. (88), Eq. (11) has been proved.

Next we derive $\mathbb{E}[\gamma^2]$. By Eq. (85) we have

$$\gamma^2 = \frac{1}{q} \left( \sum_{i=1}^{s} |\alpha_i| + \mathbf{v}_\perp^\top \mathbf{g} \right)^2 \tag{89}$$

$$= \frac{1}{q} \left( \left( \sum_{i=1}^{s} |\alpha_i| \right)^2 + (\mathbf{v}_\perp^\top \mathbf{g})^2 + 2 \cdot \left( \sum_{i=1}^{s} |\alpha_i| \right) \cdot \mathbf{v}_\perp^\top \mathbf{g} \right). \tag{90}$$

Similar to the case of $s = 1$, when $s > 1$ we have

$$\mathbb{E}[(\overline{\mathbf{v}_\perp}^\top \overline{\mathbf{g}_\perp})^2] = \frac{1}{d-s} \left( \frac{2}{\pi}(q-s-1) + 1 \right). \tag{91}$$

Noting that $\|\mathbf{g}_\perp\| = \sqrt{1 - \sum_{i=1}^{s} \alpha_i^2}$ and $\|\mathbf{v}_\perp\| = \sqrt{q-s}$, we have

$$\mathbb{E}[(\mathbf{v}_\perp^\top \mathbf{g})^2] = \mathbb{E}[(\mathbf{v}_\perp^\top \mathbf{g}_\perp)^2] = \mathbb{E}[(\overline{\mathbf{v}_\perp}^\top \overline{\mathbf{g}_\perp})^2 \|\mathbf{v}_\perp\|^2 \|\mathbf{g}_\perp\|^2] \tag{92}$$

$$= \frac{q-s}{d-s} \left( \frac{2}{\pi}(q-s-1) + 1 \right) \left( 1 - \sum_{i=1}^{s} \alpha_i^2 \right). \tag{93}$$

Taking the expectation on both sides of Eq. (90) and substituting Eq. (88) and Eq. (93), Eq. (12) has been proved. □

## C.3 ANALYSIS FOR PRIOR-OPT

The Prior-OPT estimator is defined in Eq. (13). Note that there are $s$ priors $\{\mathbf{p}_1, \ldots, \mathbf{p}_s\}$ (we assume that they have been normalized so that they have unit norm), and $\{\mathbf{u}_1, \mathbf{u}_2, \ldots, \mathbf{u}_{q-s}\}$ is a uniformly random orthonormal set of $q - s$ vectors in the $(d-s)$-dimensional subspace orthogonal to $\{\mathbf{p}_1, \ldots, \mathbf{p}_s\}$. For convenience, we first consider the case of $s = 1$, and the analysis can easily be generalized to the case of $s > 1$.

### C.3.1 THE CASE OF $s = 1$

When $s = 1$, we rewrite the Prior-OPT estimator in the following form:

$$\mathbf{v}^* = \nabla g(\theta)^\top \mathbf{p} \cdot \mathbf{p} + \nabla g(\theta)^\top \overline{\mathbf{v}_\perp} \cdot \overline{\mathbf{v}_\perp}, \tag{94}$$

where

$$\mathbf{v}_\perp := \sum_{i=1}^{q-1} \text{sign}(\nabla g(\theta)^\top \mathbf{u}_i) \cdot \mathbf{u}_i. \tag{95}$$

Here $\mathbf{p}$ is the prior vector with $\|\mathbf{p}\| = 1$, and $\{\mathbf{u}_i\}_{i=1}^{q-1}$ is the random orthonormal basis of the $(d-1)$-dimensional subspace orthogonal to $\mathbf{p}$. Note that we employ the directional derivative approximation as in Eq. (43). Furthermore, it should also be noted that $\mathbf{v}_\perp$ defined in Eq. (95) is consistent with that in Eq. (71), and thus the conclusions regarding $\mathbf{v}_\perp$ derived in Appendix C.2.1 remain valid in this section (e.g., Eq. (76)).

**Theorem C.5.** *For the Prior-OPT estimator defined in Eq. (94), we let $\gamma := \overline{\mathbf{v}^*}^\top \overline{\nabla g(\theta)}$ be its cosine similarity to the true gradient, where the notation $\overline{\mathbf{v}^*} := \frac{\mathbf{v}^*}{\|\mathbf{v}^*\|}$ is defined to be the $\ell_2$ normalization of the corresponding vector, then*

$$\mathbb{E}[\gamma] \geq \sqrt{\alpha^2 + \frac{(q-1)(1-\alpha^2)}{\pi}\left(\frac{\Gamma(\frac{d-1}{2})}{\Gamma(\frac{d}{2})}\right)^2}, \tag{96}$$

$$\mathbb{E}[\gamma] \leq \sqrt{\alpha^2 + \frac{1}{d-1}\left(\frac{2}{\pi}(q-2)+1\right)(1-\alpha^2)}, \tag{97}$$

$$\mathbb{E}[\gamma^2] = \alpha^2 + \frac{1}{d-1}\left(\frac{2}{\pi}(q-2)+1\right)(1-\alpha^2), \tag{98}$$

*where $\alpha := \mathbf{p}^\top \overline{\nabla g(\theta)}$ is the cosine similarity between the prior and the true gradient.*

*Proof.* Let $\mathbf{g} := \overline{\nabla g(\theta)}$. Then $\mathbf{v}^* = \|\nabla g(\theta)\|(\mathbf{g}^\top \mathbf{p} \cdot \mathbf{p} + \mathbf{g}^\top \overline{\mathbf{v}_\perp} \cdot \overline{\mathbf{v}_\perp})$. We also note that $\mathbf{v}_\perp$ is a linear combination of $\mathbf{u}_1$ to $\mathbf{u}_{q-1}$, all of which are orthogonal to $\mathbf{p}$, so $\mathbf{v}_\perp$ is also orthogonal to $\mathbf{p}$. Therefore, $\|\mathbf{v}^*\| = \|\nabla g(\theta)\|\sqrt{(\mathbf{p}^\top \mathbf{g})^2 + (\overline{\mathbf{v}_\perp}^\top \mathbf{g})^2}$. Hence

$$\gamma = \frac{(\mathbf{v}^*)^\top \overline{\nabla g(\theta)}}{\|\mathbf{v}^*\|} \tag{99}$$

$$= \frac{(\mathbf{p}^\top \mathbf{g})^2 + (\overline{\mathbf{v}_\perp}^\top \mathbf{g})^2}{\sqrt{(\mathbf{p}^\top \mathbf{g})^2 + (\overline{\mathbf{v}_\perp}^\top \mathbf{g})^2}} \tag{100}$$

$$= \sqrt{(\mathbf{p}^\top \mathbf{g})^2 + (\overline{\mathbf{v}_\perp}^\top \mathbf{g})^2}. \tag{101}$$

We define a new estimator

$$\widetilde{\mathbf{v}^*} := \nabla g(\theta)^\top \mathbf{p} \cdot \mathbf{p} + \mathbb{E}[\overline{\mathbf{v}_\perp}^\top \nabla g(\theta)] \cdot \overline{\mathbf{v}_\perp} \tag{102}$$

$$= \|\nabla g(\theta)\|\left(\mathbf{g}^\top \mathbf{p} \cdot \mathbf{p} + \mathbb{E}[\overline{\mathbf{v}_\perp}^\top \mathbf{g}] \cdot \overline{\mathbf{v}_\perp}\right). \tag{103}$$

Let $\widetilde{\gamma}$ be the cosine similarity between $\widetilde{\mathbf{v}^*}$ and $\nabla g(\theta)$. Then

$$\widetilde{\gamma} := \frac{(\widetilde{\mathbf{v}^*})^\top \overline{\nabla g(\theta)}}{\|\widetilde{\mathbf{v}^*}\|} \tag{104}$$

$$= \frac{(\mathbf{p}^\top \mathbf{g})^2 + \mathbb{E}[\overline{\mathbf{v}_\perp}^\top \mathbf{g}]\overline{\mathbf{v}_\perp}^\top \mathbf{g}}{\sqrt{(\mathbf{p}^\top \mathbf{g})^2 + \mathbb{E}[\overline{\mathbf{v}_\perp}^\top \mathbf{g}]^2}}. \tag{105}$$

Therefore,

$$\mathbb{E}[\widetilde{\gamma}] = \mathbb{E}\left[\frac{(\mathbf{p}^\top \mathbf{g})^2 + \mathbb{E}[\overline{\mathbf{v}_\perp}^\top \mathbf{g}]\overline{\mathbf{v}_\perp}^\top \mathbf{g}}{\sqrt{(\mathbf{p}^\top \mathbf{g})^2 + \mathbb{E}[\overline{\mathbf{v}_\perp}^\top \mathbf{g}]^2}}\right] \tag{106}$$

$$= \frac{\alpha^2 + \mathbb{E}[\overline{\mathbf{v}_\perp}^\top \mathbf{g}]^2}{\sqrt{\alpha^2 + \mathbb{E}[\overline{\mathbf{v}_\perp}^\top \mathbf{g}]^2}} \tag{107}$$

$$= \sqrt{\alpha^2 + \mathbb{E}[\overline{\mathbf{v}_\perp}^\top \mathbf{g}]^2}. \tag{108}$$

Since $\mathbb{E}[\overline{\mathbf{v}_\perp}^\top \mathbf{g}] = \frac{1}{\|\mathbf{v}_\perp\|}\mathbb{E}[\mathbf{v}_\perp^\top \mathbf{g}]$, we substitute Eq. (76) and $\|\mathbf{v}_\perp\| = \sqrt{q-1}$ into this expression, yielding $\mathbb{E}[\overline{\mathbf{v}_\perp}^\top \mathbf{g}] = \sqrt{q-1}\sqrt{1-\alpha^2}\frac{\Gamma(\frac{d-1}{2})}{\Gamma(\frac{d}{2})\sqrt{\pi}}$. Hence,

$$\mathbb{E}[\widetilde{\gamma}] = \sqrt{\alpha^2 + \frac{(q-1)(1-\alpha^2)}{\pi}\left(\frac{\Gamma(\frac{d-1}{2})}{\Gamma(\frac{d}{2})}\right)^2}. \tag{109}$$

The remaining part is to show the relationship between $\mathbb{E}[\gamma]$ and $\mathbb{E}[\widehat{\gamma}]$. Note that $\mathbf{v}^*$ is the projection of $\nabla g(\theta)$ on the 2-dimensional subspace spanned by $\mathbf{p}$ and $\overline{\mathbf{v}_\perp}$. By Proposition 1 in Meier et al. (2019), among all the vectors in the subspace spanned by $\mathbf{p}$ and $\overline{\mathbf{v}_\perp}$, $\mathbf{v}^*$ has the largest cosine similarity with $\nabla g(\theta)$. Since $\widetilde{\mathbf{v}^*}$ is a linear combination of $\mathbf{p}$ and $\overline{\mathbf{v}_\perp}$, $\gamma \geq \widetilde{\gamma}$ always holds. Therefore, $\mathbb{E}[\gamma] \geq \mathbb{E}[\widehat{\gamma}]$, which directly proves the lower bound given in Eq. (96).

Next, we derive $\mathbb{E}[\gamma^2]$. By Eq. (99) we have

$$\mathbb{E}[\gamma^2] = \alpha^2 + \mathbb{E}[(\overline{\mathbf{v}_\perp}^\top \mathbf{g})^2]. \tag{110}$$

Since $\|\mathbf{v}_\perp\| = \sqrt{q-1}$, using Eq. (81) we have

$$\mathbb{E}[(\overline{\mathbf{v}_\perp}^\top \mathbf{g})^2] = \frac{1}{\|\mathbf{v}_\perp\|^2} \mathbb{E}[(\mathbf{v}_\perp^\top \mathbf{g})^2] \tag{111}$$

$$= \frac{1}{d-1} \left( \frac{2}{\pi}(q-2) + 1 \right) (1 - \alpha^2). \tag{112}$$

Plugging Eq. (112) into Eq. (110), we obtain Eq. (98).

Finally, by applying Jensen's inequality $(\mathbb{E}[\gamma])^2 \leq \mathbb{E}[\gamma^2]$, we derive the upper bound for $\mathbb{E}[\gamma]$, which leads to the following result:

$$\mathbb{E}[\gamma] \leq \sqrt{\mathbb{E}[\gamma^2]} = \sqrt{\alpha^2 + \frac{1}{d-1} \left( \frac{2}{\pi}(q-2) + 1 \right) (1 - \alpha^2)}. \tag{113}$$

This establishes Eq. (97), thereby completing the proof. $\qquad\square$

### C.3.2 THE CASE OF $s > 1$

In the case of $s > 1$, we rewrite the Prior-OPT estimator in the following form:

$$\mathbf{v}^* = \sum_{i=1}^{s} \nabla g(\theta)^\top \mathbf{p}_i \cdot \mathbf{p}_i + \nabla g(\theta)^\top \overline{\mathbf{v}_\perp} \cdot \overline{\mathbf{v}_\perp}, \tag{114}$$

where

$$\mathbf{v}_\perp := \sum_{i=1}^{q-s} \text{sign}(\nabla g(\theta)^\top \mathbf{u}_i) \cdot \mathbf{u}_i. \tag{115}$$

Note that Eq. (115) approximates Eq. (14) under the directional derivative approximation. Furthermore, it should also be noted that $\mathbf{v}_\perp$ defined in Eq. (115) is consistent with that in Eq. (83), and thus the conclusions regarding $\mathbf{v}_\perp$ derived in Appendix C.2.2 remain valid in this section (e.g., Eq. (88)).

Now, we present the proof of Theorem 3.4 for the Prior-OPT estimator defined in Eq. (114).

*Proof.* Let $\mathbf{g} := \overline{\nabla g(\theta)}$. Then

$$\mathbf{v}^* = \|\nabla g(\theta)\| \left( \sum_{i=1}^{s} \mathbf{g}^\top \mathbf{p}_i \cdot \mathbf{p}_i + \mathbf{g}^\top \overline{\mathbf{v}_\perp} \cdot \overline{\mathbf{v}_\perp} \right). \tag{116}$$

We also note that $\mathbf{v}_\perp$ is a linear combination of $\mathbf{u}_1$ to $\mathbf{u}_{q-s}$, all of which are orthogonal to $\{\mathbf{p}_i\}_{i=1}^s$, so $\mathbf{v}_\perp$ is also orthogonal to $\{\mathbf{p}_i\}_{i=1}^s$. Therefore, $\|\mathbf{v}^*\| = \|\nabla g(\theta)\| \sqrt{\sum_{i=1}^s (\mathbf{p}_i^\top \mathbf{g})^2 + (\overline{\mathbf{v}_\perp}^\top \mathbf{g})^2}$. Hence

$$\gamma = \frac{(\mathbf{v}^*)^\top \overline{\nabla g(\theta)}}{\|\mathbf{v}^*\|} \tag{117}$$

$$= \sqrt{\sum_{i=1}^{s} (\mathbf{p}_i^\top \mathbf{g})^2 + (\overline{\mathbf{v}_\perp}^\top \mathbf{g})^2}. \tag{118}$$

We define a new estimator

$$\widetilde{\mathbf{v}^*} := \sum_{i=1}^{s} \nabla g(\theta)^\top \mathbf{p}_i \cdot \mathbf{p}_i + \mathbb{E}[\overline{\mathbf{v}_\perp}^\top \nabla g(\theta)] \cdot \overline{\mathbf{v}_\perp} \tag{119}$$

$$= \|\nabla g(\theta)\| \left( \sum_{i=1}^{s} \mathbf{g}^\top \mathbf{p}_i \cdot \mathbf{p}_i + \mathbb{E}[\overline{\mathbf{v}_\perp}^\top \mathbf{g}] \cdot \overline{\mathbf{v}_\perp} \right). \tag{120}$$

Let $\widetilde{\gamma}$ be the cosine similarity between $\widetilde{\mathbf{v}^*}$ and $\nabla g(\theta)$. Then

$$\widetilde{\gamma} := \frac{(\widetilde{\mathbf{v}^*})^\top \overline{\nabla g(\theta)}}{\|\widetilde{\mathbf{v}^*}\|} \tag{121}$$

$$= \frac{\sum_{i=1}^{s} (\mathbf{p}_i^\top \mathbf{g})^2 + \mathbb{E}[\overline{\mathbf{v}_\perp}^\top \mathbf{g}] \overline{\mathbf{v}_\perp}^\top \mathbf{g}}{\sqrt{\sum_{i=1}^{s} (\mathbf{p}_i^\top \mathbf{g})^2 + \mathbb{E}[\overline{\mathbf{v}_\perp}^\top \mathbf{g}]^2}}. \tag{122}$$

Therefore,

$$\mathbb{E}[\widetilde{\gamma}] = \mathbb{E} \left[ \frac{\sum_{i=1}^{s} (\mathbf{p}_i^\top \mathbf{g})^2 + \mathbb{E}[\overline{\mathbf{v}_\perp}^\top \mathbf{g}] \overline{\mathbf{v}_\perp}^\top \mathbf{g}}{\sqrt{\sum_{i=1}^{s} (\mathbf{p}_i^\top \mathbf{g})^2 + \mathbb{E}[\overline{\mathbf{v}_\perp}^\top \mathbf{g}]^2}} \right] \tag{123}$$

$$= \frac{\sum_{i=1}^{s} \alpha_i^2 + \mathbb{E}[\overline{\mathbf{v}_\perp}^\top \mathbf{g}]^2}{\sqrt{\sum_{i=1}^{s} \alpha_i^2 + \mathbb{E}[\overline{\mathbf{v}_\perp}^\top \mathbf{g}]^2}} \tag{124}$$

$$= \sqrt{\sum_{i=1}^{s} \alpha_i^2 + \mathbb{E}[\overline{\mathbf{v}_\perp}^\top \mathbf{g}]^2}. \tag{125}$$

Since $\mathbb{E}[\overline{\mathbf{v}_\perp}^\top \mathbf{g}] = \frac{1}{\|\mathbf{v}_\perp\|} \mathbb{E}[\mathbf{v}_\perp^\top \mathbf{g}]$, we substitute Eq. (88) and $\|\mathbf{v}_\perp\| = \sqrt{q-s}$ into this expression, yielding $\mathbb{E}[\overline{\mathbf{v}_\perp}^\top \mathbf{g}] = \sqrt{q-s} \sqrt{1 - \sum_{i=1}^{s} \alpha_i^2} \frac{\Gamma(\frac{d-s}{2})}{\Gamma(\frac{d-s+1}{2})\sqrt{\pi}}$. Hence,

$$\mathbb{E}[\widetilde{\gamma}] = \sqrt{\sum_{i=1}^{s} \alpha_i^2 + \frac{(q-s)(1 - \sum_{i=1}^{s} \alpha_i^2)}{\pi} \left( \frac{\Gamma(\frac{d-s}{2})}{\Gamma(\frac{d-s+1}{2})} \right)^2}. \tag{126}$$

The remaining part is to show the relationship between $\mathbb{E}[\gamma]$ and $\mathbb{E}[\widetilde{\gamma}]$. We note that $\mathbf{v}^*$ is the projection of $\nabla g(\theta)$ on the $(s+1)$-dimensional subspace spanned by $\{\mathbf{p}_1, \mathbf{p}_2, \ldots, \mathbf{p}_s, \overline{\mathbf{v}_\perp}\}$. By Proposition 1 in Meier et al. (2019), among all the vectors in the subspace spanned by $\{\mathbf{p}_1, \mathbf{p}_2, \ldots, \mathbf{p}_s, \overline{\mathbf{v}_\perp}\}$, $\mathbf{v}^*$ has the largest cosine similarity with $\nabla g(\theta)$. Since $\widetilde{\mathbf{v}^*}$ also lies in this subspace, $\gamma \geq \widetilde{\gamma}$ always holds. Therefore, $\mathbb{E}[\gamma] \geq \mathbb{E}[\widetilde{\gamma}]$, which directly proves the lower bound given in Eq. (15).

Next, we derive $\mathbb{E}[\gamma^2]$. By Eq. (117) we have

$$\mathbb{E}[\gamma^2] = \sum_{i=1}^{s} \alpha_i^2 + \mathbb{E}[(\overline{\mathbf{v}_\perp}^\top \mathbf{g})^2]. \tag{127}$$

Since $\|\mathbf{v}_\perp\| = \sqrt{q-s}$, using Eq. (93) we have

$$\mathbb{E}[(\overline{\mathbf{v}_\perp}^\top \mathbf{g})^2] = \frac{1}{\|\mathbf{v}_\perp\|^2} \mathbb{E}[(\mathbf{v}_\perp^\top \mathbf{g})^2] \tag{128}$$

$$= \frac{1}{d-s} \left( \frac{2}{\pi}(q-s-1) + 1 \right) \left( 1 - \sum_{i=1}^{s} \alpha_i^2 \right). \tag{129}$$

Plugging Eq. (129) into Eq. (127), we obtain Eq. (17).

Finally, by applying Jensen's inequality $(\mathbb{E}[\gamma])^2 \leq \mathbb{E}[\gamma^2]$, we derive the upper bound for $\mathbb{E}[\gamma]$, which leads to the following result:

$$\mathbb{E}[\gamma] \leq \sqrt{\mathbb{E}[\gamma^2]} = \sqrt{\sum_{i=1}^{s} \alpha_i^2 + \frac{1}{d-s} \left( \frac{2}{\pi}(q-s-1) + 1 \right) \left( 1 - \sum_{i=1}^{s} \alpha_i^2 \right)}. \tag{130}$$

This establishes Eq. (16), thereby completing the proof. $\square$

## D   DERIVATION OF THE CONDITION FOR PRIOR-OPT TO OUTPERFORM SIGN-OPT

With the formulas of $\mathbb{E}[\gamma^2]$ of Sign-OPT (Eq. (10)) and Prior-OPT (Eq. (17)), we now derive the exact value of $\alpha_i$ for which Prior-OPT can outperform Sign-OPT on gradient estimation.

Now, we rewrite the formulas of $\mathbb{E}[\gamma^2]$ of Sign-OPT and Prior-OPT as follows:

$$\mathbb{E}[\gamma^2]_{\text{Sign-OPT}} = \frac{1}{d}\left(\frac{2}{\pi}(q-1)+1\right), \tag{131}$$

$$\mathbb{E}[\gamma^2]_{\text{Prior-OPT}} = \sum_{i=1}^{s}\alpha_i^2 + \frac{1}{d-s}\left(\frac{2}{\pi}(q-s-1)+1\right)\left(1-\sum_{i=1}^{s}\alpha_i^2\right). \tag{132}$$

We need to find the value of $\alpha_i$ such that $\mathbb{E}[\gamma^2]_{\text{Prior-OPT}} > \mathbb{E}[\gamma^2]_{\text{Sign-OPT}}$.

Let $A := \sum_{i=1}^{s}\alpha_i^2$, and the inequality becomes:

$$A + (1-A)\cdot\frac{1}{d-s}\left(\frac{2}{\pi}(q-s-1)+1\right) > \frac{1}{d}\left(\frac{2}{\pi}(q-1)+1\right). \tag{133}$$

Now let us simplify the left side of Eq. (133) to $\mathbb{E}[\gamma^2]_{\text{Prior-OPT}} = A + (1-A)C_2$, where $C_2 := \frac{1}{d-s}\left(\frac{2}{\pi}(q-s-1)+1\right)$.

Then, let us simplify the right side of Eq. (133) to $\mathbb{E}[\gamma^2]_{\text{Sign-OPT}} = C_1$, where $C_1 := \frac{1}{d}\left(\frac{2}{\pi}(q-1)+1\right)$.

The inequality of Eq. (133) becomes:

$$A + (1-A)C_2 > C_1. \tag{134}$$

We rearrange the above inequality as $A(1-C_2) + C_2 > C_1$, and then we solve for $A$:

$$A > \frac{C_1 - C_2}{1 - C_2}. \tag{135}$$

Substituting the formulas of $A$, $C_1$, and $C_2$ into Eq. (135), we have:

$$\sum_{i=1}^{s}\alpha_i^2 > \frac{\frac{1}{d}\left(\frac{2}{\pi}(q-1)+1\right) - \frac{1}{d-s}\left(\frac{2}{\pi}(q-s-1)+1\right)}{1 - \frac{1}{d-s}\left(\frac{2}{\pi}(q-s-1)+1\right)}. \tag{136}$$

This is the condition of $\sum_{i=1}^{s}\alpha_i^2$ for Prior-OPT to outperform Sign-OPT. But this inequality is complex, next we show how to further simplify this inequality. Under the reasonable assumptions that $q \ll d$, which implies that the input dimension is much larger than the total number of vectors (and consequently $s \ll d$ since $s < q$), the above inequality can be simplified.

We first approximate denominator of Eq. (136), note that when $s \ll d$, we have $d - s \approx d$. Therefore, the denominator simplifies to:

$$D := 1 - \frac{1}{d-s}\left(\frac{2}{\pi}(q-s-1)+1\right) \approx 1 - \frac{1}{d}\left(\frac{2}{\pi}(q-s-1)+1\right). \tag{137}$$

Since $\frac{1}{d}\left(\frac{2}{\pi}(q-s-1)+1\right)$ is a small number because $q \ll d$ (denote it as $\epsilon$), the denominator becomes $D \approx 1 - \epsilon \approx 1$. Next, we simplify the numerator as:

$$N := \frac{1}{d}\left(\frac{2}{\pi}(q-1)+1\right) - \frac{1}{d-s}\left(\frac{2}{\pi}(q-s-1)+1\right) \tag{138}$$

$$\approx \frac{1}{d}\left(\frac{2}{\pi}(q-1)+1\right) - \frac{1}{d}\left(\frac{2}{\pi}(q-s-1)+1\right) \tag{139}$$

$$= \frac{1}{d}\left(\frac{2}{\pi}(q-1)+1 - \left(\frac{2}{\pi}(q-s-1)+1\right)\right) \tag{140}$$

$$= \frac{2s}{\pi d}. \tag{141}$$

Now, let us substitute the simplified $N$ and $D$ into the right side of Eq. (136), we have

$$\sum_{i=1}^{s} \alpha_i^2 > \frac{N}{D} \approx \frac{2s}{\pi d}. \tag{142}$$

This is the simplified condition of $\sum_{i=1}^{s} \alpha_i^2$ for Prior-OPT to outperform Sign-OPT.

Dividing both sides by $s$, we get the condition for the average squared cosine similarity $\overline{\alpha^2} := \frac{1}{s}\sum_{i=1}^{s} \alpha_i^2 > \frac{2}{\pi d}$. Since $\frac{2}{\pi d}$ is typically a very small value due to the large input dimension $d$, this threshold is relatively easy to satisfy. Therefore, Prior-OPT generally outperforms Sign-OPT when the priors have even a minimal level of informativeness (non-zero $\alpha_i$).

## E   DISCUSSIONS

### E.1   PRIOR ACCURACY $\alpha_i$

$\alpha_i = \mathbf{p}_i^\top \overline{\nabla g(\theta)}$ is the cosine similarity between the $i$-th surrogate model's gradient (the $i$-th prior) and the true gradient. The value of $\alpha_i$ is only used in the theoretical analysis and is not required for practical algorithm. Algorithm 1 does not require any $\alpha_i$ or the true gradient to run. We assume that $\alpha_i$ is known in the theoretical analysis so that we can analyze its impact on the expectation of the final estimated gradient's cosine similarity $\gamma$ to the true gradient, which derives the solutions of $\mathbb{E}[\gamma]$ and $\mathbb{E}[\gamma^2]$. Figs. 5 and 13 demonstrate the quantitative analysis for $\mathbb{E}[\gamma]$ and $\mathbb{E}[\gamma^2]$, respectively.

### E.2   DIFFERENCES BETWEEN OPT AND PRIOR-OPT

Although the gradient estimation formulas in OPT (Cheng et al., 2019) and Prior-OPT (Eq. (13)) exhibit some similarities, they differ in two key aspects.

First, the formula of Prior-OPT (Eq. (13)) is not identical to that of OPT. In Eq. (13), the last term involves the $\ell_2$ normalization of $\mathbf{v}_\perp$, where $\mathbf{v}_\perp = \sum_{i=1}^{q-s} \text{sign}(g(\theta + \sigma\mathbf{u}_i) - g(\theta)) \cdot \mathbf{u}_i$. and $\mathbf{u}_1, \ldots, \mathbf{u}_{q-s}$ are orthonormal random vectors. Consequently, Prior-OPT employs more precise finite difference estimation for the priors (the first term), while relying on sign-based estimation for the random vector components. This distinction arises because random vectors $\mathbf{u}_1, \ldots, \mathbf{u}_{q-s}$ are identically distributed, leading to a relatively consistent cosine similarity with the true gradient. This observation enables efficient sign-based estimation for random vectors. In contrast, the cosine similarities between the prior directions $\mathbf{p}_1, \ldots, \mathbf{p}_s$ and the true gradient $\nabla g(\theta)$ are unknown and may differ significantly. Thus, the coefficients for priors require more precise estimation, necessitating a separate binary search procedure. Therefore, Prior-OPT is not merely a simple extension of OPT that incorporates priors, as it handles priors and random directions differently to address these challenges.

Second, OPT does not require its random directions to be orthogonality, while Prior-OPT explicitly does. Although a small number of randomly sampled vectors are approximately orthogonal in the high-dimensional space, this is not always the case for *multiple priors*. Priors derived from potentially correlated models are less likely to be orthogonal to each other. If the Gram-Schmidt orthonormalization is omitted, the estimated gradient obtained using Eq. (7) and Eq. (13) may become less accurate, potentially degrading performance. Furthermore, the formulas of $\mathbb{E}[\gamma]$ and $\mathbb{E}[\gamma^2]$ derived from our theoretical analysis would no longer hold in such scenarios.

### E.3 PRACTICALITY OF THEORY

No matter how complex a real-world situation is, the generation of adversarial examples mainly relies on gradient vectors that increases the classification loss to cause misclassification. Our theory focuses on the similarity between the estimated gradient and the true gradient, and it is applicable to all image classifiers. One of the most common challenges in real-world scenarios is the significant difference between models, leading to discrepancies in their gradients. We address this issue by introducing the variable $\alpha_i$ in Eqs. (11), (12), (15), (16), and (17), where $\alpha_i$ represents the cosine similarity between the $i$-th prior and the true gradient and is assumed to be known in our theoretical analysis. In summary, our theory is universally applicable to real-world scenarios.

## F EXPERIMENTAL SETTINGS

In this section, we provide the hyperparameter settings for our approach and the compared methods, which include HSJA, TA, G-TA, GeoDA, Evolutionary, Triangle Attack, SurFree, Sign-OPT, SVM-OPT, SQBA, and BBA.

**Experimental Equipment.** The experiments of all methods are conducted using PyTorch 1.7.1 framework on NVIDIA V100 and A100 GPUs. NVIDIA A100 GPU has TensorFloat-32 (TF32) tensor cores to improve computation speed, and enabling TF32 tensor cores causes a large relative error compared to double precision, especially in attacks on ViTs. Therefore, in all experiments, we set `torch.backends.cuda.matmul.allow_tf32 = False` and `torch.backends.cudnn.allow_tf32 = False` to obtain higher precision.

**CIFAR-10 dataset.** In the CIFAR-10 dataset, we select four networks as target models, including a 272-layer PyramidNet+ShakeDrop network (PyramidNet-272) (Han et al., 2017; Yamada et al., 2019), two wide residual networks with 28 and 40 layers (WRN-28 and WRN-40) (Zagoruyko & Komodakis, 2016), and DenseNet-BC-190 ($k = 40$) (Huang et al., 2017). We use ResNet-110 as the surrogate model in the CIFAR-10 dataset.

**Prior-OPT and Prior-Sign-OPT.** Table 3 lists the hyperparameters of Prior-OPT and Prior-Sign-OPT. Our implementation is based on the PyTorch framework. In the targeted attack experiments on a given target class, we initialize the direction $\theta_0$ for both Prior-OPT and Prior-Sign-OPT toward the same reference image from that class, consistent with all baseline methods.

Table 3: The hyperparameters of Prior-OPT and Prior-Sign-OPT.

| Dataset | Hyperparameter | Value |
|---|---|---|
| CIFAR-10 | $q$, total number of vectors for estimating a gradient, including priors and random vectors | 200 |
| | the binary search's stopping threshold | $\frac{\beta}{500}$ |
| | the number of iterations | 1,000 |
| | $\mathbf{g}_{\max}$, the maximum gradient norm for the gradient clipping operation | 0.1 |
| ImageNet | $q$, total number of vectors for estimating a gradient, including priors and random vectors | 200 |
| | the binary search's stopping threshold | $1 \times 10^{-4}$ |
| | the number of iterations | 1,000 |
| | $\mathbf{g}_{\max}$, the maximum gradient norm for the gradient clipping operation | 1.0 |

Table 4: The hyperparameters of Sign-OPT and SVM-OPT.

| Hyperparameter | Value |
|---|---|
| $q$, the number of queries for estimating an approximate gradient | 200 |
| the number of iterations | 1,000 |
| the binary search's stopping threshold of the CIFAR-10 dataset | $\frac{\beta}{500}$ |
| the binary search's stopping threshold of the ImageNet dataset | $1 \times 10^{-4}$ |

Table 5: The hyperparameters of HSJA, TA and G-TA.

| Hyperparameter | Value |
|---|---|
| $\gamma$, threshold of the binary search | 1.0 |
| $B_0$, the initial batch size for gradient estimation | 100 |
| $B_{max}$, the maximum batch size for gradient estimation | 10,000 |
| the search method for step size | geometric progression |
| the number of iterations | 64 |
| radius ratio $r$ for the ImageNet dataset in G-TA | 1.1 |
| radius ratio $r$ for the CIFAR-10 dataset in G-TA | 1.5 |

Table 6: The hyperparameters of Evolutionary.

| Hyperparameter | Value |
|---|---|
| $c_{cov}$, the hyperparameter of updating the diagonal covariance matrix $\mathbf{C}$ | 0.001 |
| $\sigma$, the deviation for bias | 0.03 |
| $\mu$, a critical hyperparameter controlling the strength of going towards the original image | 0.01 |
| `maxlen`, the maximum length of successful attacks for calculating $\mu$ | 30 |

Table 7: The hyperparameters of GeoDA.

| Dataset | Hyperparameter | Value |
|---|---|---|
| CIFAR-10 | subspace dimension, the dimension of 2D DCT basis's subspace | 10 |
| | $\epsilon$, the step size of searching the decision boundary | 0.5 |
| ImageNet | subspace dimension, the dimension of 2D DCT basis's subspace | 75 |
| | $\epsilon$, the step size of searching the decision boundary | 5 |

Table 8: The hyperparameters of Triangle Attack.

| Dataset | Hyperparameter | Value |
|---|---|---|
| CIFAR-10 | $d$, the number of picked dimensions | 3 |
| | `ratio mask`, the ratio of the mask size for obtaining the low-frequency mask | 0.3 |
| | $\theta_{init}$, the initial angle of the subspace equals $\theta_{init} \times \pi/32$ | 2 |
| | $\alpha_{init}$, the initial angle of alpha | $\pi/2$ |
| | the maximum iteration number of attack algorithm in 2D subspace | 2 |
| ImageNet | $d$, the number of picked dimensions | 3 |
| | `ratio mask`, the ratio of the mask size for obtaining the low-frequency mask | 0.1 |
| | $\theta_{init}$, the initial angle of the subspace equals $\theta_{init} \times \pi/32$ | 2 |
| | $\alpha_{init}$, the initial angle of alpha | $\pi/2$ |
| | the maximum iteration number of attack algorithm in 2D subspace | 2 |

**Sign-OPT and SVM-OPT.** Table 4 lists the hyperparameters of Sign-OPT and SVM-OPT. For fair comparison, we set the hyperparameters of Prior-OPT and Prior-Sign-OPT to be the same as those of Sign-OPT and SVM-OPT, e.g., using the same number of vectors for the gradient estimation.

**HSJA, TA and G-TA.** Table 5 lists the hyperparameters of HSJA, TA, and G-TA. TA has no additional hyperparameters. G-TA has an additional hyperparameter, the radius ratio $r$, to control the shape of the virtual semi-ellipsoid. Specifically, $r$ is set to 1.1 for ImageNet and 1.5 for CIFAR-10.

**Evolutionary.** We follow the official source code of Evolutionary to set its hyperparameters, as shown in Table 6.

**GeoDA.** GeoDA only supports untargeted attacks, and the convergence of $\ell_2$-norm attacks of GeoDA is theoretically guaranteed. Thus, we conduct untargeted $\ell_2$-norm attack experiments using GeoDA, and the hyperparameters of GeoDA are shown in Table 7.

Table 9: The hyperparameters of SurFree.

| Hyperparameter | Value |
|---|---|
| `BS_gamma`, the stopping threshold in the binary search of $\alpha$ | 0.01 |
| `BS_max_iteration`, the maximum iterations in the binary search for $\alpha$ | 10 |
| $\rho$, the parameter for determining $\theta_{max}$ | 0.98 |
| `T`, the parameter for determining the range of $\alpha$ and the best $\theta$ | 3 |
| $\theta_{max}$, the parameter for determining the range of $\alpha$ | 30 |
| `n_ortho`, the parameter for finding the direction of the lowest $\epsilon$ in `_get_candidates` | 100 |
| the binary search's stopping threshold of the ImageNet dataset | $1 \times 10^{-4}$ |
| `frequence_range`, the parameter used in constructing `dct_mask` | $0 \sim 0.5$ |
| `with_distance_line_search`, the parameter used in `_get_candidates` | False |
| `with_interpolation`, the parameter used in `_get_candidates` | False |
| `with_alpha_line_search`, the parameter used in `_get_best_theta` | True |

Table 10: The hyperparameters of SQBA.

| Hyperparameter | Value |
|---|---|
| `threshold`, the stopping threshold in the binary search | 0.001 |
| `min_randoms`, the value indirectly determines the number of queries in each gradient estimation | 10 |

Table 11: The hyperparameters of BBA.

| Hyperparameter | Value |
|---|---|
| `use_surrogate_bias`, whether to use a surrogate model as the bias | True |
| `use_mask_bias`, whether to use regional masks as the bias | False |
| `use_perlin_bias`, whether to use Perlin Noise as the bias | False |
| `pg_factor`, the hyperparameter that controls the strength of the bias | 0.3 |

**Triangle Attack.** We set the hyperparameter "ratio mask" to 0.1 for ImageNet and 0.3 for CIFAR-10, respectively. All hyperparameters of Triangle Attack are shown in Table 8.

**SurFree.** SurFree only supports the $\ell_2$-norm attacks. We adapt the official version of SurFree's code to PyTorch for our experiments, and its hyperparameters are detailed in Table 9.

**SQBA.** SQBA only supports the untargeted $\ell_2$-norm attacks, and its hyperparameters are shown in Table 10.

**BBA.** BBA only supports the $\ell_2$-norm attacks, and the hyperparameters of BBA are shown in Table 11. We only use the bias of the surrogate model, and the hyperparameter `pg_factor` controls the strength of this bias. When `pg_factor` = 1, the orthogonal step is equivalent to one iteration of the PGD attack. Brunner et al. (2019) suggest that `pg_factor` = 0.3.

## G    ADDITIONAL EXPERIMENTAL RESULTS

In this section, we present the results of the computational overhead tests and additional experiments.

### G.1    COMPUTATIONAL OVERHEAD

The primary additional computational cost of Prior-OPT over Sign-OPT stems from: (1) the binary search procedure during gradient estimation, and (2) the time required to obtain priors. Let $d$ denote the dimension of the input image, $q$ denote the number of vectors used in gradient estimation, $f(d)$ denote the inference time of the target model for an input of dimension $d$, and $\hat{f}(d)$ denote the gradient computation time (i.e., the forward and backward pass time) of the surrogate model on an input of dimension $d$. The time complexity of gradient estimation in Sign-OPT is $O(q \cdot f(d))$. In Prior-OPT, $s$ priors are introduced. Each prior requires a binary search procedure, which involves approximately $k$ inference steps. While $k$ may vary slightly depending on the specific prior or the

input configuration, its value remains bounded and logarithmic in scale, given the nature of binary search. Consequently, the time complexity of Prior-OPT's gradient estimation can be expressed as:

$$O\left((q - s + (s + 1) \cdot k) \cdot f(d) + s \cdot \hat{f}(d)\right), \tag{143}$$

where $s \cdot \hat{f}(d)$ denotes the time to obtain $s$ priors, $(q - s) \cdot f(d)$ indicates the time for computing $\mathbf{v}_\perp$ in Eq. (14), and $(s+1) \cdot k \cdot f(d)$ represents the time of performing the binary search over $s$ priors and $\overline{\mathbf{v}_\perp}$ in Eq. (13). When $q$ is large, $s$ and $k$ are relatively small (i.e., the number of priors is small, and $k$ typically ranges in the tens), the additional overhead introduced by Prior-OPT is limited compared to Sign-OPT. While Prior-OPT introduces extra computation due to the binary search, the increase in time complexity is relatively modest, especially when $s$ remains much smaller than $q$. This shows that Prior-OPT strikes a balance between computational efficiency and gradient estimation quality.

Table 12 demonstrates the time consumption of Sign-OPT, SVM-OPT, Prior-Sign-OPT, and Prior-OPT, measured by performing untargeted attacks on the ImageNet dataset. We use a ResNet-50 surrogate model and an NVIDIA Tesla V100 GPU. The additional time overhead of Prior-Sign-OPT is mainly the time of obtaining priors on surrogate models. Prior-OPT uses Eq. (13) to estimate the gradient, invoking binary search $s + 1$ times, where $s$ is the number of surrogate models. This will result in additional time consumption compared to Prior-Sign-OPT. Note that for black-box attacks, the primary metrics are the number of queries and the attack success rate rather than runtime. In real-world scenarios, the number of queries is the main limitation, thus we need to use as few queries as possible to achieve the highest success rate. Table 13 shows the GPU memory allocations of Sign-OPT, Prior-Sign-OPT, and Prior-OPT. Prior-OPT and Prior-Sign-OPT require the transfer-based priors, and thus the additional memory allocation is mainly consumed in the forward and backward pass of the surrogate models. After obtaining a prior, GPU memory is promptly released, thus minimizing additional memory usage of our approach.

Table 12: The time consumption of attacking one image with 10,000 queries, which are measured by `seconds` on a NVIDIA Tesla V100 GPU.

| Method | ResNet-101 | SENet-154 | ResNeXt-101 | GC ViT | Swin Transformer |
|---|---|---|---|---|---|
| Sign-OPT (Cheng et al., 2020) | 112 | 197 | 91 | 131 | 88 |
| SVM-OPT (Cheng et al., 2020) | 119 | 189 | 102 | 158 | 98 |
| Prior-Sign-OPT$_{\text{ResNet50}}$ | 240 | 372 | 195 | 203 | 183 |
| Prior-OPT$_{\text{ResNet50}}$ | 342 | 476 | 321 | 357 | 203 |

Table 13: The GPU memory allocations of attacks against different target models, which are measured by `MiB` on a NVIDIA Tesla V100 GPU.

| Method | ResNet-101 | SENet-154 | ResNeXt-101 | GC ViT | Swin Transformer |
|---|---|---|---|---|---|
| Sign-OPT (Cheng et al., 2020) | 4,686 | 6,244 | 7,272 | 7,352 | 8,854 |
| SVM-OPT (Cheng et al., 2020) | 4,688 | 6,246 | 7,274 | 7,354 | 8,856 |
| Prior-Sign-OPT$_{\text{ResNet50}}$ | 5,222 | 6,750 | 7,828 | 7,856 | 9,410 |
| Prior-OPT$_{\text{ResNet50}}$ | 5,222 | 6,746 | 7,816 | 7,846 | 9,390 |

## G.2 EXPERIMENTAL RESULTS OF LARGE VISION-LANGUAGE MODEL

To evaluate the scalability of the proposed approach, we conduct experiments of attacking a CLIP model (Radford et al., 2021) with the ViT-L/14 backbone (Dosovitskiy et al., 2021), and the surrogate models include ImageNet pretrained ResNet-50, ConViT, CrossViT, MaxViT, and ViT. Here, ViT-L/14 refers to a large variant of the Vision Transformer architecture with the patch size of $14 \times 14$. It is worth noting that these surrogate models are pretrained on ImageNet and their training paradigms are entirely different from that of CLIP. The CLIP model, which stands for Contrastive Language-Image Pretraining, is trained on millions of image-text pairs from the internet using a contrastive learning approach, enabling it to generalize effectively through natural language supervision. By aligning images and text in a shared embedding space, the CLIP model functions as a zero-shot image classifier.

We encapsulate it as a 1,000-class classifier by constructing a set of text prompts that correspond to the class names. These prompts are then embedded into the same space as the images.

In the experiments, due to the differences between CLIP and standard classification models, the tested images used in previous experiments may not be correctly classified by CLIP. Therefore, we select a new set of 1,000 images that are correctly classified by both CLIP and five surrogate models (ResNet-50, ConViT, CrossViT, MaxViT, and ViT) for evaluation. The results are shown in Table 14, which demonstrate that incorporating more surrogate models (priors) significantly enhances attack performance. Notably, despite being pretrained on ImageNet with fundamentally different training methods than those used by CLIP, the surrogate models still improve performance.

Table 14: The experimental results of attacking against CLIP with the backbone of ViT-L/14, and the surrogate models include ImageNet pretrained ResNet-50, ConViT, CrossViT, MaxViT, and ViT.

| Method | Priors | Mean $\ell_2$ distortion | | | | | Attack Success Rate[1] | | | | |
|---|---|---|---|---|---|---|---|---|---|---|---|
| | | @1K | @2K | @5K | @8K | @10K | @1K | @2K | @5K | @8K | @10K |
| Sign-OPT (Cheng et al., 2020) | no prior | 58.180 | 49.435 | 40.261 | 37.020 | 35.713 | 13.4% | 15.2% | 17.4% | 19.1% | 19.4% |
| Prior-Sign-OPT$_{ResNet50}$ | 1 prior | 56.935 | 47.234 | 35.189 | 30.957 | 29.517 | 13.5% | 15.5% | 22.0% | 25.3% | 27.3% |
| Prior-Sign-OPT$_{ConViT}$ | 1 prior | 55.036 | 43.327 | 31.387 | 27.577 | 26.250 | 14.3% | 16.3% | 24.0% | 28.8% | 31.5% |
| Prior-Sign-OPT$_{ResNet50\&ConViT}$ | 2 priors | 53.658 | 40.868 | 27.988 | 23.954 | 22.737 | 14.1% | 17.6% | 28.0% | 34.3% | 37.4% |
| Prior-Sign-OPT$_{ResNet50\&ConViT\&CrossViT}$ | 3 priors | 50.875 | 36.428 | 23.410 | 19.589 | 18.414 | 15.3% | 20.5% | 37.2% | 44.1% | 46.7% |
| Prior-Sign-OPT$_{ResNet50\&ConViT\&CrossViT\&MaxViT}$ | 4 priors | 49.438 | 33.941 | 20.801 | 17.466 | 16.296 | **15.7%** | 23.3% | 44.3% | 53.2% | 57.1% |
| Prior-Sign-OPT$_{ResNet50\&ConViT\&CrossViT\&MaxViT\&ViT}$ | 5 priors | **48.214** | **32.298** | **19.114** | **15.790** | **14.726** | 15.0% | **24.1%** | **48.0%** | **56.5%** | **59.3%** |
| Prior-OPT$_{ResNet50}$ | 1 prior | 38.934 | 27.384 | 20.184 | 18.520 | 18.153 | 23.4% | 34.9% | 47.3% | 50.6% | 51.4% |
| Prior-OPT$_{ConViT}$ | 1 prior | 31.822 | 21.267 | 15.568 | 14.623 | 14.362 | 29.2% | 43.3% | 56.6% | 58.8% | 58.9% |
| Prior-OPT$_{ResNet50\&ConViT}$ | 2 priors | 29.596 | 18.088 | 11.770 | 10.724 | 10.427 | 31.3% | 50.1% | 68.3% | 72.2% | 73.4% |
| Prior-OPT$_{ResNet50\&ConViT\&CrossViT}$ | 3 priors | 26.355 | 15.251 | 9.953 | 8.834 | 8.625 | 35.0% | 55.7% | 75.2% | 79.1% | 79.5% |
| Prior-OPT$_{ResNet50\&ConViT\&CrossViT\&MaxViT}$ | 4 priors | 26.433 | 14.261 | 7.899 | 6.807 | 6.562 | 35.6% | 59.5% | 82.0% | 86.5% | 87.3% |
| Prior-OPT$_{ResNet50\&ConViT\&CrossViT\&MaxViT\&ViT}$ | 5 priors | **25.170** | **13.327** | **6.745** | **5.931** | **5.737** | **39.2%** | **63.0%** | **85.9%** | **89.4%** | **89.8%** |

[1] The distortion threshold for the attack success rate is 12.26898528811572, which is calculated as $\sqrt{0.001 \times 224 \times 224 \times 3}$.

### G.3 PERFORMANCE OF PRIOR-ONLY GRADIENT ESTIMATORS

The experimental results in previous sections demonstrate that incorporating a single transfer-based prior enhances performance. To explore this further, it is valuable to investigate an alternative approach where only prior vectors are used, rather than relying on random vectors. We can examine this approach from both theoretical and empirical perspectives.

If all random vectors are eliminated in gradient estimation, the gradient estimator's performance lacks a lower bound, making it unable to guarantee accuracy in the worst-case scenario. However, when random vectors are included in the gradient estimation, the accuracy of the estimator is guaranteed to have a lower bound. This means that, regardless of how poor the priors are, the estimator maintains a guaranteed minimum level of performance in the worst case. This can be verified by examining the $\mathbb{E}[\gamma]$ derived for Prior-Sign-OPT (Eq. (11)) and Prior-OPT (Eq. (15)). Specifically, when all random vectors are removed in Prior-Sign-OPT and $q$ is set to $s$, Eq. (11) reduces to

$$\mathbb{E}[\gamma] = \frac{1}{\sqrt{q}} \left( \sum_{i=1}^{s} |\alpha_i| \right). \tag{144}$$

In this case, $\mathbb{E}[\gamma]$ depends solely on $\alpha_i$, which reflects the accuracy of the priors. If $\alpha_i$ is extremely small, the accuracy of the estimated gradient degrades significantly. Similarly, when we remove all random vectors in Prior-OPT and $q$ is set to $s$, Eq. (15) reduces to

$$\mathbb{E}[\gamma] \geq \sqrt{\sum_{i=1}^{s} \alpha_i^2}. \tag{145}$$

This demonstrates that, without random vectors, the gradient estimation is entirely reliant on the quality of the priors (i.e., the $\alpha_i$ values), and poor priors can result in arbitrarily poor performance.

Conversely, when random vectors are included, the formula incorporating them guarantees a lower bound for $\mathbb{E}[\gamma]$. This lower bound can be derived by setting $\alpha_i = 0$ in Eqs. (11) and (15). For

Table 15: Mean $\ell_2$ distortions of targeted attacks on the ImageNet dataset against GC ViT, where Pure-Prior-Sign-OPT and Pure-Prior-OPT use only priors in the gradient estimation.

| Method | Priors | Targeted Attacks | | | | | | | | |
|---|---|---|---|---|---|---|---|---|---|---|
| | | @1K | @2K | @5K | @8K | @10K | @12K | @15K | @18K | @20K |
| Sign-OPT (Cheng et al., 2020) | no prior | 53.026 | 42.049 | 28.210 | 22.156 | 19.557 | 17.661 | 15.556 | 14.018 | 13.194 |
| Pure-Prior-Sign-OPT$_{\text{ResNet50}}$ | 1 prior | 52.337 | 51.255 | 51.024 | 51.017 | 51.017 | 51.017 | 51.017 | 51.017 | 51.017 |
| Pure-Prior-Sign-OPT$_{\text{ResNet50\&ConViT}}$ | 2 priors | 41.468 | 38.481 | 37.791 | 37.787 | 37.787 | 37.787 | 37.787 | 37.787 | 37.787 |
| Pure-Prior-OPT$_{\text{ResNet50}}$ | 1 prior | 53.673 | 53.416 | 53.385 | 53.385 | 53.385 | 53.385 | 53.385 | 53.385 | 53.385 |
| Pure-Prior-OPT$_{\text{ResNet50\&ConViT}}$ | 2 priors | 41.631 | 38.687 | 38.256 | 38.250 | 38.250 | 38.250 | 38.250 | 38.250 | 38.250 |
| Prior-Sign-OPT$_{\text{ResNet50}}$ (**ours**) | 1 prior | 52.491 | 41.333 | 26.857 | 20.829 | 18.427 | 16.681 | 14.741 | 13.337 | 12.593 |
| Prior-Sign-OPT$_{\text{ResNet50\&ConViT}}$ (**ours**) | 2 priors | 51.465 | 39.537 | 25.124 | 19.216 | 16.841 | **15.115** | **13.321** | **12.030** | **11.377** |
| Prior-OPT$_{\text{ResNet50}}$ (**ours**) | 1 prior | 50.323 | 39.615 | 25.876 | 20.309 | 18.120 | 16.488 | 14.750 | 13.535 | 12.918 |
| Prior-OPT$_{\text{ResNet50\&ConViT}}$ (**ours**) | 2 priors | 47.739 | **36.129** | **23.177** | **18.528** | **16.764** | 15.467 | 14.121 | 13.193 | 12.699 |

Table 16: Attack success rates of targeted attacks on the ImageNet dataset against GC ViT, where Pure-Prior-Sign-OPT and Pure-Prior-OPT use only priors in the gradient estimation.

| Method | Priors | Targeted Attacks | | | | | | | | |
|---|---|---|---|---|---|---|---|---|---|---|
| | | @1K | @2K | @5K | @8K | @10K | @12K | @15K | @18K | @20K |
| Sign-OPT (Cheng et al., 2020) | no prior | 1.0% | 1.9% | 8.5% | 20.3% | 30.2% | 38.7% | 48.8% | 57.5% | 61.2% |
| Pure-Prior-Sign-OPT$_{\text{ResNet50}}$ | 1 prior | 2.3% | 2.6% | 2.7% | 2.7% | 2.7% | 2.7% | 2.7% | 2.7% | 2.7% |
| Pure-Prior-Sign-OPT$_{\text{ResNet50\&ConViT}}$ | 2 priors | 8.6% | 10.7% | 11.6% | 11.6% | 11.6% | 11.6% | 11.6% | 11.6% | 11.6% |
| Pure-Prior-OPT$_{\text{ResNet50}}$ | 1 prior | 2.1% | 2.3% | 2.3% | 2.3% | 2.3% | 2.3% | 2.3% | 2.3% | 2.3% |
| Pure-Prior-OPT$_{\text{ResNet50\&ConViT}}$ | 2 priors | 9.7% | 12.5% | 12.7% | 12.7% | 12.7% | 12.7% | 12.7% | 12.7% | 12.7% |
| Prior-Sign-OPT$_{\text{ResNet50}}$ (**ours**) | 1 prior | 0.9% | 2.6% | 12.0% | 25.1% | 34.4% | 41.9% | 49.6% | 58.2% | 62.9% |
| Prior-Sign-OPT$_{\text{ResNet50\&ConViT}}$ (**ours**) | 2 priors | 0.9% | 3.3% | 15.8% | 32.2% | 40.2% | **47.9%** | **57.1%** | **62.5%** | **66.0%** |
| Prior-OPT$_{\text{ResNet50}}$ (**ours**) | 1 prior | 1.5% | 4.6% | 14.2% | 25.4% | 32.8% | 40.4% | 50.5% | 56.8% | 60.5% |
| Prior-OPT$_{\text{ResNet50\&ConViT}}$ (**ours**) | 2 priors | 2.6% | 9.2% | **24.7%** | **36.2%** | **41.0%** | 46.8% | 53.5% | 58.0% | 60.7% |

Prior-Sign-OPT, the lower bound of $\mathbb{E}[\gamma]$ is

$$\mathbb{E}[\gamma] \geq \frac{q-s}{\sqrt{q}} \cdot \frac{\Gamma(\frac{d-s}{2})}{\Gamma(\frac{d-s+1}{2})\sqrt{\pi}}. \tag{146}$$

For Prior-OPT, the lower bound of $\mathbb{E}[\gamma]$ is

$$\mathbb{E}[\gamma] \geq \sqrt{\frac{q-s}{\pi}} \cdot \frac{\Gamma(\frac{d-s}{2})}{\Gamma(\frac{d-s+1}{2})}, \tag{147}$$

thereby providing robustness with random vectors when the priors are of low quality. Furthermore, in Prior-OPT's gradient estimation, each prior requires a binary search, whereas random vectors do not. Random vectors require only a single query per vector, making them more efficient in this regard.

We present experimental results of targeted attacks using variants of the Prior-Sign-OPT and Prior-OPT algorithms, in which random vectors are excluded from the gradient estimation and only priors (i.e., gradients from surrogate models) are used. These variants are referred to as **Pure-Prior-Sign-OPT** and **Pure-Prior-OPT**. The experimental results of attacks against GC ViT (Hatamizadeh et al., 2023) on the ImageNet dataset are presented in Tables 15 and 16. The results indicate that Pure-Prior-Sign-OPT and Pure-Prior-OPT fail to outperform Sign-OPT when the query budget exceeds 2,000, even though Sign-OPT relies solely on random vectors without incorporating priors. Furthermore, as the query budget increases, the distortions and attack success rates for Pure-Prior-Sign-OPT and Pure-Prior-OPT remain relatively stable, revealing their inefficient use of additional queries.

### G.4 Effect of PGD Initialization on the Performance of SQBA and BBA Methods

Previous experiments have demonstrated that applying PGD (Projected Gradient Descent) initialization, denoted as Prior-OPT$_{\theta_0^{\text{PGD}}}$ and Prior-Sign-OPT$_{\theta_0^{\text{PGD}}}$, significantly enhances the performance of adversarial attacks. This raises the question: How would other baseline methods perform if PGD initialization were applied to them as well? To investigate this, we propose variants of the SQBA (Park et al., 2024) and BBA (Brunner et al., 2019) methods, labeled as SQBA$_{\theta_0^{\text{PGD}}}$ and BBA$_{\theta_0^{\text{PGD}}}$, in which PGD initialization is utilized on a surrogate model to generate the initial adversarial examples.

Table 17: Mean $\ell_2$ distortions of untargeted attacks on the ImageNet dataset against Inception-v4.

| Method | Untargeted Attacks | | | | | | | | | |
|---|---|---|---|---|---|---|---|---|---|---|
| | @1K | @2K | @3K | @4K | @5K | @6K | @7K | @8K | @9K | @10K |
| SQBA$_{\text{IncResV2}}$ (Park et al., 2024) | 26.134 | 19.035 | 15.200 | 12.799 | 11.189 | 10.015 | 9.129 | 8.432 | 7.878 | 7.417 |
| SQBA$_{\theta_0^{\text{PGD}} + \text{IncResV2}}$ (Park et al., 2024) | **22.698** | 16.882 | 13.731 | 11.699 | 10.314 | 9.301 | 8.543 | 7.931 | 7.426 | 7.017 |
| BBA$_{\text{IncResV2}}$ (Brunner et al., 2019) | 38.782 | 28.437 | 23.673 | 20.745 | 18.757 | 17.373 | 16.307 | 15.474 | 14.781 | 14.191 |
| BBA$_{\theta_0^{\text{PGD}} + \text{IncResV2}}$ (Brunner et al., 2019) | 26.297 | 20.370 | 17.460 | 15.647 | 14.404 | 13.484 | 12.754 | 12.177 | 11.700 | 11.295 |
| Prior-Sign-OPT$_{\text{IncResV2}}$ | 81.991 | 42.403 | 25.355 | 17.163 | 12.835 | 10.191 | 8.508 | 7.365 | 6.508 | 5.842 |
| Prior-Sign-OPT$_{\theta_0^{\text{PGD}} + \text{IncResV2}}$ | 23.596 | 15.347 | 11.565 | 9.458 | 8.074 | 7.085 | 6.330 | 5.729 | 5.249 | 4.863 |
| Prior-OPT$_{\text{IncResV2}}$ | 49.279 | 18.135 | 9.426 | **6.798** | **5.718** | **5.148** | **4.747** | **4.451** | **4.215** | **4.027** |
| Prior-OPT$_{\theta_0^{\text{PGD}} + \text{IncResV2}}$ | 22.852 | **12.194** | **8.896** | 7.452 | 6.568 | 5.947 | 5.485 | 5.114 | 4.809 | 4.548 |

Table 18: Mean $\ell_2$ distortions of untargeted attacks on the ImageNet dataset against ViT.

| Method | Untargeted Attacks | | | | | | | | | |
|---|---|---|---|---|---|---|---|---|---|---|
| | @1K | @2K | @3K | @4K | @5K | @6K | @7K | @8K | @9K | @10K |
| SQBA$_{\text{ConViT}}$ (Park et al., 2024) | 12.886 | 9.762 | 8.045 | 6.972 | 6.240 | 5.702 | 5.278 | 4.947 | 4.681 | 4.452 |
| SQBA$_{\theta_0^{\text{PGD}} + \text{ConViT}}$ (Park et al., 2024) | 10.794 | 8.424 | 7.094 | 6.227 | 5.647 | 5.204 | 4.856 | 4.572 | 4.337 | 4.143 |
| BBA$_{\text{ConViT}}$ (Brunner et al., 2019) | 22.716 | 16.153 | 13.409 | 11.886 | 10.893 | 10.155 | 9.614 | 9.193 | 8.868 | 8.595 |
| BBA$_{\theta_0^{\text{PGD}} + \text{ConViT}}$ (Brunner et al., 2019) | 11.163 | 9.431 | 8.535 | 7.958 | 7.534 | 7.227 | 6.982 | 6.783 | 6.615 | 6.477 |
| Prior-Sign-OPT$_{\text{ConViT}}$ | 46.883 | 24.551 | 14.592 | 10.329 | 8.057 | 6.669 | 5.755 | 5.142 | 4.688 | 4.313 |
| Prior-Sign-OPT$_{\theta_0^{\text{PGD}} + \text{ConViT}}$ | 9.011 | 6.935 | 5.752 | 5.025 | **4.504** | **4.108** | **3.803** | **3.549** | **3.345** | **3.174** |
| Prior-OPT$_{\text{ConViT}}$ | 26.649 | 11.706 | 7.632 | 6.025 | 5.228 | 4.728 | 4.380 | 4.117 | 3.909 | 3.754 |
| Prior-OPT$_{\theta_0^{\text{PGD}} + \text{ConViT}}$ | **8.688** | **6.646** | **5.595** | **4.962** | 4.551 | 4.245 | 4.003 | 3.808 | 3.640 | 3.511 |

Table 19: Mean $\ell_2$ distortions of untargeted attacks on the ImageNet dataset against GC ViT.

| Method | Untargeted Attacks | | | | | | | | | |
|---|---|---|---|---|---|---|---|---|---|---|
| | @1K | @2K | @3K | @4K | @5K | @6K | @7K | @8K | @9K | @10K |
| SQBA$_{\text{ConViT}}$ (Park et al., 2024) | 19.307 | 14.049 | 11.170 | 9.327 | 8.072 | 7.135 | 6.434 | 5.877 | 5.426 | 5.056 |
| SQBA$_{\theta_0^{\text{PGD}} + \text{ConViT}}$ (Park et al., 2024) | **15.652** | 11.520 | 9.197 | 7.752 | 6.754 | 6.033 | 5.479 | 5.034 | 4.673 | 4.370 |
| BBA$_{\text{ConViT}}$ (Brunner et al., 2019) | 29.928 | 21.095 | 17.061 | 14.680 | 13.103 | 11.954 | 11.020 | 10.302 | 9.694 | 9.188 |
| BBA$_{\theta_0^{\text{PGD}} + \text{ConViT}}$ (Brunner et al., 2019) | 15.959 | 12.688 | 11.054 | 9.997 | 9.230 | 8.627 | 8.164 | 7.766 | 7.430 | 7.131 |
| Prior-Sign-OPT$_{\text{ConViT}}$ | 55.864 | 34.707 | 22.793 | 16.584 | 12.893 | 10.546 | 8.895 | 7.678 | 6.712 | 5.972 |
| Prior-Sign-OPT$_{\theta_0^{\text{PGD}} + \text{ConViT}}$ | 17.159 | 11.230 | 8.642 | 7.209 | 6.250 | 5.551 | 5.009 | 4.560 | **4.205** | 3.916 |
| Prior-OPT$_{\text{ConViT}}$ | 39.497 | 18.955 | 12.320 | 9.275 | 7.641 | 6.599 | 5.828 | 5.251 | 4.817 | 4.453 |
| Prior-OPT$_{\theta_0^{\text{PGD}} + \text{ConViT}}$ | 16.949 | **10.708** | **8.251** | **6.937** | **6.031** | **5.391** | **4.913** | **4.530** | 4.219 | **3.961** |

Table 20: Mean $\ell_2$ distortions of untargeted attacks on the ImageNet dataset against ResNet-101.

| Method | Untargeted Attacks | | | | | | | | | |
|---|---|---|---|---|---|---|---|---|---|---|
| | @1K | @2K | @3K | @4K | @5K | @6K | @7K | @8K | @9K | @10K |
| SQBA$_{\text{ResNet50}}$ (Park et al., 2024) | 8.873 | 7.229 | 6.172 | 5.449 | 4.934 | 4.531 | 4.215 | 3.957 | 3.745 | 3.563 |
| SQBA$_{\theta_0^{\text{PGD}} + \text{ResNet50}}$ (Park et al., 2024) | 6.882 | 5.675 | 4.894 | 4.364 | 3.985 | 3.689 | 3.456 | 3.264 | 3.101 | 2.961 |
| BBA$_{\text{ResNet50}}$ (Brunner et al., 2019) | 14.935 | 11.764 | 10.346 | 9.484 | 8.870 | 8.421 | 8.051 | 7.754 | 7.511 | 7.295 |
| BBA$_{\theta_0^{\text{PGD}} + \text{ResNet50}}$ (Brunner et al., 2019) | 6.281 | 5.488 | 5.051 | 4.779 | 4.577 | 4.425 | 4.302 | 4.196 | 4.109 | 4.029 |
| Prior-Sign-OPT$_{\text{ResNet50}}$ | 34.150 | 18.733 | 11.452 | 7.977 | 6.111 | 4.982 | 4.247 | 3.718 | 3.323 | 3.019 |
| Prior-Sign-OPT$_{\theta_0^{\text{PGD}} + \text{ResNet50}}$ | 5.423 | 4.303 | 3.632 | 3.182 | 2.859 | 2.615 | 2.414 | **2.267** | **2.142** | **2.045** |
| Prior-OPT$_{\text{ResNet50}}$ | 18.355 | 7.100 | 4.190 | 3.214 | 2.840 | 2.612 | 2.450 | 2.324 | 2.233 | 2.158 |
| Prior-OPT$_{\theta_0^{\text{PGD}} + \text{ResNet50}}$ | **4.932** | **3.807** | **3.273** | **2.940** | **2.710** | **2.532** | **2.390** | 2.275 | 2.181 | 2.107 |

Table 21: Mean $\ell_2$ distortions of untargeted attacks on the ImageNet dataset against SENet-154.

| Method | Untargeted Attacks | | | | | | | | | |
|---|---|---|---|---|---|---|---|---|---|---|
| | @1K | @2K | @3K | @4K | @5K | @6K | @7K | @8K | @9K | @10K |
| SQBA$_{\text{ResNet50}}$ (Park et al., 2024) | 16.332 | 11.802 | 9.335 | 7.788 | 6.765 | 6.016 | 5.445 | 4.994 | 4.630 | 4.332 |
| SQBA$_{\theta_0^{\text{PGD}} + \text{ResNet50}}$ (Park et al., 2024) | 13.342 | 9.871 | 7.944 | 6.707 | 5.863 | 5.246 | 4.779 | 4.410 | 4.115 | 3.860 |
| BBA$_{\text{ResNet50}}$ (Brunner et al., 2019) | 24.402 | 17.863 | 14.923 | 13.134 | 11.915 | 11.009 | 10.330 | 9.796 | 9.348 | 8.976 |
| BBA$_{\theta_0^{\text{PGD}} + \text{ResNet50}}$ (Brunner et al., 2019) | 13.074 | 10.435 | 9.080 | 8.221 | 7.626 | 7.187 | 6.843 | 6.559 | 6.320 | 6.112 |
| Prior-Sign-OPT$_{\text{ResNet50}}$ | 45.340 | 26.404 | 17.200 | 12.317 | 9.412 | 7.551 | 6.285 | 5.400 | 4.740 | 4.223 |
| Prior-Sign-OPT$_{\theta_0^{\text{PGD}} + \text{ResNet50}}$ | 12.375 | 8.859 | 6.900 | 5.684 | 4.865 | **4.272** | **3.817** | **3.461** | **3.184** | **2.958** |
| Prior-OPT$_{\text{ResNet50}}$ | 29.578 | 14.233 | 8.955 | 6.677 | 5.542 | 4.823 | 4.316 | 3.947 | 3.630 | 3.394 |
| Prior-OPT$_{\theta_0^{\text{PGD}} + \text{ResNet50}}$ | **11.952** | **8.431** | **6.580** | **5.542** | **4.863** | 4.368 | 3.980 | 3.680 | 3.420 | 3.215 |

Table 22: Success rates of untargeted attacks on ImageNet against Inception-v4.

| Method | Untargeted Attacks | | | | | | | | | |
|---|---|---|---|---|---|---|---|---|---|---|
| | @1K | @2K | @3K | @4K | @5K | @6K | @7K | @8K | @9K | @10K |
| SQBA$_{IncResV2}$ (Park et al., 2024) | 41.9% | 55.8% | 65.9% | 74.7% | 79.2% | 85.6% | 87.7% | 89.9% | 91.4% | 93.0% |
| SQBA$_{\theta_0^{PGD} + IncResV2}$ (Park et al., 2024) | 51.5% | 63.7% | 71.3% | 79.4% | 82.8% | 87.4% | 90.2% | 91.2% | 92.6% | 93.3% |
| BBA$_{IncResV2}$ (Brunner et al., 2019) | 16.3% | 29.0% | 39.0% | 46.9% | 54.3% | 57.5% | 60.6% | 63.9% | 66.6% | 68.8% |
| BBA$_{\theta_0^{PGD} + IncResV2}$ (Brunner et al., 2019) | 43.9% | 53.4% | 60.1% | 65.6% | 69.0% | 71.7% | 74.8% | 77.0% | 78.8% | 79.6% |
| Prior-Sign-OPT$_{IncResV2}$ | 3.8% | 16.6% | 38.9% | 60.4% | 75.7% | 85.0% | 89.4% | 91.8% | 94.2% | 96.3% |
| Prior-Sign-OPT$_{\theta_0^{PGD} + IncResV2}$ | 62.6% | 72.2% | 77.8% | 84.4% | 88.3% | 90.7% | 93.4% | 94.5% | 95.8% | 96.9% |
| Prior-OPT$_{IncResV2}$ | 17.8% | 63.4% | **86.6%** | 94.2% | **96.4%** | **97.4%** | **98.4%** | **98.8%** | **99.0%** | **99.1%** |
| Prior-OPT$_{\theta_0^{PGD} + IncResV2}$ | **64.7%** | **76.6%** | 84.1% | 89.2% | 92.5% | 95.1% | 96.1% | 96.7% | 97.6% | 98.1% |

Table 23: Success rates of untargeted attacks on ImageNet against ViT.

| Method | Untargeted Attacks | | | | | | | | | |
|---|---|---|---|---|---|---|---|---|---|---|
| | @1K | @2K | @3K | @4K | @5K | @6K | @7K | @8K | @9K | @10K |
| SQBA$_{ConViT}$ (Park et al., 2024) | 55.9% | 71.7% | 81.1% | 88.4% | 91.9% | 94.4% | 96.7% | 97.4% | 97.8% | 98.3% |
| SQBA$_{\theta_0^{PGD} + ConViT}$ (Park et al., 2024) | 67.1% | 80.7% | 87.8% | 91.6% | 94.6% | 96.2% | 96.9% | 97.6% | 98.2% | 98.3% |
| BBA$_{ConViT}$ (Brunner et al., 2019) | 19.0% | 39.6% | 53.0% | 61.6% | 66.2% | 70.2% | 73.9% | 75.2% | 77.7% | 79.3% |
| BBA$_{\theta_0^{PGD} + ConViT}$ (Brunner et al., 2019) | 69.9% | 76.4% | 80.3% | 84.8% | 86.5% | 87.5% | 89.1% | 89.5% | 90.6% | 91.3% |
| Prior-Sign-OPT$_{ConViT}$ | 8.4% | 23.8% | 49.2% | 71.4% | 83.6% | 90.4% | 94.0% | 96.0% | 97.1% | 98.1% |
| Prior-Sign-OPT$_{\theta_0^{PGD} + ConViT}$ | **83.8%** | **89.7%** | **93.6%** | **95.2%** | **96.4%** | **97.5%** | **97.9%** | **98.3%** | 98.7% | **99.2%** |
| Prior-OPT$_{ConViT}$ | 26.2% | 65.7% | 83.6% | 91.3% | 94.7% | 96.2% | 97.1% | 97.7% | **98.9%** | **99.2%** |
| Prior-OPT$_{\theta_0^{PGD} + ConViT}$ | 83.6% | **89.7%** | 93.4% | 94.9% | 96.3% | 97.3% | **97.9%** | 98.1% | 98.6% | 98.9% |

Table 24: Success rates of untargeted attacks on ImageNet against GC ViT.

| Method | Untargeted Attacks | | | | | | | | | |
|---|---|---|---|---|---|---|---|---|---|---|
| | @1K | @2K | @3K | @4K | @5K | @6K | @7K | @8K | @9K | @10K |
| SQBA$_{ConViT}$ (Park et al., 2024) | 38.1% | 53.3% | 64.4% | 73.8% | 79.6% | 84.7% | 87.7% | 90.9% | 93.1% | 94.4% |
| SQBA$_{\theta_0^{PGD} + ConViT}$ (Park et al., 2024) | 50.2% | 65.5% | 74.6% | 81.9% | 86.8% | 89.7% | 92.1% | 94.1% | 95.4% | 96.2% |
| BBA$_{ConViT}$ (Brunner et al., 2019) | 10.7% | 28.2% | 42.4% | 50.4% | 56.6% | 62.9% | 67.8% | 71.5% | 74.0% | 77.6% |
| BBA$_{\theta_0^{PGD} + ConViT}$ (Brunner et al., 2019) | 51.7% | 60.3% | 66.0% | 69.4% | 74.2% | 78.1% | 79.7% | 82.7% | 85.1% | 86.5% |
| Prior-Sign-OPT$_{ConViT}$ | 1.7% | 9.2% | 27.5% | 48.2% | 62.7% | 72.0% | 78.7% | 82.7% | 86.3% | 89.0% |
| Prior-Sign-OPT$_{\theta_0^{PGD} + ConViT}$ | **63.4%** | 73.6% | 81.1% | 85.4% | 89.4% | 91.2% | 92.8% | 94.3% | 95.4% | 96.4% |
| Prior-OPT$_{ConViT}$ | 10.0% | 45.3% | 68.1% | 82.2% | 87.9% | 90.1% | 92.4% | 93.5% | 94.8% | 95.7% |
| Prior-OPT$_{\theta_0^{PGD} + ConViT}$ | 63.2% | **74.1%** | **81.4%** | **86.4%** | **89.9%** | **92.0%** | **93.5%** | **94.4%** | **95.7%** | **96.7%** |

Table 25: Success rates of untargeted attacks on ImageNet against ResNet-101.

| Method | Untargeted Attacks | | | | | | | | | |
|---|---|---|---|---|---|---|---|---|---|---|
| | @1K | @2K | @3K | @4K | @5K | @6K | @7K | @8K | @9K | @10K |
| SQBA$_{ResNet50}$ (Park et al., 2024) | 78.7% | 85.3% | 89.6% | 92.6% | 94.6% | 96.2% | 97.0% | 98.1% | 98.3% | 98.9% |
| SQBA$_{\theta_0^{PGD} + ResNet50}$ (Park et al., 2024) | 88.5% | 93.2% | 95.2% | 96.0% | 97.1% | 97.5% | 98.1% | 98.7% | 98.9% | 99.3% |
| BBA$_{ResNet50}$ (Brunner et al., 2019) | 47.6% | 62.3% | 70.2% | 75.2% | 79.4% | 81.1% | 83.7% | 84.6% | 85.7% | 86.9% |
| BBA$_{\theta_0^{PGD} + ResNet50}$ (Brunner et al., 2019) | 89.8% | 92.2% | 93.9% | 95.1% | 95.9% | 96.0% | 96.4% | 96.8% | 97.1% | 97.1% |
| Prior-Sign-OPT$_{ResNet50}$ | 10.9% | 35.4% | 65.0% | 83.0% | 91.1% | 94.0% | 95.9% | 97.3% | 98.6% | 99.4% |
| Prior-Sign-OPT$_{\theta_0^{PGD} + ResNet50}$ | 94.5% | **96.1%** | **97.2%** | 97.9% | 98.2% | 98.7% | 99.0% | 99.3% | 99.4% | 99.7% |
| Prior-OPT$_{ResNet50}$ | 42.0% | 87.4% | 96.2% | **98.3%** | **99.2%** | **99.3%** | **99.7%** | **99.7%** | **99.8%** | **100.0%** |
| Prior-OPT$_{\theta_0^{PGD} + ResNet50}$ | **94.6%** | 95.9% | 96.9% | 97.8% | 98.3% | 98.7% | 98.7% | 99.0% | 99.2% | 99.5% |

Table 26: Success rates of untargeted attacks on ImageNet against SENet-154.

| Method | Untargeted Attacks | | | | | | | | | |
|---|---|---|---|---|---|---|---|---|---|---|
| | @1K | @2K | @3K | @4K | @5K | @6K | @7K | @8K | @9K | @10K |
| SQBA$_{ResNet50}$ (Park et al., 2024) | 45.1% | 61.5% | 72.9% | 81.7% | 85.7% | 89.0% | 91.7% | 94.6% | 96.4% | 97.9% |
| SQBA$_{\theta_0^{PGD} + ResNet50}$ (Park et al., 2024) | 59.1% | 72.3% | 81.1% | 86.7% | 90.2% | 92.4% | 94.4% | 96.5% | 97.5% | 98.4% |
| BBA$_{ResNet50}$ (Brunner et al., 2019) | 16.6% | 33.9% | 45.6% | 54.9% | 61.2% | 66.3% | 69.7% | 72.7% | 74.2% | 76.9% |
| BBA$_{\theta_0^{PGD} + ResNet50}$ (Brunner et al., 2019) | 60.8% | 69.7% | 75.4% | 79.9% | 83.2% | 85.6% | 87.0% | 88.6% | 89.9% | 90.6% |
| Prior-Sign-OPT$_{ResNet50}$ | 3.7% | 16.2% | 36.6% | 58.3% | 75.2% | 83.1% | 88.3% | 92.1% | 95.0% | 96.7% |
| Prior-Sign-OPT$_{\theta_0^{PGD} + ResNet50}$ | 71.9% | 79.6% | 87.2% | 91.3% | 94.2% | **96.3%** | **97.2%** | **97.6%** | 97.9% | 98.3% |
| Prior-OPT$_{ResNet50}$ | 16.3% | 55.9% | 77.5% | 88.3% | 93.0% | 95.4% | 97.0% | **97.6%** | **98.6%** | **98.9%** |
| Prior-OPT$_{\theta_0^{PGD} + ResNet50}$ | **72.5%** | **81.3%** | **88.5%** | **91.6%** | **94.3%** | 96.2% | 97.1% | 97.5% | 98.0% | 98.6% |

The mean $\ell_2$ distortions of the experimental results on the ImageNet dataset are presented in Tables 17, 18, 19, 20, and 21, while the corresponding attack success rates are shown in Tables 22, 23, 24, 25, and 26. The distortion threshold for attack success rates is 16.3769 for attacks on Inception-v4 and 12.2689 for attacks on other networks, calculated as $\sqrt{0.001 \times d}$, where $d$ is the dimension of the input image. As shown in these tables, the PGD initialization improves the performance of both SQBA and BBA, resulting in reduced mean $\ell_2$ distortions and higher attack success rates. Furthermore, our approach with PGD initialization outperforms both SQBA and BBA.

## G.5 COMPARISON WITH STATE-OF-THE-ART METHODS

Table 27: Untargeted attack results of ViTs on the ImageNet dataset, where "Mean $\ell_2$" denotes the average $\ell_2$ distortion of the final adversarial examples, "AUC" denotes the area under the curve of mean $\ell_2$ distortions versus the number of queries (lower is better), and "ASR" denotes the attack success rate of the final adversarial examples.

| Method | ViT | | | GC ViT | | | Swin Transformer | | |
|---|---|---|---|---|---|---|---|---|---|
| | Mean $\ell_2$ | AUC | ASR | Mean $\ell_2$ | AUC | ASR | Mean $\ell_2$ | AUC | ASR |
| HSJA (Chen et al., 2020) | 5.637 | 102956.7 | 96.7% | 7.955 | 163915.1 | 82.8% | 10.635 | 228806.0 | 70.3% |
| TA (Ma et al., 2021b) | 5.674 | 104023.3 | 96.6% | 9.102 | 176063.8 | 76.7% | 10.513 | 230351.0 | 68.4% |
| G-TA (Ma et al., 2021b) | 5.643 | 102013.4 | 96.4% | 8.671 | 170511.6 | 77.6% | 9.929 | 219877.9 | 72.6% |
| GeoDA (Rahmati et al., 2020) | 6.313 | 83176.7 | 91.0% | 12.998 | 172173.2 | 54.3% | 19.120 | 245094.6 | 31.5% |
| Evolutionary (Dong et al., 2019) | 6.719 | 128659.9 | 89.8% | 8.615 | 174592.0 | 79.1% | 15.738 | 266695.9 | 52.6% |
| SurFree (Maho et al., 2021) | 6.303 | 104053.9 | 91.6% | 10.967 | 193400.6 | 65.4% | 13.059 | 200688.5 | 58.3% |
| Triangle Attack (Wang et al., 2022) | 10.097 | 99746.4 | 69.0% | 30.119 | 298578.9 | 21.2% | 29.005 | 288358.2 | 23.1% |
| BBA$_{\text{ResNet50}}$ (Brunner et al., 2019) | 9.567 | 125221.1 | 74.0% | 11.294 | 161711.4 | 67.5% | 14.084 | 185551.7 | 59.9% |
| BBA$_{\text{ConViT}}$ (Brunner et al., 2019) | 8.595 | 105826.6 | 79.3% | 9.188 | 128468.5 | 77.6% | 12.375 | 156081.9 | 59.5% |
| SQBA$_{\text{ResNet50}}$ (Park et al., 2024) | 5.201 | 79423.0 | 95.7% | 6.186 | 100435.8 | 89.2% | 7.557 | 115845.7 | 83.2% |
| SQBA$_{\text{ConViT}}$ (Park et al., 2024) | 4.452 | **60295.8** | 98.3% | 5.056 | 79670.8 | 94.4% | 5.883 | 82141.3 | 91.0% |
| Sign-OPT (Cheng et al., 2020) | 4.572 | 111439.9 | 98.3% | 7.185 | 166001.9 | 85.9% | 9.899 | 238907.0 | 74.7% |
| SVM-OPT (Cheng et al., 2020) | 5.070 | 120008.9 | 97.1% | 7.325 | 171869.8 | 83.9% | 10.526 | 249491.3 | 72.1% |
| Prior-Sign-OPT$_{\text{ResNet50}}$ | 4.850 | 119961.6 | 97.4% | 6.723 | 165586.5 | 87.1% | 9.254 | 234462.4 | 75.7% |
| Prior-Sign-OPT$_{\text{ConViT}}$ | 4.313 | 105379.9 | 98.1% | 5.972 | 151725.9 | 89.0% | 7.622 | 198431.3 | 84.2% |
| Prior-Sign-OPT$_{\text{ResNet50\&ConViT}}$ | 3.967 | 99940.6 | 98.6% | 5.286 | 137011.9 | 92.9% | 6.331 | 177589.1 | 89.2% |
| Prior-Sign-OPT$_{\theta_0^{\text{PGD}} + \text{ResNet50}}$ | 4.331 | 88120.1 | 97.7% | 5.243 | 98749.0 | 92.6% | 8.112 | 128167.1 | 80.3% |
| Prior-OPT$_{\text{ResNet50}}$ | 5.195 | 106791.0 | 97.3% | 6.066 | 134255.9 | 90.7% | 9.625 | 190534.1 | 73.0% |
| Prior-OPT$_{\text{ConViT}}$ | 3.754 | 62928.1 | **99.2%** | 4.453 | 92662.3 | 95.7% | 5.558 | 102428.7 | 91.8% |
| Prior-OPT$_{\text{ResNet50\&ConViT}}$ | **3.609** | 60449.0 | **99.2%** | **3.700** | 76896.1 | 98.3% | 4.896 | 91211.7 | 94.5% |
| Prior-OPT$_{\theta_0^{\text{PGD}} + \text{ResNet50}}$ | 5.009 | 90005.1 | 96.4% | 5.502 | 98555.8 | 92.8% | 8.552 | 128766.0 | 76.4% |

Table 28: Untargeted attack results of CNNs on the ImageNet dataset, where "Mean $\ell_2$" denotes the average $\ell_2$ distortion of the final adversarial examples, "AUC" denotes the area under the curve of mean $\ell_2$ distortions versus the number of queries (lower is better), and "ASR" denotes the attack success rate of the final adversarial examples.

| Method | ResNet-101 | | | ResNeXt-101 ($64 \times 4$d) | | | SENet-154 | | |
|---|---|---|---|---|---|---|---|---|---|
| | Mean $\ell_2$ | AUC | ASR | Mean $\ell_2$ | AUC | ASR | Mean $\ell_2$ | AUC | ASR |
| HSJA (Chen et al., 2020) | 5.158 | 96234.2 | 95.8% | 5.484 | 110376.8 | 95.0% | 9.385 | 177364.9 | 74.9% |
| TA (Ma et al., 2021b) | 5.239 | 96858.5 | 95.9% | 5.565 | 110870.1 | 95.0% | 9.379 | 172600.0 | 73.8% |
| G-TA (Ma et al., 2021b) | 5.225 | 95901.1 | 96.3% | 5.524 | 109990.5 | 95.0% | 5.430 | 119281.1 | 92.9% |
| GeoDA (Rahmati et al., 2020) | 6.364 | 82320.0 | 91.9% | 6.898 | 88947.7 | 89.3% | 8.209 | 107267.4 | 80.9% |
| Evolutionary (Dong et al., 2019) | 5.406 | 107841.6 | 93.2% | 6.042 | 123706.5 | 91.3% | 6.111 | 130032.0 | 90.1% |
| SurFree (Maho et al., 2021) | 6.627 | 104285.4 | 88.1% | 7.550 | 123394.0 | 83.7% | 8.247 | 131295.4 | 79.5% |
| Triangle Attack (Wang et al., 2022) | 12.123 | 117731.5 | 61.3% | 11.883 | 116639.5 | 63.7% | 15.019 | 145508.7 | 48.9% |
| BBA$_{\text{ResNet50}}$ (Brunner et al., 2019) | 7.295 | 83314.7 | 86.9% | 9.393 | 116579.5 | 74.9% | 8.976 | 115007.9 | 76.9% |
| SQBA$_{\text{ResNet50}}$ (Park et al., 2024) | 3.563 | 46450.8 | 98.9% | 4.058 | 59316.7 | 97.8% | 4.332 | 67106.3 | 97.9% |
| Sign-OPT (Cheng et al., 2020) | 4.754 | 101907.7 | 95.9% | 5.108 | 120545.5 | 95.4% | 5.111 | 124730.7 | 93.5% |
| SVM-OPT (Cheng et al., 2020) | 4.842 | 105778.8 | 95.8% | 5.255 | 126799.4 | 95.0% | 5.125 | 127568.9 | 93.7% |
| Prior-Sign-OPT$_{\text{ResNet50}}$ | 3.019 | 79126.4 | 99.4% | 3.518 | 100999.4 | 98.9% | 4.223 | 114089.3 | 96.7% |
| Prior-Sign-OPT$_{\theta_0^{\text{PGD}} + \text{ResNet50}}$ | **2.045** | 27148.4 | 99.7% | **2.450** | 35290.7 | 99.4% | **2.958** | 48708.7 | 98.3% |
| Prior-OPT$_{\text{ResNet50}}$ | 2.158 | 37218.3 | **100.0%** | 2.692 | 53085.7 | 99.7% | 3.394 | 68609.1 | **98.9%** |
| Prior-OPT$_{\theta_0^{\text{PGD}} + \text{ResNet50}}$ | 2.107 | **25627.6** | 99.5% | 2.486 | **33291.5** | 99.7% | 3.215 | **48447.6** | 98.6% |

Tables 27, 28, and 29 show the experimental results of untargeted attacks against ViTs and CNNs on the ImageNet dataset, where "AUC" indicates area under the curve of the mean $\ell_2$ distortions versus the number of queries, "Mean $\ell_2$" denotes the average $\ell_2$ distortion of the final adversarial examples, and "ASR" indicates the attack success rate of the final adversarial examples. Here, the final adversarial examples are generated with the query budget of 10,000. The ASR is defined as the percentage of samples with distortions below a threshold $\epsilon$, which is set to $\epsilon = \sqrt{0.001 \times d}$ in the

Table 29: Untargeted attack results of Inception networks on the ImageNet dataset, where "Mean $\ell_2$" denotes the average $\ell_2$ distortion of the final adversarial examples, "AUC" denotes the area under the curve of mean $\ell_2$ distortions versus the number of queries (lower is better), and "ASR" denotes the attack success rate of the final adversarial examples.

| Method | Inception-V3 | | | Inception-V4 | | |
|---|---|---|---|---|---|---|
| | Mean $\ell_2$ | AUC | ASR | Mean $\ell_2$ | AUC | ASR |
| HSJA (Chen et al., 2020) | 12.014 | 211938.7 | 81.1% | 11.645 | 227700.5 | 82.1% |
| TA (Ma et al., 2021b) | 12.378 | 208706.8 | 79.8% | 11.694 | 219707.3 | 82.0% |
| G-TA (Ma et al., 2021b) | 12.076 | 205670.7 | 81.5% | 11.448 | 216797.7 | 83.3% |
| GeoDA (Rahmati et al., 2020) | 9.437 | 124150.7 | 87.8% | 9.688 | 128665.4 | 87.7% |
| Evolutionary (Dong et al., 2019) | 9.809 | 192654.1 | 86.4% | 10.839 | 215405.4 | 81.6% |
| SurFree (Maho et al., 2021) | 11.648 | 186094.3 | 79.1% | 13.818 | 221197.5 | 69.7% |
| Triangle Attack (Wang et al., 2022) | 20.878 | 205534.2 | 46.5% | 22.132 | 214723.3 | 42.5% |
| BBA$_{\text{IncResV2}}$ (Brunner et al., 2019) | 13.952 | 169881.3 | 69.0% | 14.191 | 182033.7 | 68.8% |
| BBA$_{\text{Xception}}$ (Brunner et al., 2019) | 14.657 | 185798.3 | 67.2% | 15.282 | 199287.2 | 63.6% |
| SQBA$_{\text{IncResV2}}$ (Park et al., 2024) | 7.020 | 98767.0 | 94.3% | 7.417 | 110451.8 | 93.0% |
| SQBA$_{\text{Xception}}$ (Park et al., 2024) | 6.933 | 97022.6 | 94.7% | 7.115 | 102939.0 | 92.2% |
| Sign-OPT (Cheng et al., 2020) | 8.134 | 195118.7 | 91.9% | 8.786 | 217576.3 | 89.8% |
| SVM-OPT (Cheng et al., 2020) | 7.995 | 193289.7 | 92.3% | 8.839 | 219673.6 | 89.0% |
| Prior-Sign-OPT$_{\text{IncResV2}}$ | 5.314 | 156261.6 | 97.8% | 5.842 | 174243.9 | 96.3% |
| Prior-Sign-OPT$_{\text{Xception}}$ | 5.831 | 163715.5 | 96.8% | 5.958 | 176950.5 | 95.7% |
| Prior-Sign-OPT$_{\text{IncResV2\&Xception}}$ | 4.225 | 130427.5 | 98.6% | 4.199 | 142032.3 | 99.0% |
| Prior-Sign-OPT$_{\theta_0^{\text{PGD}} + \text{IncResV2}}$ | 4.713 | 75088.7 | 97.1% | 4.863 | 83066.8 | 96.9% |
| Prior-OPT$_{\text{IncResV2}}$ | 4.067 | 79685.9 | 99.3% | 4.027 | 85290.8 | 99.1% |
| Prior-OPT$_{\text{Xception}}$ | 4.539 | 88915.0 | 99.3% | 4.261 | 90492.0 | 99.3% |
| Prior-OPT$_{\text{IncResV2\&Xception}}$ | **3.387** | **64461.8** | **99.7%** | **3.167** | **66110.1** | **99.8%** |
| Prior-OPT$_{\theta_0^{\text{PGD}} + \text{IncResV2}}$ | 4.496 | 65031.0 | 98.4% | 4.548 | 70165.3 | 98.1% |

ImageNet dataset and $\epsilon = 1.0$ in the CIFAR-10 dataset, where $d$ is the image dimension. Tables 27, 28, and 29 show that the Prior-OPT with two surrogate models performs the best in most cases, and the PGD initialization of $\theta$ (e.g., Prior-OPT$_{\theta_0^{\text{PGD}} + \text{ResNet50}}$) can effectively reduce the AUC.

Table 30 demonstrates that Prior-OPT and Prior-Sign-OPT deliver competitive performance in $\ell_\infty$-norm attacks on ImageNet, surpassing Sign-OPT in average $\ell_\infty$ distortions, further validating our approach's effectiveness across attack types.

Fig. 7 shows the experimental results of untargeted $\ell_2$-norm attacks on the ImageNet dataset. The results demonstrate that Prior-OPT significantly outperforms all compared methods, including SQBA and BBA that also use surrogate models. The results also show that using multiple surrogate models can further boost performance. In addition, the PGD initialization ensures the algorithm's initial attack direction $\theta_0$ is already good, which enables it to achieve better untargeted attack performance even with a small number of queries (e.g., the query budget of 1,000). Fig. 8 shows that on ImageNet, Prior-Sign-OPT outperforms Prior-OPT in targeted $\ell_2$-norm attacks on CNN models, especially when using multiple surrogate models compared to a single one.

Figs. 9 and 10 show the attack success rates of untargeted and targeted attacks on the ImageNet dataset. In untargeted $\ell_2$-norm attacks (Fig. 9), Prior-OPT with two surrogate models achieves the highest success rate, and both Prior-OPT and Prior-Sign-OPT outperform the baseline Sign-OPT. For targeted attacks (Fig. 10), Prior-Sign-OPT exhibits superior performance compared to Prior-OPT. One plausible explanation is that Prior-Sign-OPT employs a more query-efficient strategy by leveraging the sign of directional derivatives, which requires only a single query per direction. When $\alpha_i$ is small, Prior-OPT, which relies on binary search to fully exploit prior information, becomes less efficient due to its high query cost. Consequently, Prior-Sign-OPT holds a relative advantage in such scenarios.

Figs. 11 and 12 present the experimental results on the CIFAR-10 dataset. The results demonstrate that SQBA and Prior-Sign-OPT achieve the highest performance among all evaluated methods. In future work, we aim to further improve the performance of our approach on the CIFAR-10 dataset.

Figs. 13a, 13b, and 13c present the ablation studies of $\mathbb{E}[\gamma^2]$ based on our theoretical results, with Eq. (10) for Sign-OPT, Eq. (12) for Prior-Sign-OPT, and Eq. (17) for Prior-OPT. Fig. 13d demonstrates the performance of Prior-Sign-OPT in untargeted $\ell_2$-norm attacks against the Swin Transformer on the ImageNet dataset, evaluated with different values of $q$.

Table 30: Mean $\ell_\infty$ distortions of untargeted attacks across various query budgets on ImageNet.

| Target Model | Method | Mean $\ell_\infty$ distortions | | | | |
|---|---|---|---|---|---|---|
| | | @1K | @2K | @5K | @8K | @10K |
| Inception-v3 | TA (Ma et al., 2021b) | **0.397** | 0.379 | 0.359 | 0.348 | 0.342 |
| | Sign-OPT (Cheng et al., 2020) | 0.726 | 0.403 | 0.156 | 0.100 | 0.084 |
| | SVM-OPT (Cheng et al., 2020) | 0.723 | 0.389 | 0.155 | 0.102 | 0.088 |
| | Prior-Sign-OPT$_{IncResV2}$ | 0.678 | 0.365 | 0.117 | 0.086 | **0.080** |
| | Prior-Sign-OPT$_{IncResV2\&Xception}$ | 0.640 | 0.318 | **0.102** | **0.083** | **0.080** |
| | Prior-OPT$_{IncResV2}$ | 0.581 | 0.267 | 0.138 | 0.115 | 0.109 |
| | Prior-OPT$_{IncResV2\&Xception}$ | 0.502 | **0.208** | 0.119 | 0.111 | 0.109 |
| Inception-v4 | TA (Ma et al., 2021b) | **0.420** | 0.402 | 0.381 | 0.370 | 0.365 |
| | Sign-OPT (Cheng et al., 2020) | 0.794 | 0.450 | 0.175 | 0.111 | 0.093 |
| | SVM-OPT (Cheng et al., 2020) | 0.811 | 0.446 | 0.178 | 0.113 | 0.096 |
| | Prior-Sign-OPT$_{IncResV2}$ | 0.756 | 0.408 | 0.136 | 0.097 | 0.089 |
| | Prior-Sign-OPT$_{IncResV2\&Xception}$ | 0.700 | 0.348 | **0.107** | **0.086** | **0.082** |
| | Prior-OPT$_{IncResV2}$ | 0.645 | 0.302 | 0.157 | 0.131 | 0.123 |
| | Prior-OPT$_{IncResV2\&Xception}$ | 0.558 | **0.218** | 0.117 | 0.108 | 0.106 |
| ResNet-101 | TA (Ma et al., 2021b) | 0.301 | 0.285 | 0.267 | 0.258 | 0.253 |
| | Sign-OPT (Cheng et al., 2020) | 0.437 | 0.247 | 0.101 | 0.066 | 0.057 |
| | SVM-OPT (Cheng et al., 2020) | 0.461 | 0.254 | 0.110 | 0.075 | 0.066 |
| | Prior-Sign-OPT$_{ResNet50}$ | 0.404 | 0.218 | **0.074** | **0.053** | **0.049** |
| | Prior-OPT$_{ResNet50}$ | **0.289** | **0.138** | 0.075 | 0.064 | 0.060 |
| ResNeXt-101 (64 × 4d) | TA (Ma et al., 2021b) | **0.362** | 0.344 | 0.323 | 0.313 | 0.307 |
| | Sign-OPT (Cheng et al., 2020) | 0.611 | 0.326 | 0.131 | 0.090 | 0.078 |
| | SVM-OPT (Cheng et al., 2020) | 0.667 | 0.336 | 0.131 | 0.089 | 0.078 |
| | Prior-Sign-OPT$_{ResNet50}$ | 0.574 | 0.303 | 0.104 | **0.075** | **0.069** |
| | Prior-OPT$_{ResNet50}$ | 0.428 | **0.196** | **0.097** | 0.080 | 0.075 |
| SENet-154 | TA (Ma et al., 2021b) | **0.355** | 0.336 | 0.316 | 0.306 | 0.300 |
| | Sign-OPT (Cheng et al., 2020) | 0.563 | 0.326 | 0.132 | 0.082 | 0.067 |
| | SVM-OPT (Cheng et al., 2020) | 0.570 | 0.325 | 0.132 | 0.082 | 0.068 |
| | Prior-Sign-OPT$_{ResNet50}$ | 0.536 | 0.314 | **0.113** | **0.074** | **0.065** |
| | Prior-OPT$_{ResNet50}$ | 0.448 | **0.246** | 0.129 | 0.102 | 0.094 |
| ViT | TA (Ma et al., 2021b) | **0.399** | 0.379 | 0.358 | 0.348 | 0.342 |
| | Sign-OPT (Cheng et al., 2020) | 0.602 | 0.302 | 0.105 | 0.072 | **0.064** |
| | SVM-OPT (Cheng et al., 2020) | 0.651 | 0.310 | 0.107 | 0.075 | 0.068 |
| | Prior-Sign-OPT$_{ResNet50}$ | 0.597 | 0.334 | 0.118 | 0.084 | 0.077 |
| | Prior-Sign-OPT$_{ResNet50\&ConViT}$ | 0.539 | 0.273 | **0.090** | **0.069** | 0.065 |
| | Prior-OPT$_{ResNet50}$ | 0.591 | 0.352 | 0.178 | 0.136 | 0.123 |
| | Prior-OPT$_{ResNet50\&ConViT}$ | 0.429 | **0.217** | 0.124 | 0.110 | 0.106 |
| GC ViT | TA (Ma et al., 2021b) | **0.380** | 0.365 | 0.348 | 0.339 | 0.335 |
| | Sign-OPT (Cheng et al., 2020) | 0.680 | 0.434 | 0.186 | 0.119 | 0.098 |
| | SVM-OPT (Cheng et al., 2020) | 0.678 | 0.427 | 0.183 | 0.116 | 0.097 |
| | Prior-Sign-OPT$_{ResNet50}$ | 0.670 | 0.445 | 0.183 | 0.116 | 0.097 |
| | Prior-Sign-OPT$_{ResNet50\&ConViT}$ | 0.642 | 0.389 | **0.141** | **0.092** | **0.079** |
| | Prior-OPT$_{ResNet50}$ | 0.652 | 0.455 | 0.248 | 0.185 | 0.163 |
| | Prior-OPT$_{ResNet50\&ConViT}$ | 0.538 | **0.305** | 0.160 | 0.131 | 0.122 |
| Swin Transformer | TA (Ma et al., 2021b) | **0.536** | 0.515 | 0.491 | 0.479 | 0.472 |
| | Sign-OPT (Cheng et al., 2020) | 1.009 | 0.625 | 0.258 | 0.159 | 0.128 |
| | SVM-OPT (Cheng et al., 2020) | 1.036 | 0.622 | 0.251 | 0.157 | 0.131 |
| | Prior-Sign-OPT$_{ResNet50}$ | 1.000 | 0.647 | 0.262 | 0.162 | 0.133 |
| | Prior-Sign-OPT$_{ResNet50\&ConViT}$ | 0.909 | 0.513 | 0.169 | **0.105** | **0.088** |
| | Prior-OPT$_{ResNet50}$ | 0.942 | 0.619 | 0.309 | 0.226 | 0.198 |
| | Prior-OPT$_{ResNet50\&ConViT}$ | 0.662 | **0.321** | **0.159** | 0.129 | 0.120 |

Figs. 14, 16, and 18 show examples of adversarial images generated using different numbers of queries in targeted attacks with Sign-OPT, Prior-Sign-OPT, and Prior-OPT methods. Figs. 15, 17, and 19 show the corresponding adversarial perturbations for the Sign-OPT, Prior-Sign-OPT, and Prior-OPT methods. Initially, all methods start with an image from the target class and iteratively minimize the $\ell_2$-norm distance to the original image, while maintaining the predicted label as the target class. Prior-Sign-OPT and Prior-OPT achieve a faster reduction in perturbation magnitude compared to Sign-OPT, as shown in Figs. 15, 17, and 19.

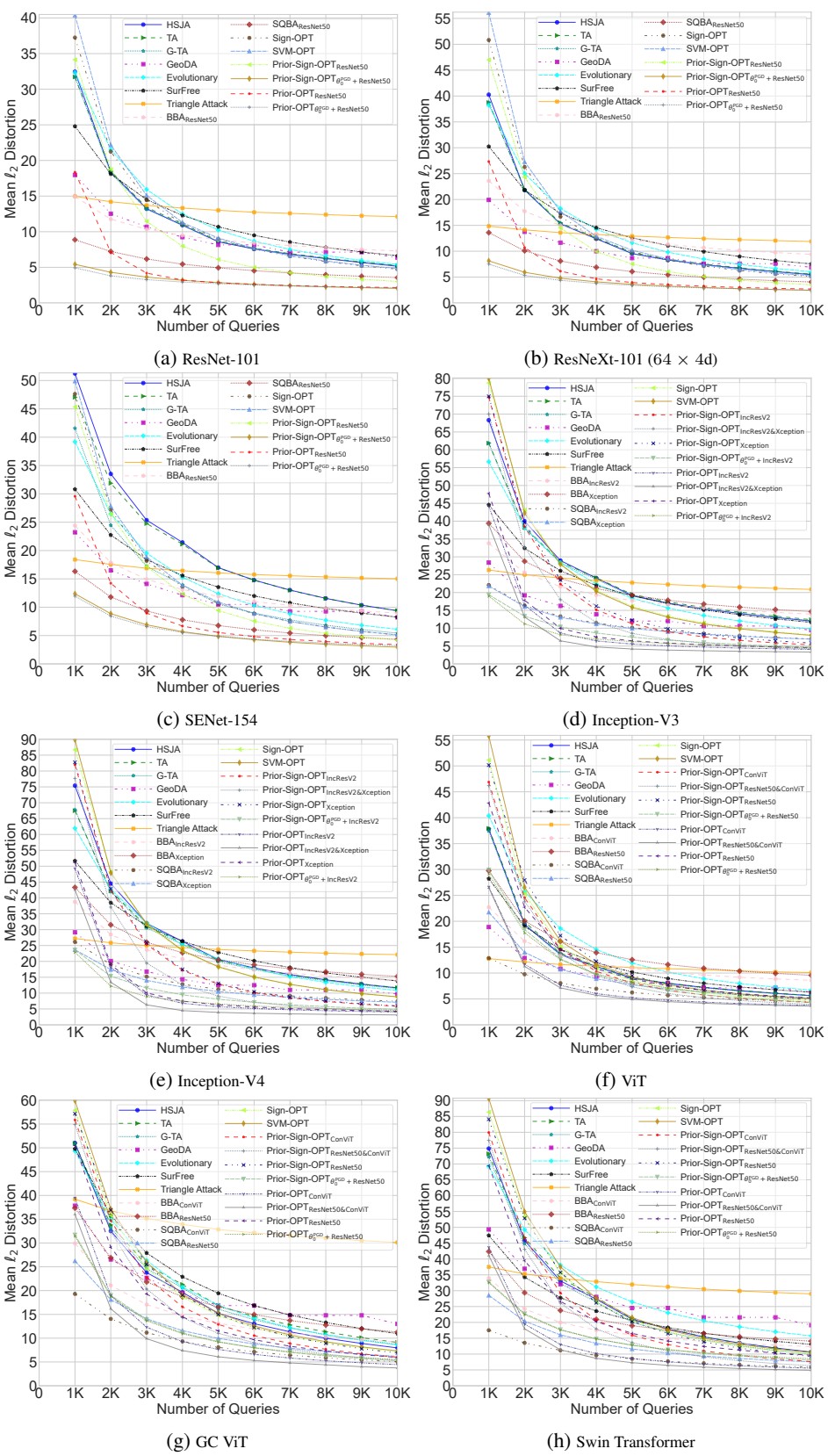

Figure 7: Mean distortions of untargeted $\ell_2$-norm attack under different query budgets on the ImageNet dataset.

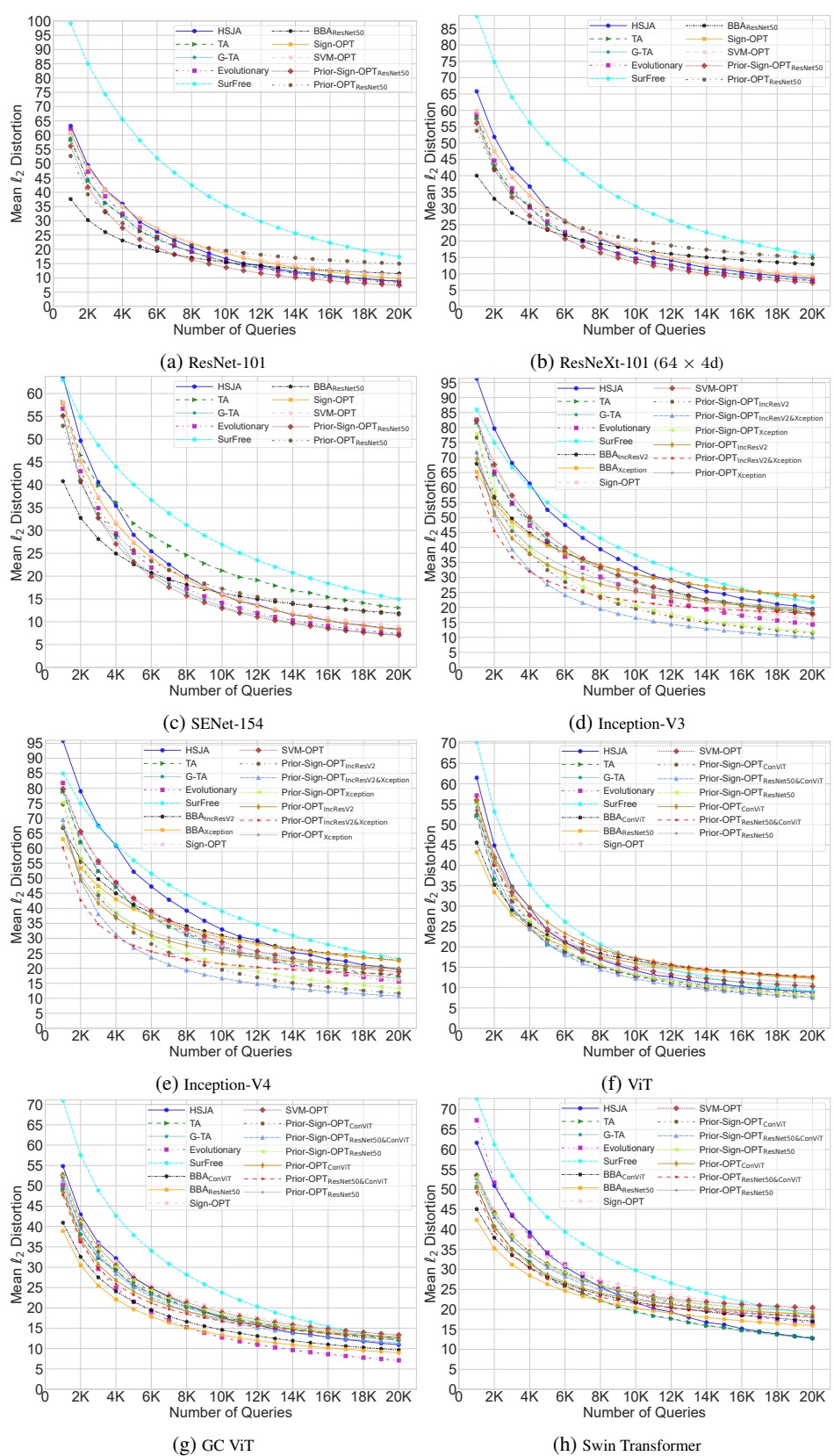

Figure 8: Mean distortions of targeted $\ell_2$-norm attacks under different query budgets on the ImageNet dataset.

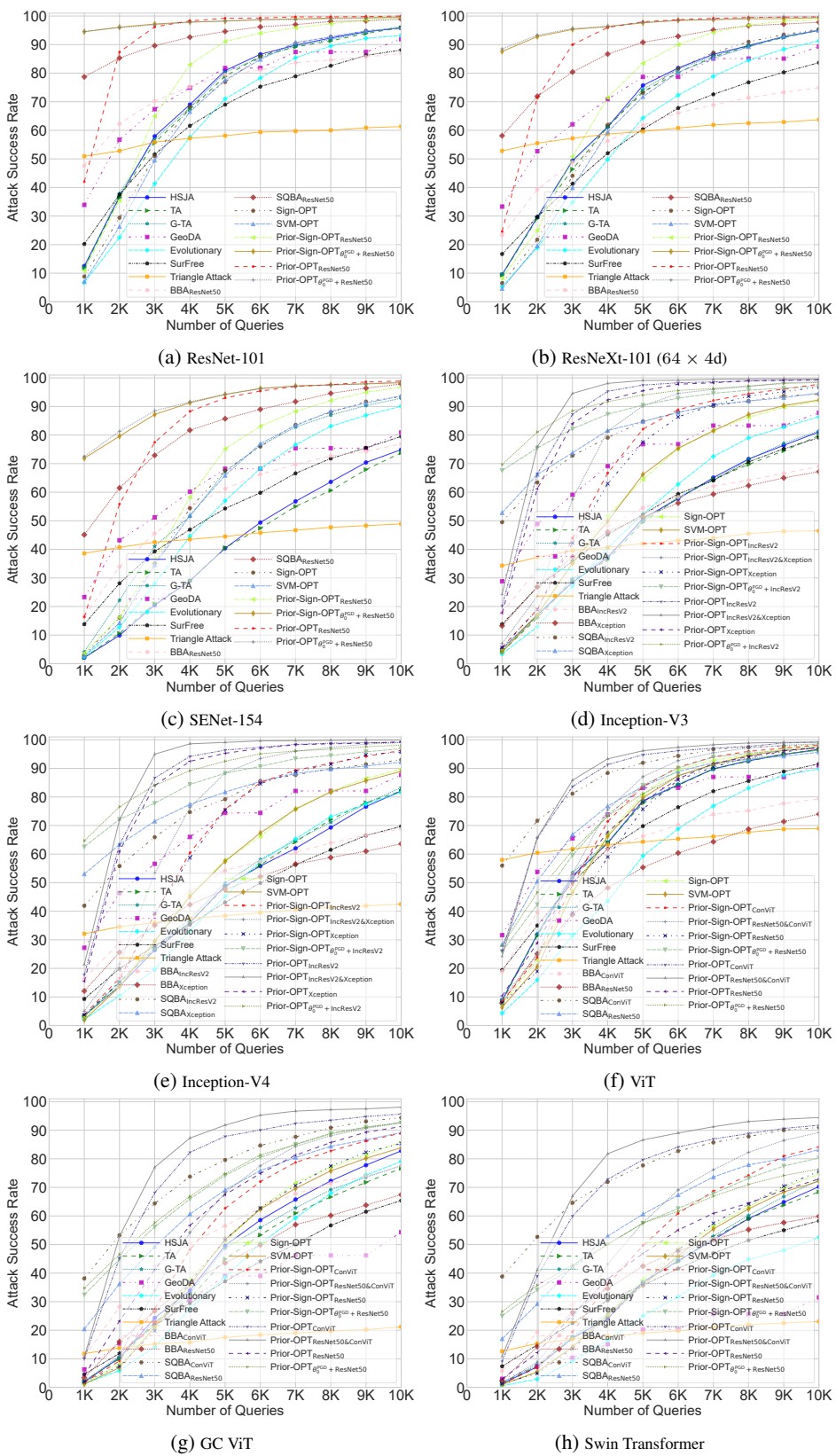

Figure 9: Attack success rates of untargeted $\ell_2$-norm attacks under different query budgets on the ImageNet dataset.

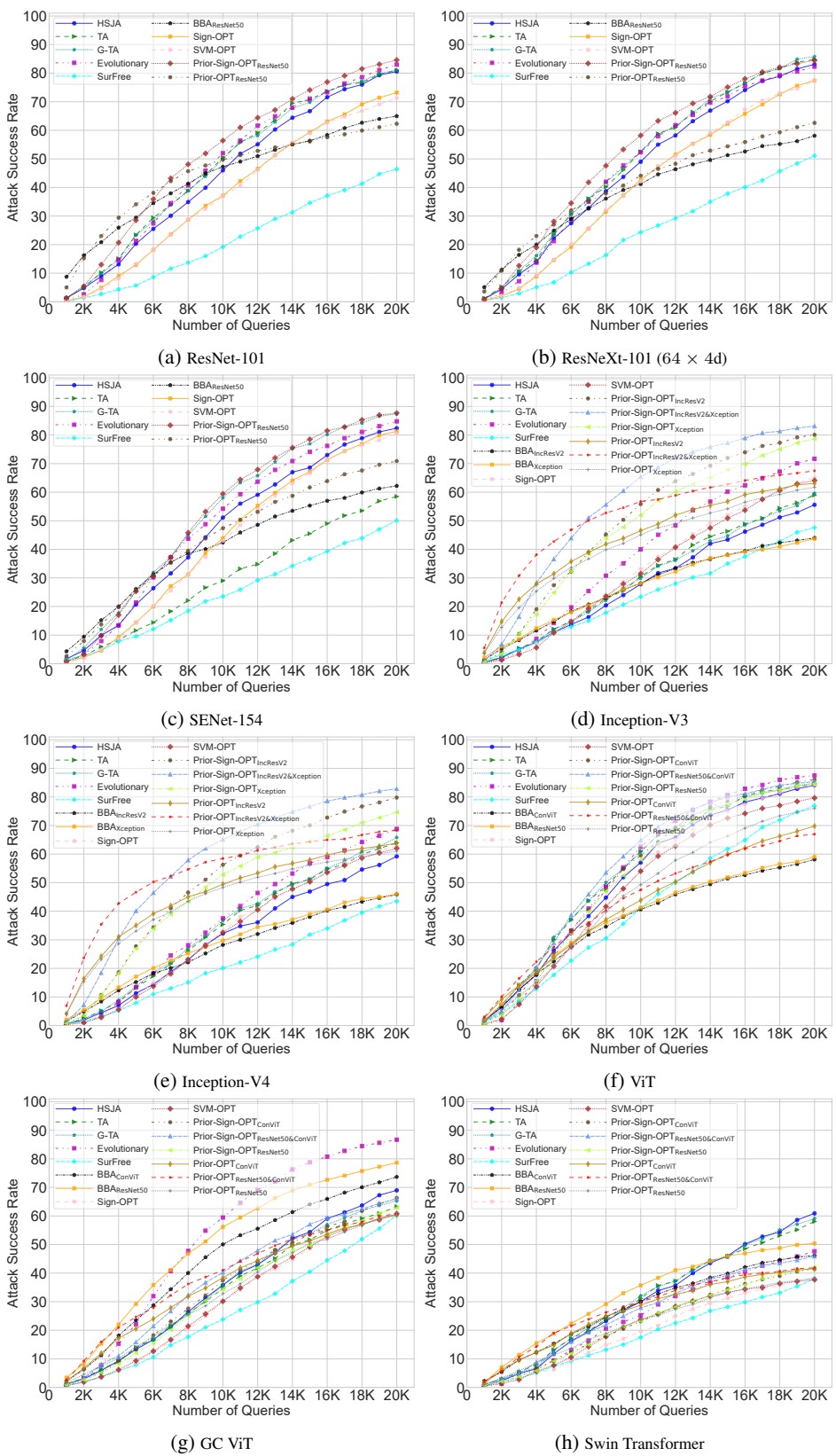

Figure 10: Attack success rates of targeted $\ell_2$-norm attacks under different query budgets on the ImageNet dataset.

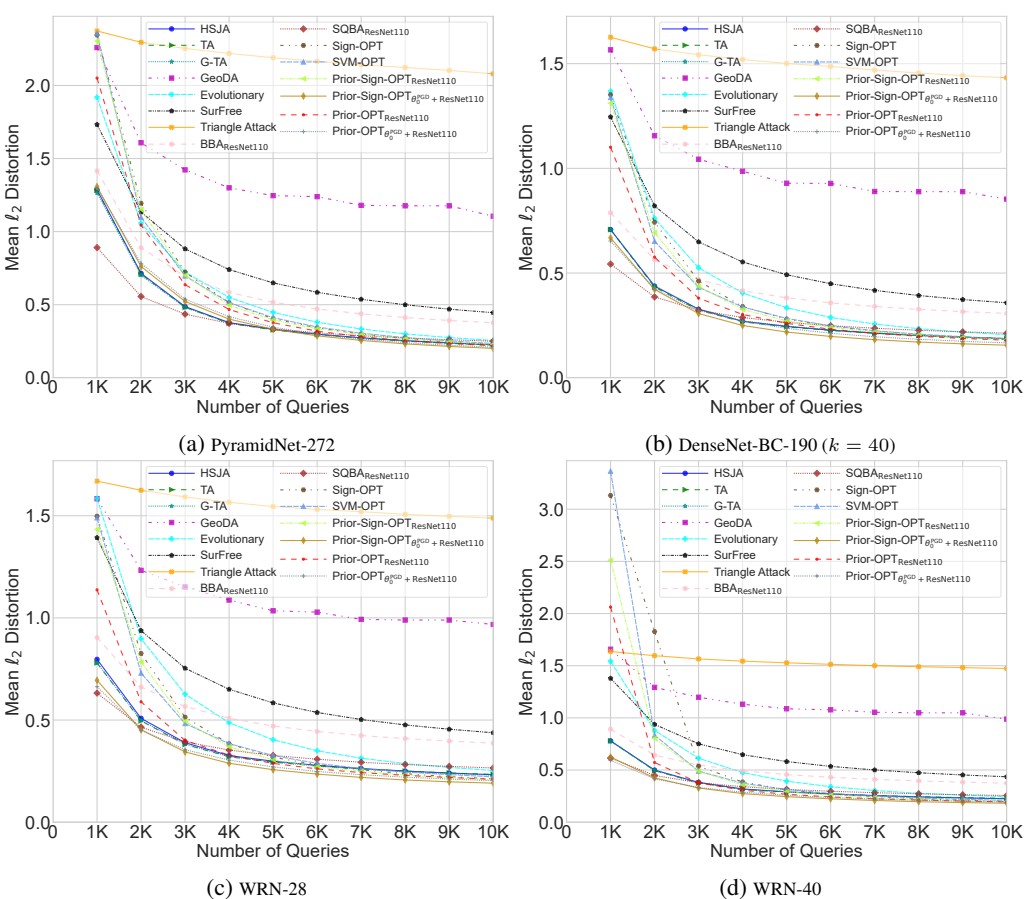

Figure 11: Mean distortions of untargeted $\ell_2$-norm attack under different query budgets on the CIFAR-10 dataset.

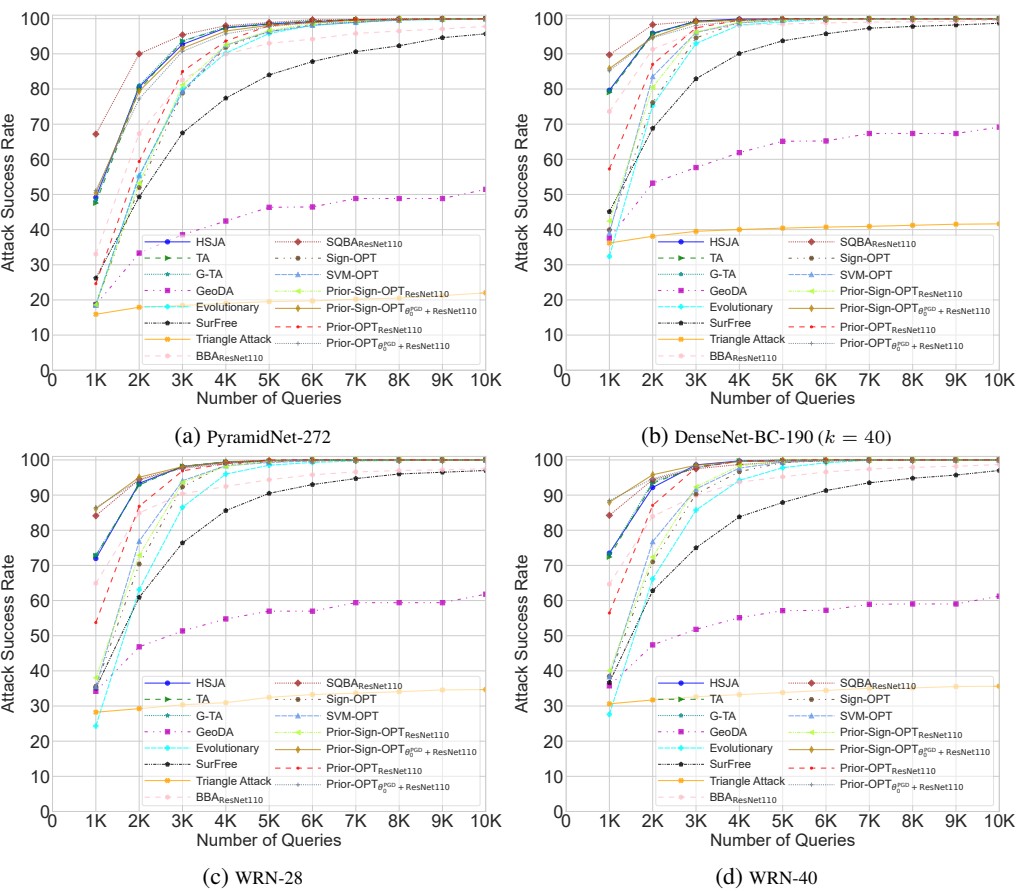

Figure 12: Attack success rates of untargeted $\ell_2$-norm attacks under different query budgets on the CIFAR-10 dataset.

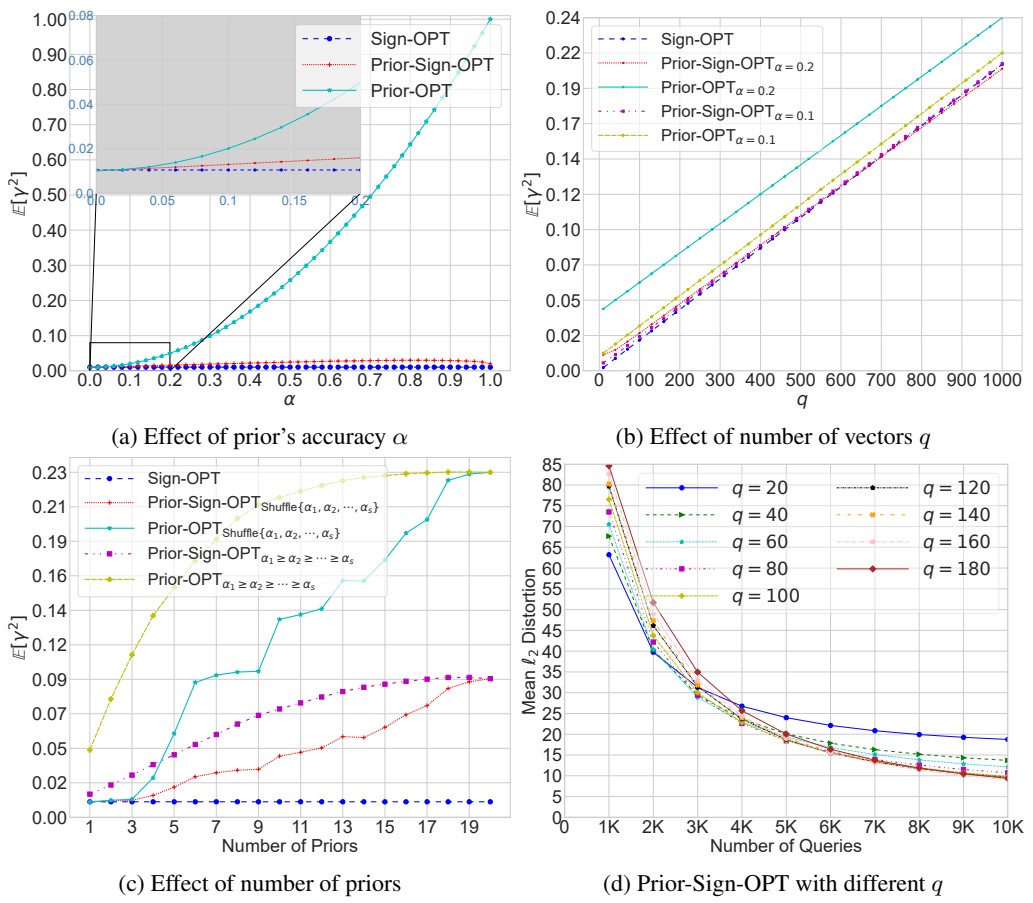

Figure 13: Experimental results of ablation studies of $\mathbb{E}[\gamma^2]$. Figs. 13a, 13b, and 13c are based on theoretical results (Eqs. (10), (12) and (17)) with $d = 3072$. Fig. 13d demonstrates the results of attacking against Swin Transformer on the ImageNet dataset using Prior-Sign-OPT with different $q$.

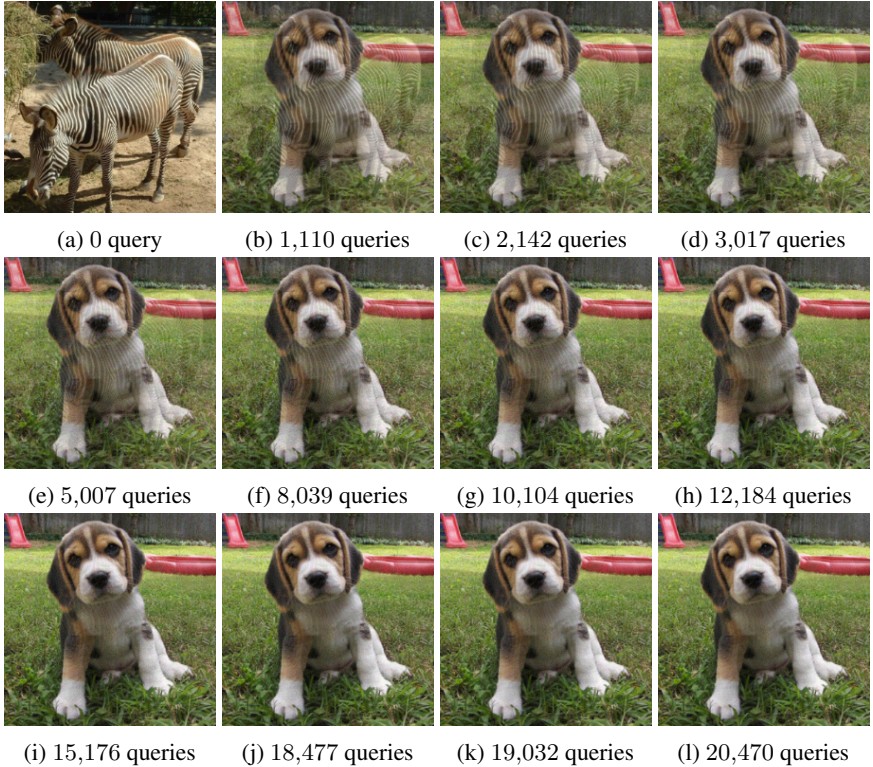

Figure 14: Adversarial images generated with different queries in Sign-OPT targeted attacks against ResNet-101.

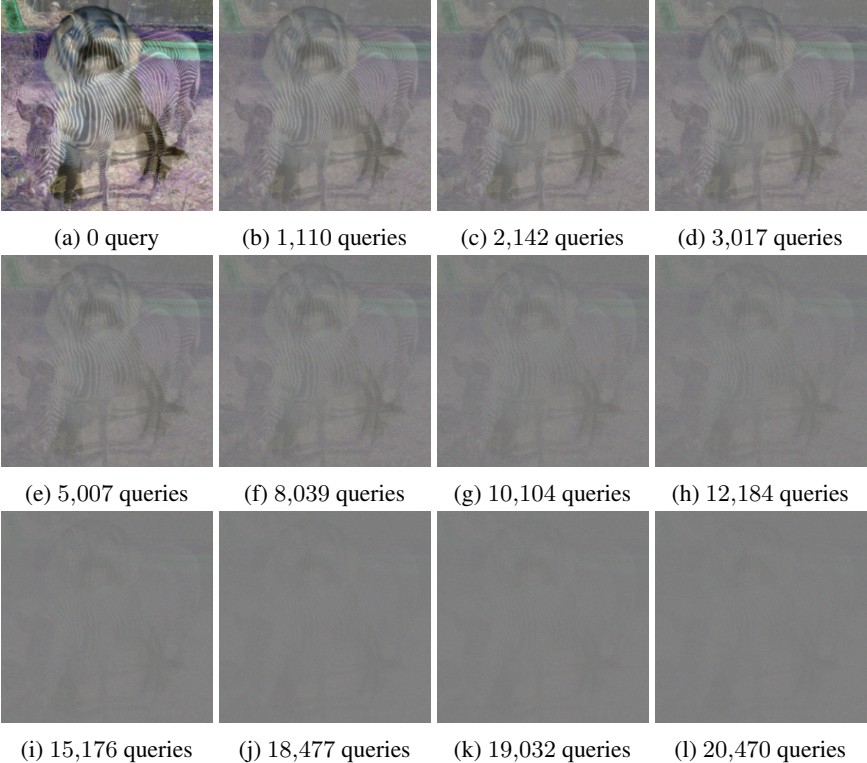

Figure 15: The corresponding adversarial perturbations generated with different queries in Sign-OPT targeted attacks against ResNet-101.

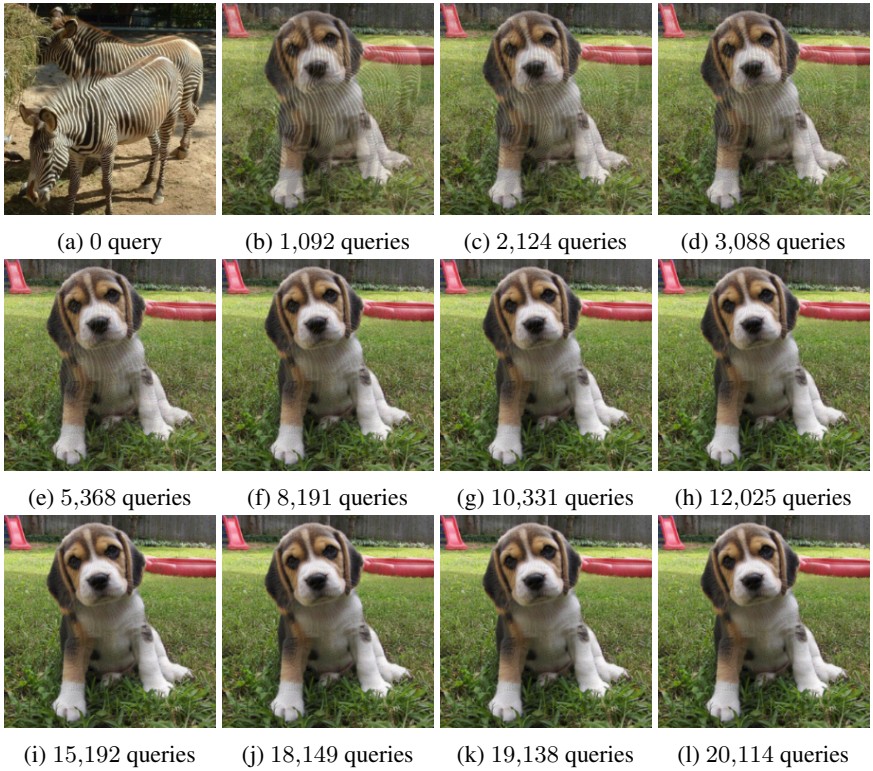

Figure 16: Adversarial images generated with different queries in Prior-Sign-OPT targeted attacks against ResNet-101.

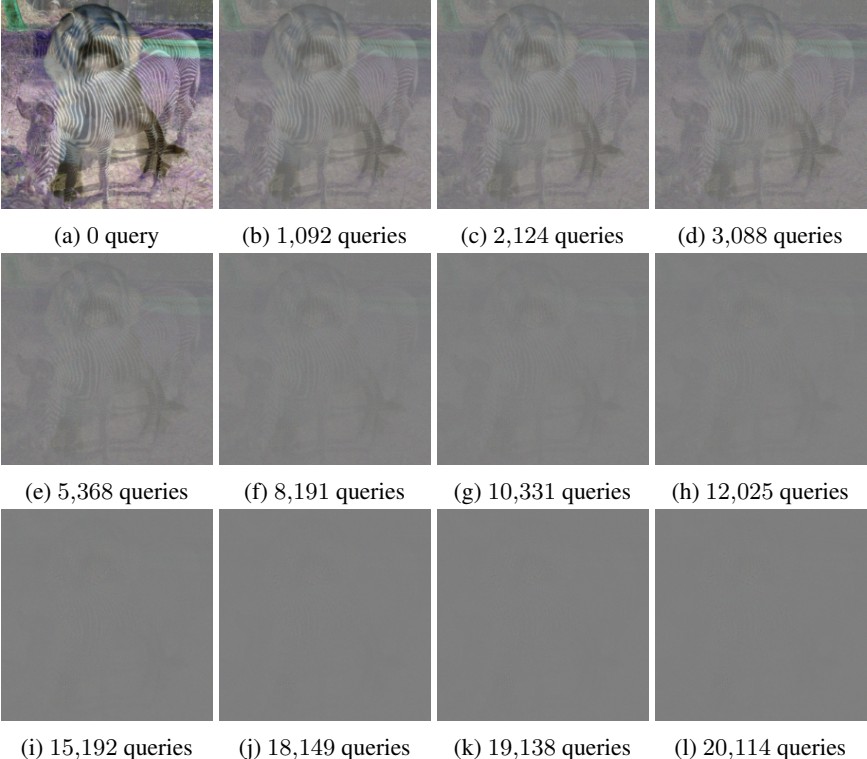

Figure 17: The corresponding adversarial perturbations generated with different queries in Prior-Sign-OPT targeted attacks against ResNet-101.

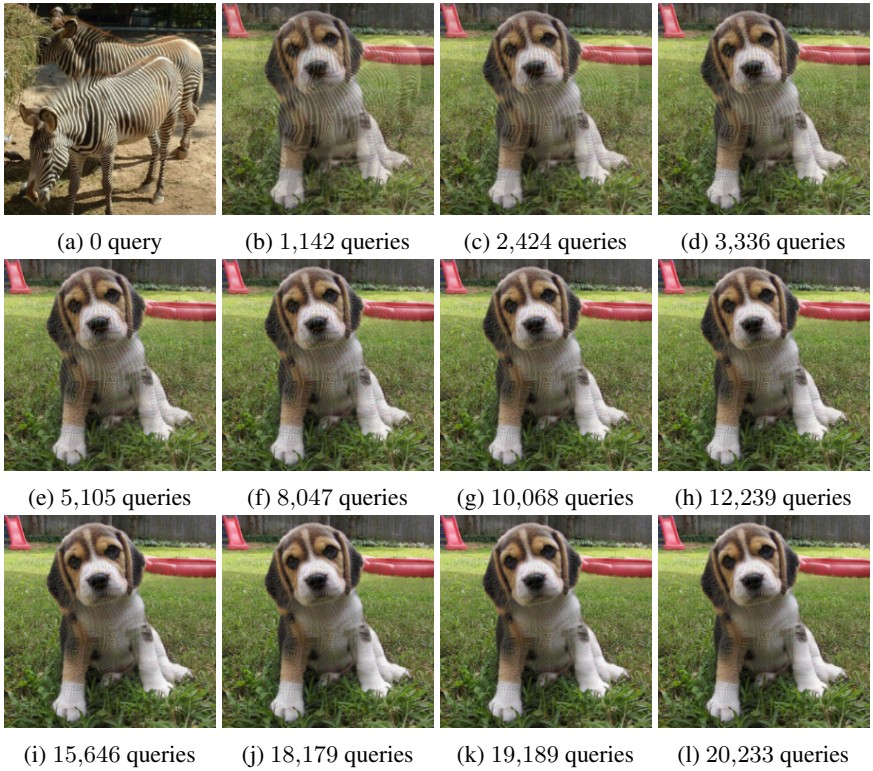

Figure 18: Adversarial images generated with different queries in Prior-OPT targeted attacks against ResNet-101.

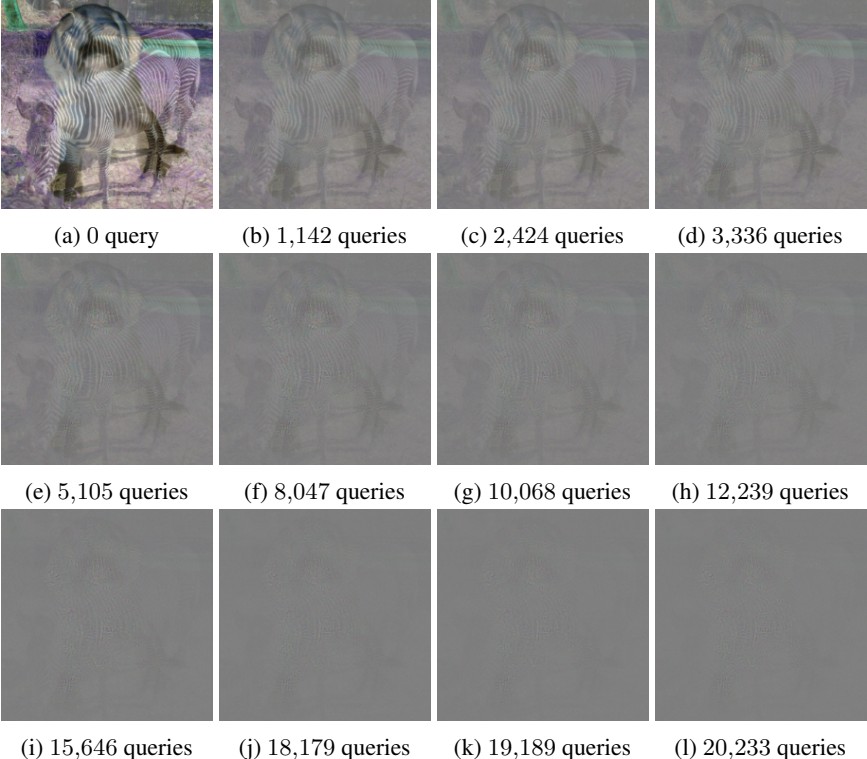

Figure 19: The corresponding adversarial perturbations generated with different queries in Prior-OPT targeted attacks against ResNet-101.

