# OpenReview forum: "Boosting Ray Search Procedure of Hard-label Attacks with Transfer-based Priors"
_ICLR.cc/2025/Conference — ICLR 2025 Spotlight_

### Official Review · Reviewer_cAs8 · 2024-11-03

**Soundness:** 3
**Presentation:** 3
**Contribution:** 3
**Rating:** 8
**Confidence:** 3

**Summary:**

The paper addresses the challenge of hard-label adversarial attacks and enhances the efficiency of searching for the optimal ray direction by introducing a prior-guided approach that leverages transfer-based priors from surrogate models. Theoretical analysis is provided to validate the effectiveness of their gradient estimation technique, and extensive experiments on ImageNet and CIFAR-10 demonstrate significant improvements in query efficiency compared to 11 state-of-the-art methods.

**Strengths:**

1. The paper provides a solid theoretical foundation for the proposed gradient estimation technique.
2. Extensive experiments on multiple datasets demonstrate the practical effectiveness of the proposed method.
3. The focus on improving query efficiency is critical, as it addresses one of the major challenges in hard-label attacks, making the method more applicable in real-world scenarios.

**Weaknesses:**

1. The following work [1] also proposes to use a pre-trained model to reduce the query complexity of hard-label adversarial attacks. Please clarify the novelty based on this work.
2. The paper does not adequately address the scalability of the proposed method, particularly in scenarios involving larger models or more complex datasets.
3. The reliance on transfer-based priors from surrogate models raises concerns about overfitting and the applicability of the method to unseen models or datasets.

[1] Yan, Z., Guo, Y., Liang, J., & Zhang, C. (2021). Policy-driven attack: Learning to query for hard-label black-box adversarial examples. In International Conference on Learning Representations.

**Questions:**

See the weaknesses for details

---

> ### Author Response · Authors · 2024-11-22
> **Response to Reviewer cAs8 (pt 1/2)**
>
> We would like to thank the reviewer for the positive comments, and we shall carefully revise the paper following your suggestions.
>
>
> **Q1:** The following work [1] (Policy Driven Attack, PDA) also proposes to use a pre-trained model to reduce the query complexity of hard-label adversarial attacks. Please clarify the novelty based on this work.
>
> **A1:** Policy Driven Attack (PDA) is a method derived from HSJA with added pretraining modifications and relies on two types of surrogate models prior to performing the attack. The first is a traditional classifier, which is used to generate image-gradient pairs as training data. The second is a policy network, pre-trained on the generated data, which is subsequently utilized to predict the gradient during the attack. While effective in certain scenarios, PDA has two significant limitations when compared to our approach.
>
> First, PDA requires a pretraining process, and its performance heavily depends on the surrogate model being highly similar or identical to the target model. If the surrogate model differs substantially from the target model, the policy network may struggle to predict accurate gradients, leading to degraded attack performance. This is a fundamental limitation, as hard-label attacks are designed to operate in a black-box setting. If gradients from the target model are accessible for pretraining, the attack essentially transitions into a white-box setting, undermining the essence of a black-box attack and rendering the claimed novelty less meaningful.
>
> Second, PDA employs reinforcement learning during the attack phase for fine-tuning, which is computationally expensive and time-consuming. This significantly impacts the practicality and efficiency of PDA, especially in scenarios requiring a large number of attacks or real-time constraints.
>
> In contrast, our approach, Prior-OPT, does not require pretraining of a policy network or any fine-tuning steps. This makes it simpler, more efficient, and easier to deploy. Unlike PDA, Prior-OPT avoids the use of reinforcement learning, making it less computationally intensive. Furthermore, when there is a significant discrepancy between the surrogate model and the target model, our gradient estimation mechanism adaptively assigns a lower weight to the surrogate gradient, mitigating the impact of model mismatch.
>
> **Therefore, PDA and Prior-OPT are fundamentally different methods.** Theoretical analysis further supports that our approach guarantees a lower bound on the expected cosine similarity of the estimated gradient, ensuring its reliability and effectiveness in diverse settings.

---

> ### Author Response · Authors · 2024-11-22
> **Response to Reviewer cAs8 (pt 2/2)**
>
> **Q2:** The paper does not adequately address the scalability of the proposed method, particularly in scenarios involving larger models or more complex datasets.
>
> **A2:**  To evaluate the scalability of our proposed method, we conduct experiments of attacking a large **CLIP** with the backbone of **ViT-B/32**, and the surrogate models include **ImageNet pretrained ResNet-50, ConViT, CrossViT, MaxViT, and ViT**. It is worth noting that these surrogate models are pretrained on ImageNet and their training paradigms are entirely different from the CLIP model.
> The CLIP model with ViT-B/32 backbone was trained on *400 million image-text pairs from the internet by using contrastive learning*, enabling it to generalize well via natural language supervision. The CLIP model can serve as a zero-shot image classifier by leveraging its ability to align images and text in a shared embedding space. We encapsulate it as a 1000-class image classifier by constructing a set of text prompts representing the classes (e.g., "a photo of a cat," "a photo of a dog") and embedding them into the same space as the image. During inference, the image is embedded, and the class with the highest similarity score between the image embedding and the text embeddings is selected as the predicted label.
>
> The tested images consist of 6 randomly selected pictures downloaded from the Internet, making the data and target model both unseen. The results under a maximum query budget of 10,000 are summarized in the following tables.
>
> *Table 1: Attack Success Rate (ASR), average queries across all samples (with a query count of 10,000 assigned to unsuccessful samples), and the $\ell_2$ distortion of the final samples for untargeted attacks against **CLIP** on **6 randomly downloaded images**. The surrogate models include ImageNet pretrained ResNet-50, ConViT, CrossViT, MaxViT, and ViT.*
> | Metric | no prior(Sign-OPT) | 1 prior | 2 priors | 3 priors |4 priors | 5 priors |
> | :- | :- |:- | :- | :- |:- | :- |
> | Prior-OPT's ASR | 16.7% | 33.3% | 66.7% | 83.3% | **100%** | 83.3% |
> | Prior-OPT's Avg Queries | 8502 | 8082 | 6003 | 3654 | 2778 | **2660** |
> | Prior-OPT's Distortion | 17.176 | 23.549 | 9.895 | 9.969 | 8.645| **6.796** |
>
> As demonstrated in the table, incorporating more surrogate models (priors) significantly enhances attack performance, highlighting the robustness and generalization of our method. Notably, even though the surrogate models are ImageNet-pretrained and their training methods differ entirely from those of the CLIP model, they still contribute to improved performance. Let us now conduct an analysis of the robustness of our method in real-world applications.
>
> **Generalization to Real-World Scenarios:**
>
> Adversarial example generation fundamentally relies on gradient vectors, which increase classification loss to induce misclassification. Our method and theoretical framework focus on the cosine similarity between two vectors: the estimated gradient and the true gradient. This ensures the universality of our approach across various image classifiers.
>
> In real-world scenarios, a common challenge is the gradient discrepancy between surrogate and target models, which leads to differing gradient directions. Our approach addresses this issue through the $\alpha$ variable, defined in Eq. (11), Eq. (12), Eq. (15), Eq. (16), and Eq. (17). $\alpha$ represents the cosine similarity between the $i$-th prior and the true gradient and is a key component of our theoretical analysis, ensuring robustness in diverse environments.
>
> **Applicability to Modern Architectures and Datasets:**
>
> Modern state-of-the-art image classification networks, primarily based on CNNs and ViTs, dominate the field, with benchmarks such as ImageNet and CIFAR-10 serving as standard datasets. The leading models on the ImageNet leaderboard consistently rely on these architectures. This supports the relevance and applicability of our approach to prevailing real-world scenarios.
>
> ***
>
> **Q3:** The reliance on transfer-based priors from surrogate models raises concerns about overfitting and the applicability of the method to unseen models or datasets.
>
> **A3:** See **A2**.

---

### Official Review · Reviewer_hzgv · 2024-11-04

**Soundness:** 3
**Presentation:** 3
**Contribution:** 3
**Rating:** 8
**Confidence:** 3

**Summary:**

This paper proposes a new method for hard-label attacks which uses transfer-based priors to estimate the gradient of the loss in hard-label attacks, which is quite query efficient.

**Strengths:**

1. This paper proposes a surrogate function to estimate compute the transfer-based priors, which can avoid the non-differentiable issue of using binary search to learn the prior.

2. The gradient of the loss in Eq. (3) can be estimated by the projection of the true gradient onto the subspace spanned by these transfer-based priors and some random directions.

**Weaknesses:**

An efficiency experiment is required to show how the proposed method improves the ray search efficiency.

**Questions:**

N/A

---

> ### Author Response · Authors · 2024-11-21
> **Response to Reviewer hzgv (pt 1/2)**
>
> We would like to thank the reviewer for the positive comments, and we shall carefully revise the paper following your suggestions.
>
> **Q1:** An efficiency experiment is required to show how the proposed method improves the ray search efficiency.
>
> **A1:** In query-based attacks, the term *efficiency* refers to *query efficiency*, which evaluates the performance of the proposed method in two aspects:
>
>  (1) for a fixed number of queries, the reduction in distortion or the improvement in attack success rate;
>
>  (2) for a fixed number of iterations, the average number of queries required to successfully attack all samples. These metrics provide a comprehensive measure of the method's efficiency in enhancing the ray search process.
>
> Throughout our paper, we have presented extensive experimental results (Tables 1, 2, 14, 15, 16, 17, and Figures 3, 4, 7, 8, 9, 10, 11, 12) to illustrate the query efficiency advantages of our approach.
>
>
> Now, we present the additional experimental results as following tables, which will be added to the revised version of our paper.
>
> *Table 1: Attack Success Rate (ASR), average successful queries of all samples , and the $\ell_2$ distortion of final sample on the untargeted attacks against **Swin Transformer** on the **ImageNet** dataset.*
> | Metric | no prior(Sign-OPT) | 1 prior | 2 priors | 3 priors | 4 priors | 5 priors |
> | :- | :- |:- | :- | :- |:- | :- |
> | Prior-Sign-OPT's ASR | 74.7% | 84.2% | 89.2% | 94.6% | 96.3% | **97.3%** |
> | Prior-OPT's ASR | 74.7% | 91.8% | 94.5% | 98.3% | 99.3% | **99.6%** |
> | Prior-Sign-OPT's Avg Queries | 6408 | 5606 | 5072 | 4275 | 3773 | **3479** |
> | Prior-OPT's Avg Queries | 6408 | 3420 | 2991 | 2248 | 1955 | **1856** |
> | Prior-Sign-OPT's $\ell_2$ Distortion | 9.899 | 7.622 | 6.331 | 4.778 | 4.169 | **3.467** |
> | Prior-OPT's $\ell_2$ Distortion | 9.899 | 5.558 | 4.896 | 3.687 | 2.85 | **2.577** |
>
>
> *Table 2: Attack Success Rate (ASR), average successful queries of all samples, and the $\ell_2$ distortion of final sample on the untargeted attacks against **GC ViT** on the **ImageNet** dataset.*
> | Metric | no prior(Sign-OPT) | 1 prior | 2 priors | 3 priors | 4 priors | 5 priors |
> | :- | :- |:- | :- | :- |:- | :- |
> | Prior-Sign-OPT's ASR | 85.9% | 89.0% | 92.6% | 95.7% | 98.3% | **98.4%** |
> | Prior-OPT's ASR | 85.9% | 95.7% | 98.1% | 98.9% | 99.7% |**99.8%** |
> | Prior-Sign-OPT's Avg Queries | 5452 | 4875 | 4424 | 3740 | 3175 | **3011** |
> | Prior-OPT's Avg Queries | 5452 | 2879 | 2433 | 1990 | 1701 | **1668** |
> | Prior-Sign-OPT's Distortion | 7.185 | 5.972 | 5.342 | 4.215 | 3.212 | **2.893** |
> | Prior-OPT's Distortion | 7.185 | 4.453 | 3.747 | 3.264 | 2.171 | **2.159** |
>
>
> *Table 3: Attack Success Rate (ASR), average successful queries of all samples, and the $\ell_2$ distortion of final sample on the untargeted attacks against **ResNet-101** on the **ImageNet** dataset.*
> | Metric | no prior(Sign-OPT) | 1 prior | 2 priors | 3 priors | 4 priors | 5 priors |
> | :- | :- |:- | :- | :- |:- | :- |
> | Prior-Sign-OPT's ASR | 95.9% | 99.4% | 99.4% | 99.4% | **99.7%** | **99.7%** |
> | Prior-OPT's ASR | 95.9% | 100.0% | 99.9% | 100.0% | 99.9% | **100.0%** |
> | Prior-Sign-OPT's Avg Queries | 3541 | 2786 | 2518 | 2198 | 2099 | **2042** |
> | Prior-OPT's Avg Queries | 3541 | 1312 | 1237 | 1107 | 1103 | **1102** |
> | Prior-Sign-OPT's Distortion | 4.754 | 3.019 | 2.593 | 2.122 | 1.989 | **1.951** |
> | Prior-OPT's Distortion | 4.754 | 2.158 | 1.979 | 1.653 | **1.596** | 1.605 |
>
> We report the average queries of 1000 samples as the following table, where the query times of failed samples are recorded as 10,000.
>
> *Table 4: Average successful queries on the **ImageNet** dataset.*
> | Method/Mean Queries | InceptionV3 | InceptionV4 | ResNet-101 | ResNeXt-101 | SENet-154 | ViT | GC ViT |
> | :- | :- |:- | :- | :- | :- | :- | :- |
> | HSJA | 5538 |5702 | 3325 | 3785 | 6235 | 3556 | 5661 |
> | TA |  5614 |5628 | 3423 | 3863 | 6308 | 3575 | 5994 |
> | G-TA |  5530 |5588 | 3369 | 3820 | 4323 | 3519 | 5861 |
> | GeoDA | 3541 |3734 | 3024 | 3322 | 4277 | 3078 | 6914 |
> | Evolutionary | 5238 |5749 | 4094 | 4575 | 4956 | 4840 | 6223 |
> | SurFree | 5169 |5952 | 3907 | 4583 | 5040 | 3821 | 6495 |
> | Triangle Attack | 5985 |6326 | 4455 | 4272 | 5714 | 3783 | 8381 |
> | SVM-OPT | 4473 |4983 | 3619 | 4088 | 4443 | 3636 | 5551 |
> | Sign-OPT  | 4491 |4953 | 3541 | 3921 | 4373 | 3533 | 5452 |
> | SQBA  | 2265 |2700 | 898 | 1643 | 2216 | 2747 | 3896 |
> | BBA | 5177 |5326 | 2894 | 4564 | 4687 | 5244 | 6078 |
> | $\text{Prior-Sign-OPT} _ {\text{IncResV2}/\text{ResNet50}}$ | 3612 |3952 | 2786 | 3319 | 4021 | 3851 | 5455 |
> | $\text{Prior-Sign-OPT} _ {\theta _ 0^\text{PGD} + \text{IncResV2}/\text{ResNet50}}$ | 1435 |1685 | **262** | 495 | 1118 | 2852 | 3259 |
> | $\text{Prior-OPT} _ {\text{IncResV2}/\text{ResNet50}}$ | 1880 |1999 | 1312 | 1731 | 2311 | 3468 | 4312 |
> | $\text{Prior-OPT} _ {\theta _ 0^\text{PGD} + \text{IncResV2}/\text{ResNet50}}$ | **1148** |**1357**| 274 | **463** | **1092** | **2906** | **3207** |

---

> ### Author Response · Authors · 2024-11-21
> **Response to Reviewer hzgv (pt 2/2)**
>
> Also, we report the attack success rate (ASR) under the specific query budget, which is defined as the percentage of samples with distortions below a threshold $\epsilon$. In $\ell_2$ norm attacks, we set the threshold $\epsilon = \sqrt{0.001 \times d}$ on the ImageNet dataset, where $d$ is the image dimension.
>
> *Table 5: Attack Success Rate (ASR) under different query budgets of untargeted attacks against **Inception-V4** on the **ImageNet** dataset.*
> | Method | @1K | @3K | @5K | @7K |  @8K |  @10K |
> | :- | :- |:- | :- | :- | :- | :- |
> | HSJA | 3.4% |26.4% | 48.9% | 62.0% | 69.3% | 82.1% |
> | TA | 3.3% |26.8% | 48.4% | 64.3% | 72.1% | 82.0% |
> | G-TA | 3.3% |27.4% | 49.9% | 64.8% | 71.1% | 83.3% |
> | GeoDA | 27.2% |56.6% | 74.4% | 82.1% | 82.1% | 87.7% |
> | Evolutionary | 2.1% |19.7% | 45.2% | 65.3% | 73.1% | 81.6% |
> | SurFree | 9.3% |27.9% | 43.1% | 56.3% | 61.5% | 69.7% |
> | Triangle Attack | 32.1% |36.0% | 38.4% | 40.6% | 41.4% | 42.5% |
> | SVM-OPT | 2.0% |29.5% | 57.5% | 75.7% | 81.7% | 89.0% |
> | Sign-OPT  | 3.1% |29.7% | 57.9% | 75.8% | 81.9% | 89.8% |
> | $\text{SQBA} _ \text{IncResV2}$ | 41.9% |65.9% | 79.2% | 87.7% | 89.9% | 93.0% |
> | $\text{BBA} _ \text{IncResV2}$ | 16.3% |39.0% | 54.3% | 60.6% | 63.9% | 68.8% |
> | $\text{Prior-Sign-OPT} _ \text{IncResV2}$ | 3.8% | 38.9% | 75.7% | 89.4% | 91.8% | 96.3% |
> | $\text{Prior-OPT} _ \text{IncResV2}$ | 17.8% | **86.6%** | **96.4%** |  **98.4%** | **98.8%** | **99.1%** |
> | $\text{Prior-Sign-OPT} _ {\theta _ 0^\text{PGD} + \text{IncResV2}}$ | 62.6% | 77.8% | 88.3% |  93.4% | 94.5% | 96.9% |
> | $\text{Prior-OPT} _ {\theta _ 0^\text{PGD} + \text{IncResV2}}$ | **64.7%** | 84.1% | 92.5% |  96.1% | 96.7% | 98.1% |
>
>
> *Table 6: Attack Success Rate (ASR) under different query budgets of untargeted attacks against **ResNet-101** on the **ImageNet** dataset.*
> | Method | @1K | @3K | @5K | @7K |  @8K |  @10K |
> | :- | :- |:- | :- | :- | :- | :- |
> | HSJA | 12.5% |57.9% | 80.9% | 89.8% | 92.3% | 95.8% |
> | TA | 12.0% |55.7% | 78.9% | 89.2% | 91.3% | 95.9% |
> | G-TA | 11.6% |56.9% | 80.3% | 89.5% | 91.8% | 96.3% |
> | GeoDA | 33.9% |67.4% | 81.8% | 87.4% | 87.4% | 91.9% |
> | Evolutionary | 6.8% |41.3% | 71.0% | 85.4% | 89.5% | 93.2% |
> | SurFree | 20.2% |51.6% | 69.0% | 78.9% | 82.6% | 88.1% |
> | Triangle Attack | 50.9% |55.9% | 58.1% | 59.7% | 60.0% | 61.3% |
> | SVM-OPT | 7.0% |49.5% | 77.6% | 90.7% | 92.7% | 95.8% |
> | Sign-OPT  | 8.8% |51.0% | 76.9% | 90.5% | 92.7% | 95.9% |
> | $\text{SQBA} _ \text{ResNet50}$ | 78.7% |89.6% | 94.6% | 97.0% | 98.1% | 98.9% |
> | $\text{BBA} _ \text{ResNet50}$ | 47.6% |70.2% | 79.4% | 83.7% | 84.6% | 86.9% |
> | $\text{Prior-Sign-OPT} _ \text{ResNet50}$ | 10.9% | 65.0% | 91.1% | 95.9% | 97.3% | 99.4% |
> | $\text{Prior-OPT} _ \text{ResNet50}$ | 42.0% | 96.2% | **99.2%** |  **99.7%** | **99.7%** | **100.0%** |
> | $\text{Prior-Sign-OPT} _ {\theta _ 0^\text{PGD} + \text{ResNet50}}$ | 94.5% | **97.2%** | 98.2% |  99.0% | 99.3% | 99.7% |
> | $\text{Prior-OPT} _ {\theta _ 0^\text{PGD} + \text{ResNet50}}$ | **94.6%** | 96.9% | 98.3% |  98.7% | 99.0% | 99.5% |
>
>
> *Table 7: Attack Success Rate (ASR) under different query budgets of untargeted attacks against **GC ViT** on the **ImageNet** dataset.*
> | Method | @1K | @3K | @5K | @7K |  @8K |  @10K |
> | :- | :- |:- | :- | :- | :- | :- |
> | HSJA | 1.9% |24.3% | 49.6% | 65.8% | 72.3% | 82.8% |
> | TA | 2.2% |22.7% | 44.9% | 61.0% | 66.7% | 76.7% |
> | G-TA | 2.2% |22.9% | 48.7% | 62.9% | 69.2% | 77.6% |
> | GeoDA | 6.3% |22.7% | 39.0% | 46.2% | 46.2% | 54.3% |
> | Evolutionary | 1.3% |14.1% | 38.5% | 59.4% | 67.9% | 79.1% |
> | SurFree | 4.6% |21.7% | 37.1% | 50.1% | 56.7% | 65.4% |
> | Triangle Attack | 11.8% |15.2% | 17.5% | 19.0% | 19.7% | 21.2% |
> | SVM-OPT | 1.2% |21.0% | 51.8% | 69.7% | 75.8% | 83.9% |
> | Sign-OPT  | 1.4% |21.0% | 50.9% | 71.6% | 77.1% | 85.9% |
> | $\text{SQBA} _ \text{ResNet50}$ | 20.4% |49.2% | 69.1% | 80.5% | 84.4% | 89.2% |
> | $\text{BBA} _ \text{ResNet50}$ | 3.4% |26.7% | 43.5% | 57.0% | 60.2% | 67.5% |
> | $\text{Prior-Sign-OPT} _ \text{ResNet50}$ | 2.0% | 19.8% | 51.7% | 70.7% | 77.5% | 85.4% |
> | $\text{Prior-Sign-OPT} _ \text{ConViT}$ | 1.7% | 27.5% | 62.7% | 78.7% | 82.7% | 89.0%
> | $\text{Prior-OPT} _ \text{ResNet50}$ | 4.2% | 41.2% | 67.5% |  81.3% | 85.7% | 91.3% |
> | $\text{Prior-OPT} _ \text{ConViT}$ | 10.0% | **68.1%** | **87.9%** |  **92.4%** |**93.5%** | **95.7%** |
> | $\text{Prior-Sign-OPT} _ {\theta _ 0^\text{PGD} + \text{ResNet50}}$ | 32.4% | 56.2% | 74.2% |  85.1% | 89.0% | 92.6% |
> | $\text{Prior-OPT} _ {\theta _ 0^\text{PGD} + \text{ResNet50}}$ | **34.7%** | 57.7% | 74.8% |  85.0% | 88.7% | 92.8% |
> |

---

### Official Review · Reviewer_z4Ag · 2024-11-04

**Soundness:** 3
**Presentation:** 3
**Contribution:** 3
**Rating:** 6
**Confidence:** 3

**Summary:**

This paper introduces two novel hard-label attack methods, Prior-Sign-OPT and Prior-OPT, which leverage transfer-based priors and random sampling vectors to enhance the query efficiency and success rate of black-box adversarial attacks. Theoretical analysis and experiments on ImageNet and CIFAR-10 datasets demonstrate that these methods significantly outperform 11 state-of-the-art techniques while reducing the number of queries needed.

**Strengths:**

1. This paper proposes a gradient estimation method to conduct hard-label attack and provides proof for it.

2. The paper is well-written.

**Weaknesses:**

1. The paper benefit from showing the time complexity.
2. This paper is a comprehensive study based on transfer and query, hence, I hope you could compare it with the results from the paper "Blackbox Attacks via Surrogate Ensemble Search. Neurips 2022."

**Questions:**

Please see the weakness

---

> ### Author Response · Authors · 2024-11-21
> **Response to Reviewer z4Ag**
>
> We would like to thank the reviewer for the positive comments, and we shall carefully revise the paper following your suggestions.
>
> **Q1:** The paper benefit from showing the time complexity.
>
> **A1:** The primary additional computational cost of Prior-OPT compared to Sign-OPT arises from the binary search procedure performed on the priors during gradient estimation. Let $d$ represent the dimension of the input image, $q$ the number of vectors used in gradient estimation, and $f(d)$ the inference time of the target model for an input of dimension $d$. The time complexity of gradient estimation in Sign-OPT is $O(q \cdot f(d))$.
> In Prior-OPT, $s$ priors are introduced. Each prior requires a binary search procedure, which involves approximately $k$ inference steps. While $k$ may vary slightly depending on the specific prior or the input configuration, its value generally remains bounded and logarithmic in scale, given the nature of binary search. Consequently, the time complexity of the gradient estimation step of Prior-OPT can be expressed as:
>
> $$O((q-s+(s+1)\cdot k)\cdot f(d) + s \cdot \hat{f}(d)).$$
>
> When $q$ is large, $s$ and $k$ is relatively small (*i.e.,* the number of priors is small, and $k$ typically ranges in the tens), $s \cdot \hat{f}(d)$ denotes the time of taking $s$ priors, the additional overhead introduced by Prior-OPT is limited compared to Sign-OPT. While Prior-OPT introduces extra computation due to the binary search procedure on the priors, the increase in time complexity is relatively modest, especially when $s$ remains much smaller than $q$. This demonstrates that Prior-OPT achieves a balance between computational efficiency and improved gradient estimation. We add this analysis in Appendix F.1.
>
> **Table 12 in Appendix F.1** of our paper presented the time required to attack a single image using 10,000 queries, measured in seconds on a NVIDIA Tesla V100 GPU. For further details, please refer to Appendix F.1. Below, we paste the summary of time overhead results from Appendix F.1.
>
> *Table 1: The time consumptions of attacking one image with 10000 queries, which are measured by `seconds` on a NVIDIA Tesla V100 GPU.*
> | Method / Target Model | ResNet-101 | SENet-154 | ResNeXt-101 | GC ViT | Swin Transformer |
> | :--   | :-: |  :-: | :-: |  :-: |  :-: |
> | Sign-OPT | 112 | 197 | 91  | 131 | 88|
> | SVM-OPT | 119 | 189 | 102  | 158 | 98|
> | $\text{Prior-Sign-OPT} _ \text{ResNet50}$ | 240 | 372 | 195 | 203 | 183 |
> | $\text{Prior-OPT} _ \text{ResNet50}$ | 342 | 476 | 321 | 357 | 203 |
>
>
> ***
>
> **Q2:** This paper conducts a comprehensive study on transfer and query attacks. I suggest comparing your results with those from "Blackbox Attacks via Surrogate Ensemble Search (BASES)" (NeurIPS 2022).
>
> **A2:** Thank you for your suggestion. While BASES primarily employs a score-based attack method, and it is not a typical hard-label attack. It leverages a set of surrogate models to generate adversarial examples, which are then transferred to hard-label models.
>
> We perform untargeted and targeted attack experiments with BASES on 100 images from the ImageNet dataset. The attack success rates (ASR) are summarized in the following tables, where the surrogate model set includes ResNet-50, SENet-154, ResNeXt-101 (64 × 4d), VGG-13, SqueezeNet v1.1 for attacking against ResNet-101, and ResNet-50, ConViT, CrossViT, MaxViT, ViT for attacking against GC ViT and Swin Transformer. The victim surrogate model is VGG-19 in BASES.
>
> *Table 2: The attack success rate of **untargeted attack** of 100 images on the ImageNet dataset.*
> | Method/Target Model | ResNet-101 | GC ViT | Swin Transformer |
> | :- | :- |:- | :- |
> | Sign-OPT |98.0% | 87.0% | 70.0% |
> | $\text{{BASES}} _ \text{{5 surrogates + VGG19}}$ | 93.0% | 70.0% | 56.0% |
> | $\text{{Prior-Sign-OPT}} _ \text{{5 priors}}$ | **100.0%** | 96.9% | 95.9% |
> | $\text{{Prior-OPT}} _ \text{{5 priors}}$ | **100.0%** | **100.0%** | **99.0%** |
> | $\text{{Prior-Sign-OPT}} _ {{\theta _ 0^\text{{PGD}} + \text{{5 priors}}}}$ | **100.0%** | 98.0% | 97.0% |
> | $\text{{Prior-OPT}} _ {{\theta _ 0^\text{{PGD}}} + \text{{5 priors}}}$ | **100.0%** | **100.0%** | 97.0% |
>
> Interestingly, our approach can integrate BASES as an initial sample during initialization, denoted as **$\text{{Prior-Sign-OPT}}_\text{{5 priors}}$ + BASES**, as shown the following table, further enhancing performance.
>
>
> *Table 3: The attack success rate of **targeted attack** of 100 images on the ImageNet dataset.*
> | Method/Target Model | ResNet-101 | GC ViT | Swin Transformer |
> | :- | :- |:- | :- |
> | Sign-OPT | 69.0% | 51.0% | 40.0% |
> | $\text{{BASES}} _ \text{{5 surrogates + VGG19}}$ |82.0% | 60.0% | 48.0% |
> | $\text{{Prior-Sign-OPT}} _ \text{{5 priors}}$ |  88.0% | 69.0% | 53.0%  |
> | $\text{{Prior-OPT}} _ \text{{5 priors}}$ |  77.0% | 59.0% | 52.0% |
> | $\text{{Prior-Sign-OPT}}_\text{{5 priors}}$ + BASES | **92.0%** | **81.4%** | **61.9%** |
> | $\text{{Prior-OPT}}_\text{{5 priors}}$ +BASES| 86.0% | 73.2% | 57.7% |

---

> ### Comment · Reviewer_z4Ag · 2024-11-24
>
> Thanks for your response. I have updated the score to 6.

---

> ### Author Response · Authors · 2024-11-25
> **Thank you for your feedback.**
>
> Thank you for acknowledging the value of our hard-label attack and for increasing your score!
>
> We have updated the latest experimental results for **"Prior-Sign-OPT + BASES"** in the table provided in our previous response. By integrating BASES as a plugin to generate an initial sample during attack initialization, our approach further enhances performance. Notably, Prior-Sign-OPT + BASES achieves the highest attack success rate among all methods. Please see the updated table for details.

---

### Official Review · Reviewer_DsPD · 2024-11-05

**Soundness:** 3
**Presentation:** 2
**Contribution:** 3
**Rating:** 8
**Confidence:** 3

**Summary:**

The paper addresses the problem of hard-label black-box adversarial attacks. The paper builds on an existing approach that seeks the optimal ray direction from a benign image to minimize the distance to the adversarial region. While this method has shown promise, its high query cost remains a major drawback. A variation of this method that uses ‘signed gradient’ suffers from gradient accuracy limitation. To address these issues, the authors propose incorporating a transfer-based prior to reduce the query burden and improve performance. Additionally, they provide theoretical analysis of the gradient estimation quality achieved through this transfer-based prior and provide insight into its effectiveness.

**Strengths:**

**Originality**: The concept of using transfer priors is not entirely new, as it was demonstrated in other works e.g., Dong et al. (2022). However, this paper's novelty lies in applying them to the more practical hard-label setting, where information is limited.

**Quality**: The methodology demonstrates technical soundness, particularly the idea of using transfer-based priors and then the approach to gradient approximation through closest projection on the subspace.

**Clarity**: The writing could be further improved. Since the paper is based on previous work - Sign-OPT, it can be confusing for someone who is not well familiar with those works. The figures also make it hard to understand the key idea of the paper. In particular, the first figure is confusing as it has a lot of information which is unnecessary and does not motivate the problem or explain the idea well.

**Significance**: The paper makes substantial contributions to adversarial machine learning by introducing a novel attack that leverages transfer-based priors in hard-label black-box setting. The paper demonstrates how this additional information from surrogate models can be effectively used.

**Weaknesses:**

1. The main contribution of the paper seems to have limited novelty. It essentially boils down to employing Eq (7), which is similar to the Sign-OPT method but with the addition of using gradients from surrogate classifiers.
2. It seems that the results don't clearly favor one approach over another. In some cases, Prior-Sign-OPT performs well, while in others, Prior-OPT outperforms it. Lines 319-321 mention that Prior-OPT outperforms Sign-OPT under certain conditions, but these conditions are not clearly explained, so the reasoning is unclear.
3. Another concern is the inconsistent performance in targeted attacks. For instance, in Figures 10(g) and 10(h), the proposed methods are not among the top three, throughout the plots. In other cases, other methods perform equally well (especially for smaller number of queries), yet no explanation is provided for these discrepancies.
4. The method underperforms when the number of queries is small. For example, in Table 1, for the ViT experiments, other methods achieve lower mean L2 norm distortions with fewer queries. Specifically, for targeted attacks up to 5k queries, other methods generate adversarial examples with smaller perturbations as compared to the proposed methods.

**Questions:**

1. **Lines 196-197**: The notation for qi (used for the transfer-based prior) and q (representing the total number of vectors) is confusing. It would help to revise the notation to make the distinction clearer and avoid confusion.

2. **Figure 1**: The figure is unclear and lacks proper labeling. Please add clear labels to define the regions and ensure the equations are more readable and understandable

3. **Figures 9 & 10, and Table 1**: These results highlight that using PGD (Projected Gradient Descent) significantly boosts performance. The question arises: How would the other methods perform if PGD were applied to them as well? Were there any experiments done to investigate that?

4. The results indicate that even adding a single transfer-based prior improves performance. However, it would be valuable to test an alternative where, instead of using random vectors, only prior vectors are used. Are there any experiments that investigate this scenario?

---

> ### Author Response · Authors · 2024-11-21
> **Response to Reviewer DsPD (pt 1/4)**
>
> We would like to thank the reviewer for the positive comments, and we shall carefully revise the paper following your suggestions.
>
> **Q1:** Is the main contribution of the paper a limited novelty, essentially applying Eq (7), similar to the Sign-OPT method, with the added use of gradients from surrogate classifiers?
>
> **A1:**  Compared to the original Sign-OPT, Prior-Sign-OPT is based on the orthogonal vectors constructed through Gram-Schmidt orthogonalization.
> It approximates the subspace projection to achieve a more accurate gradient estimation, which optimizes its query efficiency.
> Prior-Sign-OPT represents an initial step in the development of Prior-OPT, serving as a preliminary stage rather than the final algorithm.
> Notably, the gradient estimation formula of Prior-OPT (Eq. (13)) is fundamentally different from that of Sign-OPT (Eq. (8)).
>
> Additionally, the effectiveness of Prior-OPT and Prior-Sign-OPT is rigorously demonstrated through extensive theoretical analysis and comprehensive experimental validation.
>
> To the best of our knowledge, this is the first work to derive the expected cosine similarity between estimators of the *OPT family* (*i.e.,* Sign-OPT, Prior-Sign-OPT, Prior-OPT) and the true gradient, providing a theoretical guarantee for performance improvement.
>
> ***
>
> **Q2:**  Lines 319-321 mention that Prior-OPT outperforms Sign-OPT under certain conditions, what are the conditions under which Prior-OPT surpasses Sign-OPT?
>
> **A2:** The effectiveness of Prior-OPT is analyzed through the expected squared  cosine similarity $\mathbb{E}[\gamma^2]$ between the estimated gradient and the true gradient.
>
> For Sign-OPT, $\mathbb{E}[\gamma^2] = \frac{1}{d}\left(\frac{2}{\pi}(q-1)+1\right)$.
>
> For Prior-OPT, $\mathbb{E}[\gamma^2] = \sum_{i=1}^{s} \alpha_i^2+\frac{1}{d-s}\left(\frac{2}{\pi}(q-s-1)+1\right)\left(1-\sum_{i=1}^{s} \alpha_i^2\right)$, where $\alpha_i$ is the cosine similarity between the $i$-th prior and the true gradient, $s$ is the number of priors, $q$ is the number of number of queries (also the number of sampled vectors), and $d$ is the dimension of the input image.
>
> **Conditions for Prior-OPT to Outperform Sign-OPT:**
>
> Prior-OPT surpasses Sign-OPT when the following inequality holds:
>
> $$\sum _ {i=1} ^ s \alpha _ i^2  > \frac{\frac{1}{d} \left(\frac{2}{\pi}(q-1)+1\right) - \frac{1}{d-s}\left(\frac{2}{\pi}(q-s-1)+1\right)}{1-\frac{1}{d-s}\left( \frac{2}{\pi} (q-s-1)+1\right)}$$
>
> To simplify this inequality, under the assumption that $s \ll q \ll d$, the right-hand side approximates to $\frac{2s}{\pi d}$. Thus, the condition simplifies to:
>
> $$\sum _ {i=1} ^ s \alpha _ i^2 > \frac{2s}{\pi d}$$
>
> Equivalently, the average squared cosine similarity must satisfy:
>
>  $$\overline{\alpha^2}>\frac{2}{\pi d}$$
>
> Since $\frac{2}{\pi d}$ is typically a very small value due to the large input dimension $d$, this threshold is relatively easy to satisfy.
> Therefore, Prior-OPT generally outperforms Sign-OPT when the priors have even a minimal level of informativeness (non-zero $\alpha_i$).
>
> The detailed derivation steps of above conclusion is in Appendix D of our paper.
>
> ***
>
> **Q3:** Why is the performance inconsistent in targeted attacks? For example, in Figures 10(g) and 10(h), the proposed methods are not among the top three, and in some cases, other methods perform equally well with fewer queries.
>
> **A3:** Our method is an improvement based on Sign-OPT, which still significantly outperforms Sign-OPT in Fig. 10(g) and Fig. 10(h). However, since Sign-OPT performs considerably worse than the best methods in targeted attacks against GC ViT and Swin Transformer, our method also falls short of surpassing the best methods in these specific cases. Nevertheless, our method demonstrates superior performance in most other scenarios, as shown in Fig. 10(a) to Fig. 10(f).
>
> In targeted attacks, the gradient direction must specifically point toward a narrow adversarial region corresponding to the target class. This constraint tends to reduce the cosine similarity $\alpha_i$, often leading to $\alpha_i \approx 0$ for $i\in[1,s]$.
> In such cases, when $s\ll q$, the expected squared cosine similarity $\mathbb{E}[\gamma^2]$ for both Sign-OPT and Prior-OPT becomes approximately equal, reducing the advantage of using priors. Consequently, the performance of Prior-OPT converges to that of Sign-OPT in targeted attack scenarios.
>
> Furthermore, in targeted attacks, Prior-Sign-OPT employs a more query-efficient approach by leveraging the sign of directional derivatives, requiring only a single query per sampled vector. In contrast, Prior-OPT involves multiple binary search steps (as described in Eq. (13)), which significantly increases query costs. This efficiency allows Prior-Sign-OPT to maintain higher performance under limited query budgets in targeted attacks.
>
> Another possible explanation is the increased complexity of the optimization process in targeted attacks. This aspect warrants further investigation in future work.
>
> ***

---

> > ### Comment · Reviewer_DsPD · 2024-11-27
> >
> > ## Response to A1:
> > I disagree that the Sign-OPT and Prior-OPT have ‘fundamentally’ different formulations.
> >
> > 1. Eq (4) in Sign-OPT paper and Prior-Sign-OPT method from Eq (7) in this paper are the same with the exception of adding prior vectors.
> > 2. Similarly, Prior-OPT from Eq (13) in this paper and OPT attack (Cheng 2019a - Algorithm 2, line 5) are the same with the exception of adding prior vectors.
> >
> > The impact of orthogonalization is minimal unless the dimension d is small.
> >
> > Regarding the orthogonalization, it will not make a huge difference unless the dimension d is low. In high dimensions, a small number of randomly sampled vectors (<<d) are approximately orthogonal, reducing the need for explicit orthogonalization.
> >
> > Therefore, I still believe the primary contribution lies in 1) incorporating priors into the original OPT or Sign-OPT framework, 2) supported by theoretical proof that priors enhance performance.
> >
> > ## Response to A2:
> > Thank you for the clarification. I have reviewed the detailed proof in the appendix, and it is clear.
> >
> > ## Response to A3:
> > So this is one possible limitation of the proposed method. Please include this explanation in the appendix to inform readers about the limitation.

---

> ### Author Response · Authors · 2024-11-21
> **Response to Reviewer DsPD (pt 2/4)**
>
> ---
>
> **Q4:** The method underperforms when the number of queries is small. For example, in Table 1, for the ViT experiments, other methods achieve lower mean $\ell_2$ norm distortions with fewer queries. Specifically, for targeted attacks up to 5k queries, other methods generate adversarial examples with smaller perturbations as compared to the proposed methods.
>
> **A4:** The observed underperformance of our method is highlighted in Table 1 for the ViT's targeted attack experiments under low query budgets. The limitation arises from a trade-off inherent in our approach. Specifically, our method prioritizes gradient estimation accuracy through a prior-based mechanism, which becomes increasingly effective as the number of queries grows. However, when the query budget is small, the priors require sufficient queries during the binary search phase to influence the attack's query efficiency. This limitation is particularly pronounced in targeted attacks, where the accuracy of priors is significantly reduced, resulting in performance closer to that of Sign-OPT (as discussed in **A3**). Another possible reason is the complexity of optimization, and we will investigate this reason in the future work.
>
> To address this issue, we propose integrating a **dimension reduction step** during the sampling phase of gradient estimation, inspired by the QEBA approach. This modification is straightforward, requiring only minor changes in the sampling process. Specifically, the method involves initially sampling random vectors in a reduced-dimensional space, followed by resizing these vectors to match the original input dimension. Preliminary experimental results show that this technique can reduce the distortion by approximately half in both untargeted and targeted attacks.
>
> Notably, dimension reduction techniques are also employed in other state-of-the-art methods, such as AHA, SurFree, and Triangle Attack, demonstrating their effectiveness in improving performance in low-query settings. We believe that incorporating this enhancement into our method will enable it to achieve competitive, if not superior, performance across all query budgets. We plan to explore this approach in greater detail as part of our future work.
>
> ---
>
> **Q5:** Lines 196-197: The notation for $\mathbf{q}_i$ (used for the transfer-based prior) and $q$ (representing the total number of vectors) is confusing. It would help to revise the notation to make the distinction clearer and avoid confusion.
>
> **A5:** Thank you for pointing this out. The fonts of $\mathbf{q}_i$ and $q$ are indeed different. Specifically, $\mathbf{q}_i$ is in bold font, typically representing a vector, and in our case, it is used to denote transfer-based priors. In contrast, $q$ is in normal font, representing a scalar. To improve clarity and avoid confusion in notation, we will revise our paper to replace $q$ with $n$.
>
> ---
>
> **Q6:** Figure 1: The figure is unclear and lacks proper labeling. Please add clear labels to define the regions and ensure the equations are more readable and understandable.
>
> **A6:** Thank you for your feedback. We have revised the paper to include clearer labels in Figure 1, making the regions and equations more readable and understandable. Please refer to the updated PDF file for the revised figure.
>
> ---

---

> > ### Comment · Reviewer_DsPD · 2024-11-27
> >
> > ## Response to A4:
> > It is unclear to me how the trade-off is inherent in the method. The proposed method achieves a high attack accuracy but requires large L2 norm for low query budgets. Why is that? For higher query budgets, other methods perform comparably (e.g., untargeted SQBA in Fig. 9(a) at 10k queries, Sign-OPT in Fig. 10(f), and HSJA in Fig. 10(h)). As mentioned in response to A3, I suggest explicitly and clearly discussing this trade-off in a ‘Limitations’ section.
> >
> > ## Response to A6:
> > The figure remains unclear. I would suggest to label points or areas with single letters and include a legend explaining each label. This will better convey the key idea of the method.

---

> ### Author Response · Authors · 2024-11-21
> **Response to Reviewer DsPD (pt 3/4)**
>
> **Q7:** Figures 9 & 10, and Table 1: These results highlight that using PGD (Projected Gradient Descent) significantly boosts performance. The question arises: How would the other methods perform if PGD were applied to them as well? Were there any experiments done to investigate that?
>
> **A7:** We conduct additional experiments on 100 tested images to investigate the impact of using the PGD attack on a surrogate model to generate initial adversarial examples for enhancing the performance of SQBA and BBA attacks. The results of these experiments are presented below.
>
> *Table 1: The mean distortion across 100 images under different query budgets of attacking against **GC ViT** on the **ImageNet** dataset.*
> | Method | @1K | @3K | @5K | @7K |  @8K |  @10K |
> | :- | :- |:- | :- | :- | :- | :- |
> | $\text{SQBA}_\text{ConViT}$ | 20.451 |11.417 | 7.994 | 6.284 | 5.710 | 4.845 |
> | $\text{SQBA(PGD)}_\text{ConViT}$ | 16.411 |9.787 | 7.297 | 5.922 | 5.441 | 4.732 |
> | $\text{BBA}_\text{ConViT}$ | 31.198 |17.674 | 13.546 | 11.266 | 10.477 | 9.302 |
> | $\text{BBA(PGD)}_\text{ConViT}$ | 16.578 |11.402 | 9.551 | 8.465 | 8.048 | 7.388 |
> | $\text{Prior-Sign-OPT}_\text{ConViT}$ | 56.526 | 22.944 | 13.500 | 9.393 | 8.122 | 6.407 |
> | $\text{Prior-Sign-OPT(PGD)}_\text{ConViT}$ | 16.154 | 8.591 | 6.155 | **4.829** | **4.325** | **3.688** |
> | $\text{Prior-OPT} _ \text{ConViT}$ | 37.896 | 11.615 | 7.620 |  5.924 | 5.309 | 4.531 |
> | $\text{Prior-OPT(PGD)} _ \text{ConViT}$ | **15.773** | **8.456** | **6.050** |  4.856 | 4.460 | 3.896 |
>
> *Table 2: The mean distortion across 100 images under different query budgets of attacking against **ResNeXT-101(64x4d)** on the **ImageNet** dataset.*
> | Method | @1K | @3K | @5K | @7K |  @8K |  @10K |
> | :- | :- |:- | :- | :- | :- | :- |
> | $\text{SQBA} _ \text{ResNet50}$ | 14.356 |8.740 | 6.630 | 5.499 | 5.096 | 4.506 |
> | $\text{SQBA(PGD)} _ \text{ResNet50}$ | 11.636 |7.229 | 5.507 | 4.618 | 4.300 | 3.795 |
> | $\text{BBA} _ \text{ResNet50}$ | 22.887 |14.473 | 11.833 | 10.529 | 10.058 | 9.329 |
> | $\text{BBA(PGD)} _ \text{ResNet50}$ | 9.458 |7.046 | 6.147 | 5.653 | 5.466 | 5.196 |
> | $\text{Prior-Sign-OPT} _ \text{ResNet50}$ | 45.631 | 14.033 | 7.557 | 5.204 | 4.549 | 3.664 |
> | $\text{Prior-Sign-OPT(PGD)} _ \text{ResNet50}$ | 7.584 | 4.841 | 3.748 | 3.125 | 2.903 | **2.586** |
> | $\text{Prior-OPT} _ \text{ResNet50}$ | 24.573 | 6.214 | 4.313 |  3.494 | 3.242 | 2.894 |
> | $\text{Prior-OPT(PGD)} _ \text{ResNet50}$ | **6.989** | **4.459** | **3.526** |  **3.030** | **2.866** |2.601|
>
>
>
> *Table 3: The mean distortion across 100 images under different query budgets of attacking against **Inception-V3** on the **ImageNet** dataset.*
> | Method | @1K | @3K | @5K | @7K |  @8K |  @10K |
> | :- | :- |:- | :- | :- | :- | :- |
> | $\text{SQBA} _ \text{IncResV2}$ | 22.885 |14.193 | 10.969 | 9.214 | 8.547 | 7.647 |
> | $\text{SQBA(PGD)} _ \text{IncResV2}$ | 21.327 |13.430 | 10.377 | 8.717 | 8.124 | 7.229 |
> | $\text{BBA} _ \text{IncResV2}$ | 35.944 |23.146 | 18.862 | 16.705 | 16.014 | 14.848 |
> | $\text{BBA(PGD)} _ \text{IncResV2}$ | 23.468 |16.996 | 14.488 | 13.081 | 12.582 | 11.843 |
> | $\text{Prior-Sign-OPT} _ \text{IncResV2}$ | 72.029 | 22.190 | 11.126 | 7.595 | 6.642 | 5.390 |
> | $\text{Prior-Sign-OPT(PGD)} _ \text{IncResV2}$ | 21.040 | 11.098 | 8.039 | 6.433 | 5.876 | 5.073 |
> | $\text{Prior-OPT} _ \text{IncResV2}$ | 43.531 | **8.841** | **5.927** |  **4.935** | **4.688** | **4.252** |
> | $\text{Prior-OPT(PGD)} _ \text{IncResV2}$ | **19.547** | 9.444 | 7.245 |  6.212 | 5.845 | 5.245 |
>
> *Table 4: The mean distortion across 100 images under different query budgets of attacking against **Inception-V4** on the **ImageNet** dataset.*
> | Method | @1K | @3K | @5K | @7K |  @8K |  @10K |
> | :- | :- |:- | :- | :- | :- | :- |
> | $\text{SQBA} _ \text{IncResV2}$ | 26.833 |15.797 | 11.687 | 9.612 | 8.893 | 7.850 |
> | $\text{SQBA(PGD)} _ \text{IncResV2}$ | 23.004 |14.442 | 10.995 | 9.152 | 8.512 | 7.542 |
> | $\text{BBA} _ \text{IncResV2}$ | 39.044 |23.368 | 18.470 | 16.287 | 15.557 | 14.451 |
> | $\text{BBA(PGD)} _ \text{IncResV2}$ | 27.811 |18.466 | 14.957 | 13.266 | 12.616 | 11.756 |
> | $\text{Prior-Sign-OPT} _ \text{IncResV2}$ | 79.743 | 25.845 | 13.218 | 8.795 | 7.634 | 6.036 |
> | $\text{Prior-Sign-OPT(PGD)} _ \text{IncResV2}$ | 23.780 | 11.962 | 8.531 | 6.721 | 6.105 | 5.210 |
> | $\text{Prior-OPT} _ \text{IncResV2}$ | 49.645 | 11.024 | **6.234** |  **5.162** | **4.845** | **4.392** |
> | $\text{Prior-OPT(PGD)} _ \text{IncResV2}$ | **22.946** | **9.956** | 7.251 |  6.048 | 5.654 | 4.996 |
>
> From the results above, we conclude that while PGD initialization enhances all methods, our Prior-OPT and Prior-Sign-OPT achieve significantly lower distortions, thereby delivering the best performance among all attacks.

---

> ### Author Response · Authors · 2024-11-21
> **Response to Reviewer DsPD (pt 4/4)**
>
> **Q8:** The results show that adding a single transfer-based prior improves performance. Would it be useful to test using only prior vectors instead of all random vectors?
>
> **A8:** If all random vectors are eliminated in gradient estimation, the gradient estimator's performance lacks a lower bound, making it unable to guarantee accuracy in the worst-case scenario. However, when random vectors are included in the gradient estimation, the accuracy of the estimator is ensured to have a lower bound. This means that, regardless of how poor the priors are, the estimator maintains a guaranteed minimum level of performance in the worst case.
>
> This can be verified through the formulas for $\mathbb{E}[\gamma]$ and $\mathbb{E}[\gamma^2]$ derived for Prior-Sign-OPT and Prior-OPT.
>
> Specifically, in **Prior-Sign-OPT**, $\mathbb{E}[\gamma]$ is given by:
>
> $\mathbb{E}[\gamma] = \frac{1}{\sqrt{q}}\left[\sum_{i=1}^s|\alpha_i|+(q-s)\sqrt{1-\sum_{i=1}^s \alpha_i^2} \cdot \frac{\Gamma(\frac{d-s}{2})}{\Gamma(\frac{d-s+1}{2})\sqrt{\pi}}\right]$, where $\alpha_i$ denotes the cosine similarity between the $i$-th prior and the true gradient, $s$ is the number of priors, and $q$ is the total number of vectors. When **all random vectors are removed in Prior-Sign-OPT**, we set $q = s$, and the formula reduces to:
>
> $$\mathbb{E}[\gamma] = \frac{1}{\sqrt{q}}\left[\sum_{i=1}^s|\alpha_i|\right].$$
>
> In this case, $\mathbb{E}[\gamma]$ depends solely on $\alpha_i$, which reflects the accuracy of the priors. If $\alpha_i$ is extremely low, the accuracy of the estimated gradient degrades significantly.
> Similarly, when we **remove all random vectors in Prior-OPT** and set $q=s$, the formula of $\mathbb{E}[\gamma]$ reduces to:
>
> $$ \mathbb{E}[\gamma] \ge \sqrt{\sum _ {i=1}^s \alpha _ i^2}.$$
>
> This demonstrates that, without random vectors, the gradient estimation is entirely reliant on the quality of the priors (*i.e.,* the $\alpha_i$ value), and poor priors can result in arbitrarily poor performance. Conversely, when random vectors are included, the formula involving random vectors ensures a lower bound for $\mathbb{E}[\gamma]$.
> This lower bound can be derived by setting $\alpha_i=0$ in the above formula.
> For **Prior-Sign-OPT**, the lower bounds of $\mathbb{E}[\gamma]$ and $\mathbb{E}[\gamma^2]$ are
>
> $$ \mathbb{E}[\gamma] \ge \frac{q-s}{\sqrt{q}} \cdot \frac{\Gamma(\frac{d-s}{2})}{\Gamma(\frac{d-s+1}{2})\sqrt{\pi}},$$
>
> $$ \mathbb{E}[\gamma^2] \ge \frac{1}{q}\left(\frac{q-s}{d-s} \left( \frac{2}{\pi} (q-s-1)+1\right)\right).$$
>
> For **Prior-OPT**, the lower bounds of $\mathbb{E}[\gamma]$ and $\mathbb{E}[\gamma^2]$ are
>
> $$ \mathbb{E}[\gamma] \ge \sqrt{\frac{q-s}{\pi}} \cdot \frac{\Gamma(\frac{d-s}{2})}{\Gamma(\frac{d-s+1}{2})},$$
>
> $$ \mathbb{E}[\gamma^2] \ge \frac{1}{d-s}\left(\frac{2}{\pi} (q-s+1)+1 \right),$$
>
> thereby providing robustness even when the priors are of low quality.
>
> Furthermore, in Prior-OPT's gradient estimation, each prior requires a binary search, whereas random vectors do not. Random vectors need only a single query per vector, making them more efficient in this regard.
>
> We present **targeted attack results on 100 images** using *only priors* with the Prior-Sign-OPT and Prior-OPT algorithms, referred to as **"Pure-Prior-Sign-OPT"** and **"Pure-Prior-OPT"**, respectively, in the following tables.
>
> *Table 5: The results of **targeted attacks** against **GC ViT** on the **ImageNet** dataset.*
> | Method | @1K | @3K | @5K | @7K |  @8K |  @10K | @12K | @15K | @18K | @20K |
> | :- | :- |:- | :- | :- | :- | :- | :- | :- | :- | :- |
> | Sign-OPT  | 53.883 |36.842 | 29.111 | 24.948 | 23.456 | 21.142 | 19.557 | 17.633 | 16.204 | 15.441 |
> | $\text{Pure-Prior-Sign-OPT} _ \text{ResNet50}$ | 53.975 | 52.703 | 52.702 | 52.702 | 52.702 | 52.702 |  52.702 |  52.702 |  52.702 |  52.702 |
> | $\text{Pure-Prior-Sign-OPT} _ \text{ResNet50,ConViT}$ | **43.325** | 39.493 | 39.385 | 39.385 | 39.385 | 39.385 | 39.385 |  39.385 |  39.385 |  39.385 |
> | $\text{Pure-Prior-OPT} _ \text{ResNet50}$ | 54.909 | 54.746 | 54.746 | 54.746 | 54.746 | 54.746 |  54.746 |  54.746 |  54.746 |  54.746 |
> | $\text{Pure-Prior-OPT} _ \text{ResNet50,ConViT}$ | 44.424 | 40.739 | 40.708 | 40.708 | 40.708 | 40.708 |  40.708 |  40.708 |  40.708 |  40.708 |
> | $\text{Prior-Sign-OPT} _ \text{ResNet50,ConViT}$ **(Ours)** | 51.681 | 33.762 | 26.524 | 22.329 | 20.832 | **18.536** | **16.944** | **15.181** | **13.903** | **13.288** |
> | $\text{Prior-OPT} _ \text{ResNet50,ConViT}$  **(Ours)** | 49.075 | **30.903** | **24.592** |  **21.330** | **20.294** | 18.559 |  17.382 | 16.077 | 15.198 | 14.666 |
>
> Based on the above results, Pure-Prior-Sign-OPT and Pure-Prior-OPT fail to outperform Sign-OPT, which relies solely on random vectors without incorporating priors. Moreover, as the query budget increases, the distortion achieved by these methods decreases very slowly or remains nearly unchanged, highlighting their inefficiency in utilizing additional queries to improve attack effectiveness.

---

> > ### Comment · Reviewer_DsPD · 2024-11-27
> >
> > ## Response to A7:
> > Thank you for sharing. The addition is valuable, but it does not convey the whole picture. It would be ideal to include attack success rates. Please consider adding these results in the appendix for the final version of the paper to make the paper more comprehensive.
> >
> > ## Response to A8:
> > Thank you for the explanation. I understand the theoretical perspective presented in the paper. However, I have one question: how likely is it to encounter bad priors? I assume a bad prior would align with the gradient direction that increases the correct class confidence rather than decreasing it. This could have been clarified through experiments, but with only L2 norm information provided, it is difficult to draw conclusions.
> >
> > As mentioned in my response to A7, please include attack success rates in the final version to provide a more comprehensive evaluation.
> >
> > --
> > In Appendix D, line 1420, q is mentioned as the number of queries, while in the main paper, it is number of total vectors.

---

> > > ### Comment · Reviewer_DsPD · 2024-11-27
> > >
> > > Thank you authors for your detailed responses. While most of my concerns have been addressed, some questions remain, as noted in my responses. Addressing these would further strengthen the paper. There are some limitations to the method, and I would want them to be clearly outlined in the final version. Overall, the paper makes a good contribution and can be a valuable addition to the adversarial research community.

---

> ### Author Response · Authors · 2024-11-27
> **Further Explanation Regarding the Concern in A1**
>
> ### **Further Explanation Regarding the Concern in A1**
>
>
> **A1+:** Thank you for your thoughtful comments and for highlighting these points.
> We would like to provide additional explanation of **the differences between Prior-OPT and OPT** to address your concerns.
>
> **First,** the formulation of Prior-OPT (Eq. (13)) is *not identical* to OPT. In Eq. (13), the last term involves the $\ell_2$ normalization of $\mathbf{v} _ \perp$, where $\mathbf{v} _ \perp = \sum_{i=1}^{q-s} \text{sign}({g(\theta + \sigma \mathbf{u}_i) - g(\theta)}) \cdot \mathbf{u}_i$ and $\mathbf{u} _ {1},\cdots,\mathbf{u} _ {q-s}$ are orthogonal random vectors.
> As a result, Prior-OPT employs more precise finite difference estimations for the prior terms (the first term), while relying on sign-based estimation for the random vector components.
> This distinction arises because random vectors $(\mathbf{u} _ {1},\mathbf{u} _ {2},\cdots)$ have identical distributions in $d$-dimensional space, leading to relatively consistent cosine similarity with the true gradient.
> Given the approximation $\nabla g(\theta)^\mathsf{T} \mathbf{u} \approx \frac{g(\theta + \sigma \mathbf{u}) - g(\theta)}{\sigma}$, it is reasonable to replace $\nabla g(\theta)^\mathsf{T} \mathbf{u} _ i$ with $\text{sign}(\nabla g(\theta)^\mathsf{T} \mathbf{u} _ i)$ since magnitudes (absolute values) of these inner products do not vary significantly across random directions.
> As a result, the coefficients of random vectors can be set to uniform magnitudes ($+1$ or $-1$) to reduce the number of queries.
> In contrast, the cosine similarity between the prior direction $\mathbf{p}_i$ and the true gradient $\nabla g(\theta)$ is unknown and may differ significantly from that of the random directions. Thus, the coefficients for priors requires a more precise estimation, necessitating a separate binary search procedure.
> Therefore, Prior-OPT goes beyond a simple modification of OPT by directly adding priors, as it fundamentally handles priors and random directions differently to address these challenges.
>
> **Second,** regarding the orthogonalization in high-dimensional spaces, while we agree that a small number of randomly sampled vectors ($\ll d$) are approximately orthogonal, this is not always the case for *multiple priors*.
> Priors derived from potentially correlated models are *less likely* to be orthogonal to each other.
> If the Gram-Schmidt orthogonalization is removed, the gradient estimate obtained using Eq. (7) and Eq. (13) may become less accurate, potentially degrading performance. Furthermore, the expected formulas ($\mathbb{E}[\gamma]$ and $\mathbb{E}[\gamma^2]$) derived from our theoretical analysis would no longer hold in such scenarios.
>
> We appreciate your perspective that the main contribution lies in incorporating priors into the OPT or Sign-OPT framework, supported by theoretical proof. However, we believe these key differences in handling priors and random directions, along with the role of orthogonalization, contribute to the deeper understanding of our approach.
>
> ---
>
> About **A2**:
>
> **The condition under which Prior-OPT surpasses Sign-OPT, along with its derivation, have been verified.** *Thank you for the clarification. I have reviewed the detailed proof in the appendix, and it is clear.*
>
> ---
>
> ### **Response to Follow-Up Suggestion of A3**
>
> **A3+:** Thanks for your suggestion. Yes, we will include this explanation in Appendix in the final version.

---

> ### Author Response · Authors · 2024-11-27
> **Response to the Follow-Up Question/Suggestion of A4 and A6**
>
> ### **Response to the Follow-Up Question of A4**
>
> **Q4+:** **(1)** Why does the proposed approach achieve high attack accuracy but require a large mean $\ell _ 2$​-norm for low query budgets? **(2)** For higher query budgets, why do other methods perform comparably?
>
> **A4+:** **(1)** In targeted attacks on ViT, while priors become less effective for certain samples, they remain useful for a considerable proportion of samples. This contributes to the higher ASR of Prior-OPT in the early stages of attacks (low query budgets) (see Fig. 10(f)). However, for the remaining samples, the limited effectiveness of priors makes it more challenging for Prior-OPT to succeed. In contrast, Prior-Sign-OPT leverages a gradient estimation technique that requires fewer queries to achieve high ASR. This distinction explains why Prior-Sign-OPT outperforms Prior-OPT in the later stages of the attack (see Fig. 10(f)).
>
> It should be noted that in **the difference between the calculation methods of attack success rate (ASR) and mean $\ell _ 2$ distortion**. ASR represents the proportion of successfully attacked samples, whereas mean $\ell _ 2$ distortion is the average distortion across all samples.
> Our approach achieves a high ASR under low query budgets because it successfully attacks a large number of samples. However, a small subset of samples exhibits relatively high $\ell _ 2$ distortion, which increases the overall mean distortion.
>
> This is analogous to a group of students in a class where most students perform well and pass the exam (high ASR), but a few students with very low scores lower the class's average grade (high mean $\ell _ 2$ distortion).
>
> In fact, in the targeted attack on ViT shown in Table 1, the mean $\ell _ 2$ distortion of our $\text{Prior-Sign-OPT} _ \text{{ResNet50\\&ConViT}}$ is very close to that of the best-performing TA method: 53.925 vs 52.110 in 1K queries, 38.418 vs 36.455 in 2K queries, and 20.673 vs 20.536 in 5K queries. These differences are minimal.
>
>
> **(2)** This is because other methods converge significantly slower than our approach. For example, in Fig. 9(a), Prior-OPT converges much faster than other methods. When the query budget reaches its maximum (10K), SQBA gradually converges to a similar minimum distortion, as shown in Fig. 9(a).
>
> Thus, while some methods can eventually achieve a similar minimum distortion on certain models, their convergence rate remains considerably slower compared to our approach. Convergence rate is important because, in practice, query budgets are often limited, especially when only a very small number of queries is allowed.
>
> Additionally, in Fig. 9(a), $\text{{Prior-Sign-OPT}} _ \text{{ResNet50\\&ConViT}}$ is highly effective across both low and high query budgets, consistently achieving the lowest distortion due to its PGD-based initialization.
>
> We will incorporate these explanations into the final version of our paper.
>
> ---
>
> ### **Response to the Additional Suggestion in A6**
>
> **Q6+:** The figure remains unclear. I would suggest to label points or areas with single letters and include a legend explaining each label. This will better convey the key idea of the method.
>
> **A6+:** We will make a revision to our figure in the final version.

---

> ### Author Response · Authors · 2024-11-27
> **Response to Follow-Up Questions of A7, A8, and A9**
>
> ### **Response to the Follow-Up Question of A7**
>
> **Q7+:** The addition is valuable, but it does not convey the whole picture. It would be ideal to include attack success rates. Please consider adding these results in the appendix for the final version of the paper to make the paper more comprehensive.
>
> **A7+:** Thank you for your valuable suggestion to improve our paper. We will include the attack success rates of SQBA (PGD) and BBA (PGD) in the final revised version. Due to time constraints, the experiments are still ongoing. We will provide the results in the final version once the programs are fully completed.
>
> ---
>
> ### **Response to the Follow-Up Question of A8**
>
> **Q8+:**  How likely is it to encounter bad priors? I assume a bad prior would align with the gradient direction that increases the correct class confidence rather than decreasing it. This could have been clarified through experiments, but with only L2 norm information provided, it is difficult to draw conclusions.
>
> **A8+:** As you mentioned, the likelihood of encountering bad priors needs to be validated through experiments. However, even if bad priors are encountered, their impact is negligible due to the properties of the method, as demonstrated below.
>
> Let us consider the case where a "bad prior vector" is exactly the negative of the true gradient, which would increase the confidence of the correct class rather than decreasing it.
>
> For Prior-Sign-OPT, the prior-related term in the gradient estimation formula (Eq. (7)) is:
>
> $$\sum _ {i=1}^{s} \text{sign}(g(\theta + \sigma \mathbf{p} _ i) - g(\theta))\cdot \mathbf{p} _ i,$$
>
> and for Prior-OPT, the prior-related term in the gradient estimation formula (Eq. (13)) is:
>
> $$\sum _ {i=1}^{s} \frac{g(\theta + \sigma \mathbf{p} _ i) - g(\theta)}{\sigma} \cdot \mathbf{p} _ i.$$
>
> In both formulas, under the assumption of $\nabla g(\theta)^\mathsf{T} \mathbf{p} _ i \approx \frac{g(\theta + \sigma \mathbf{p} _ i) - g(\theta)}{\sigma}$, if we replace $\mathbf{p}_i$ with $-\mathbf{p}_i$, the final output remains unchanged.
> This result can also be verified through our theoretical analysis. In Eqs. (11), (12), (15), (16), and (17), the expectations $\mathbb{E}[\gamma]$ and $\mathbb{E}[\gamma^2]$ depend only on $|\alpha _ i|$ or $\alpha _ i^2$.
> Therefore, even if $\alpha_i$ changes to $-\alpha_i$, the expectations remains the same.
>
> ---
>
> **Q9:** In Appendix D, line 1420, $q$ is mentioned as the number of queries, while in the main paper, it is number of total vectors.
>
> **A9:** In our paper, $q$ consistently represents the total number of vectors used in gradient estimation. The reference to $q$ as the number of queries in line 1420 of Appendix D is a typo, and we sincerely apologize for the oversight. **This typo has been corrected in the updated version.** As mentioned in **A5**, to help readers clearly distinguish between $\mathbf{q}_i$ and $q$, we will replace $q$ with $n$ in the final version of the paper.
>
> ---
>
> Finally, we would like to sincerely thank the reviewer for your valuable and insightful comments, which have significantly contributed to enhancing the quality of our paper. We will carefully address and incorporate your suggestions into the final version.

---

### Author Response · Authors · 2024-11-25
**General Response to All the Reviewers**

We appreciate the reviewers for dedicating their time and effort to provide insightful and constructive feedback on our paper. In response, we have carefully addressed each comment in detail. Additionally, we would like to: (1) summarize the primary contributions acknowledged by reviewers, and (2) outline the improvements made in the revised manuscript.

**Key Contributions Acknowledged by Reviewers:**

- **Novelty and Originality:**  Hard-label attacks are among the most challenging tasks, as they rely solely on the top-1 predicted label, making the limited information a significant obstacle. The proposed approach is grounded in a solid theoretical foundation, effectively addressing the non-differentiable issue associated with using binary search to obtain the priors. It further enables the application of transfer priors in a more practical hard-label setting through subspace projection approximation. To the best of our knowledge, this is the first work to derive the expected cosine similarity between the estimators of the OPT family (i.e., Sign-OPT, Prior-Sign-OPT, Prior-OPT) and the true gradient, offering a theoretical guarantee for improved performance. [cAs8,hzgv,DsPD,z4Ag]

- **Significance**: The paper makes substantial contributions to adversarial machine learning by introducing a novel attack that leverages transfer-based priors in hard-label black-box setting. The paper demonstrates how this additional information from surrogate models can be effectively used. [DsDP]

- **Clarity**: The paper is well-written, with geometrical intuitions effectively illustrated through detailed figures. [DsPD,z4Ag]

- **Quality**: The methodology demonstrates technical soundness, particularly the idea of using transfer-based priors and then the approach to gradient approximation through closest projection on the subspace.  [DsPD]


**Major Updates in the Revised Version**: Based on the reviewers’ suggestions, we made the following major revisions in the updated version of our paper:

- **Revised Figure**: We have updated Fig. 1, Fig. 2(c), and Fig. 2(d) by refining the adversarial region and adding clearer labels to enhance readability and comprehension.

- **Add Theoretical Condition under which Prior-OPT outperforms Sign-OPT**: We added the theoretical condition under which Prior-OPT outperforms Sign-OPT, along with the detailed derivation steps to support this condition. Specifically, we add a new section (Appendix D) to include the detailed steps of the derivation of this condition. This revision is highlighted in blue text in the revised manuscript (line 318-322, and Appendix D).

- **Add Time Complexity Analysis**: We have included a theoretical analysis of the time complexity for gradient estimation in Appendix F.1, with the revisions highlighted in blue text.

---

### Meta-Review · Area_Chair_12xp · 2024-12-19

**Metareview:**

This paper studies black-box adversarial attacks in the hard-label setting, where only information on the top predicted class is available to the attacker. Devising successful attacks in this setting is difficult, as very little information is available to the adversary. Existing methods set up the problem via optimizing a problem that seeks the shortest path towards a region of a different label, but the resulting problem is discontinuous, incurs hard querying costs, and the resulting attacks might be limited. This work incorporates transfer-based priors to aid this optimization problem, in the form of the gradient of a surrogate model, using a subspace projection. The work theoretically analyzes these attacks, deriving the expected cosine similarity between the resulting estimations of the attacks and the true gradient. The paper also incorporates extensive numerical experiments, demonstrating state-of-the-art results.

All reviewers recognize that the strengths of this paper (the novelty in the transfer-prior formulation, theoretical analysis, and numerical results) are significant. Weaknesses noted include the lack of clarity in some figures and further clarification of details in the presentation.

Overall, reviewers are supportive of this paper, and I concur.

**Additional Comments On Reviewer Discussion:**

Reviewers engaged with the authors in the discussion period, resulting in extensive clarification from the authors and guidelines on how to improve the presentation of some portions of the work (e.g. Figures). The discussion has led to a better version of the paper.

---

### Decision · Program_Chairs · 2025-01-22

Accept (Spotlight)